

# Validation and comparison of cloud properties retrieved from passive satellites over the Southern Ocean

Arathy A.Kurup[1,2,4], Caroline Poulsen[3], Steven T. Siems[1,2,4], and Daniel J.V.Robbins[1,a]

[1]School of Earth, Atmosphere and Environment, Monash University, Melbourne, VIC 3800, Australia
[2]ARC SRI Securing Antarctica's Environmental Future, Melbourne, VIC 3800, Australia
[3]Bureau of Meteorology, Melbourne, VIC, 3001, Australia
[4]ARC Centre of Excellence for the Weather of the 21st Century, Monash University, Melbourne, VIC 3800, Australia
[a]now at: RAL Space, STFC Rutherford Appleton Laboratory, Harwell, OX11 0QX, UK

**Correspondence:** Arathy A Kurup (arathy.aneeshkumarkurup@monash.edu)

**Abstract.**

The clouds over the Southern Ocean (SO) play a vital role in defining the Earth's energy budget. The cloud properties over the SO are known to be different from their Northern Hemisphere counterparts. As a result, monitoring cloud properties over the SO, including macro- and microphysical properties, is of particular interest.

We analysed three passive remote sensing satellite datasets, the MODIS Collection 6.1, the AVHRR CMSAF CLARA-A3, and the AVHRR PATMOS, over the SO. We validated the cloud mask, cloud top height, and cloud phase for 2015 using Level 2 data retrieved from the passive sensors with active CloudSat-CALIOP sensors. We compared the effective radius and cloud optical depth amongst the three passive sensors datasets.

This research found that there are substantial uncertainties in cloud top height, cloud optical depth, and cloud thermodynamic
phase, over the SO. The extent of which varies depending on the cloud property and retrieval algorithm used. The cloud mask comparison revealed that only around two-thirds of passive sensor observations agree with active sensor observations, and in the case of AVHRR PATMOS the agreement is lower. In the comparison of cloud top height, a mean absolute bias of 0.65 km (AVHRR CMSAF), 1.03 km (MODIS), and 1.31 km (AVHRR PATMOS) was observed for single-layer cloud scenes cases. This mean bias increased to 1.86 km (AVHRR CMSAF), 3.22 km (MODIS), and 3.34 km (AVHRR PATMOS) for multilayered
cloud scenes. Ice phase dominates the multilayer cloud top thermodynamic phase in 2015, while liquid is the dominant top phase for single-layer cases. In general, the passive sensor and active sensor phases agree for liquid phase and ice phase except for AVHRR PATMOS, which frequently misidentified liquid phase as ice phase. In the comparison of cloud effective radius, it was observed that the disagreement between the passive sensors was greater in presence of multilayer clouds. The effective radius disagreement was largely higher for ice clouds. We found that the presence of sea ice strongly influences the retrieval of
cloud optical depth at high latitudes, with most passive optical depths higher over sea ice than over ocean. This work highlights the areas where passive cloud retrieval algorithms over the SO could be improved.





## 1 Introduction

The role of clouds is vital to the Earth's energy budget. The Southern Ocean (SO) is one of the cloudiest places on the planet. Climate simulations and reanalyses (Trenberth and Fasullo, 2010; Schuddeboom and McDonald, 2021; Zelinka et al., 2020) over the SO shows a persistent negative bias in shortwave (SW) radiation and an overestimation of the outgoing long wave (LW) radiation. Schuddeboom and McDonald (2021) compared Coupled Model Intercomparison Project Phase 6 (CMIP6) simulations with satellite data and identified substantial discrepancies in the representation of clouds, particularly with low-level clouds. This inconsistency leads to the simulations exhibiting an inverse correlation between the mean and compensating errors of SW cloud radiative effects (CRE), which is prevalent over the SO. Zelinka et al. (2020) examined the climate sensitivity of the CMIP6 for 27 global climate simulations, and found that the previous climate models had made the SW low cloud feedback component stronger. They found that a substantial shift in the SW cloud feedback from high-latitudes in the Coupled Model Intercomparison Project Phase 5 (CMIP5) models to extratropics has occurred in the CMIP6 models, especially over the SO. The uncertainty in the models demonstrates a need to understand Southern Ocean atmospheric and ocean properties, particularly cloud properties.

Compared against the clouds over the North Atlantic and the North Pacific, SO clouds have a higher probability of cloud glaciation (Davies et al., 2017), and the increased presence of supercooled liquid water (Huang et al., 2015; Morrison et al., 2011; Ovarlez et al., 2002). Davies et al. (2017) examined the plane parallel albedo bias, and concluded that SO clouds have a smaller heterogeneity bias than those over the Northern Hemisphere. Hu et al. (2010) examined Cloud-Aerosol Lidar with Orthogonal Polarisation (CALIOP; Young et al., 2008) observations and discovered that the supercooled liquid water retrieval at mid-latitudes, is dependent on the cloud top temperature and cloud top height, and that supercooled water clouds were more common over the SO. Huang et al. (2015) compared clouds over the North Atlantic against those over the SO, using a merged radar-lidar product and discovered that the presence of boundary layer clouds and mid-level clouds, with smaller droplet sizes was more prevalent over the SO. Mace et al. (2009) employed a merged radar-lidar product, to demonstrate that mid- and low-level multilayer clouds are present in more than half of the scenes over the SO (poleward of the ocean polar front), which is more frequent than their North Atlantic and Pacific counterparts. In a study conducted using upper air soundings from a collection of SO field campaigns, Truong et al. (2020) found the presence of multilayer clouds in over half the scenes over the high-latitudes of the SO. Further investigation uncovered a bias in the thermodynamic structure, specifically the frequency of the occurrence of multilayer clouds, in the ECMWF's fifth-generation atmospheric reanalysis (ERA5) (Truong et al., 2022) over the Southern Ocean. ERA5 more commonly simulated relatively thick single-layer clouds than inferred from the soundings. They demonstrated that the radiative transfer through these clouds is sensitive to cloud microphysics, as thin multilayer clouds can help reduce downward shortwave surface radiation over the SO. These findings highlight the unique characteristics of SO, and a need to understand the influence of multilayer clouds in retrieving cloud properties over the SO.

Previous research on cloud property retrievals, revealed a dependency on the presence of multilayer clouds for CRE retrievals, which was further reinforced in studies such as L'Ecuyer et al. (2019), Hinkelman and Marchand (2020) and Yost et al. (2023). L'Ecuyer et al. (2019) concluded that multilayer clouds contribute to enhancing LW radiation by 10.4 W/m$^2$ and



reducing SW radiation by 22.3 W/m$^2$ and are often misclassified as single-layer, thick ,mid-level-clouds, leading to global CRE biases. Hinkelman and Marchand (2020) evaluated the Clouds and the Earth's Radiant Energy System (CERES, Minnis et al., 2011) observations and Cloud-Aerosol Lidar with Orthogonal Polarisation (CloudSat) CALIOP data (Sassen et al., 2008 and Wang et al., 2013) against field observations from Macquarie Island Cloud and Radiation Experiment (MICRE; Marchand, 2020). They observed that there exists a bias of -10 W/m$^2$ in LW CRE and concluded that the factors contributing to this bias, includes low clouds, multilayer clouds, and precipitating clouds. The CERES Visible Infrared Imaging Radiometer Suite (VIIRS) observations, were evaluated with CALIOP observations by Yost et al. (2023) and it was found that multilayer clouds are one of the main contributors to errors in the retrieval of cloud top height.

Previous studies have conducted regional analyses to resolve some of the uncertainties over the SO, either typically emphasising comparisons with land observations, or concentrating on single satellite records (Ahn et al., 2018; Hinkelman and Marchand, 2020; Kang et al., 2021; McFarquhar et al., 2021; Xi et al., 2022). In recent years there have been several field campaigns over the SO: such as Clouds Aerosols Precipitation Radiation and atmospheric Composition over the Southern Ocean (CAPRICORN; McFarquhar et al., 2021) I and II; the Southern Ocean Clouds, Radiation, Aerosol Transport Experimental Study (SOCRATES; McFarquhar et al., 2021); Measurements of Aerosols, Radiation, and Clouds over the Southern Ocean (MARCUS; McFarquhar et al., 2019, McFarquhar et al., 2021); and MICRE. Ahn et al. (2018) compared cloud phase retrieved between Moderate Resolution Imaging Spectroradiometer (MODIS, Menzel et al., 2008), CALIOP, and in-situ observations and found that mixed-phase clouds are often underestimated by the satellite observations. Xi et al. (2022) also observed that the presence of mixed phases dominated the MARCUS campaign observations. Kang et al. (2021) examined MODIS, CERES and Himawari satellite observations against SOCRATES field observations. They found that a low bias of the cloud effective radius is due to compensating errors between non- or lightly precipitating cases and heavily precipitating cases. Despite these campaigns, the lack of long-term data continuity and spatial coverage remains a challenge.

Satellite observations are one of the primary tools for monitoring the weather and climate globally and are particularly important over the remote SO, where in-situ observations are sparse. Satellite instruments such as Advanced Very-High-Resolution Radiometer (AVHRR; Pavolonis and Heidinger, 2004), have been making observations since the early 1980s, which if sufficiently stable and accurate, could begin to provide information on the cloud response to a changing climate. Passive satellite cloud retrievals forward models typically assume a single-layer cloud. This results in large negative biases for retrievals of cloud top height particularly in the presence of multilayer clouds. Hence in this paper, we focus on the quality of satellite retrievals over the SO.

In this study, we analyse three well-known global passive satellite datasets: AVHRR Pathfinder Atmospheres-Extended (PATMOS-X, Heidinger et al., 2014), AVHRR Satellite Application Facility on Climate Monitoring (CMSAF; Karlsson et al., 2023b), and MODIS (Menzel et al., 2015) over the SO. We validate the cloud top height and phase with merged CloudSat CALIOP data, and compare the cloud optical depth and effective radius. We perform the analysis at the monthly level (L3), followed by an in depth analysis into instantaneous observations (L2). The results are analysed separately for single and multilayer clouds and a case study illustrates the main findings.



## 2 Data

The analysis considers cloud information from two different satellite instruments MODIS and AVHRR and 3 different algorithms/data providers. Active satellite instruments CloudSat-CALIOP, are used to validate the data. A brief overview of all the datasets used is given below.

### 2.1 Cloudsat-CALIOP P1 R05

The CloudSat and Cloud-Aerosol Lidar and Infrared Pathfinder Satellite instruments (CALIPSO), were launched in April 2006. The satellites are part of the NASA Afternoon Constellation, known as the A-train, a coordinated group of satellites crossing the equator from south to north within seconds of each other in an afternoon orbit. CloudSat exited the A-train in February 2018, to a lower orbit, due to technical difficulties, and CALIPSO rejoined it in September 2018 to form the C-Train. CALIPSO officially ceased its operation in September 2023, while CloudSat was switched to daytime-only observations in December 2021. CloudSat (Stephens et al., 2002) uses the Cloud Profiling Radar (CPR) to observe the vertical profiles of the clouds. The CPR onboard CloudSat has a vertical resolution of 500 m and horizontal resolution of 1.7 km operating at W-band (94 GHz). CALIPSO (Winker et al., 2003;Young et al., 2008) uses CALIOP , a lidar with two wavelengths of 532 nm and 1064 nm, to measure vertical profiles of cloud and aerosol. The CALIOP GEWEX cloud product (NASA/LARC/SD/ASDC, 2019), is the L3 product derived from temporal averaging of 5 km cloud merged layer products version 4, for a month.

The merged CloudSat-CALIOP product, is obtained from collocating the observations from both the CPR and CALIOP instruments. The temperature profile for the merged product is obtained from the European Centre for Medium-Range Weather Forecasts (ECMWF) and MODIS radiance data is used as supplementary information for cloud classification. The dataset is made available by the Cooperative Institute for Research in the Atmosphere (CIRA), Colorado State University. The CloudSat-CALIOP merged dataset includes the L2 retrieved cloud property data, and is provided under different product names for different cloud and radiation properties. In this study, we have used the 2B-CLDCLASS-LIDAR (Sassen et al., 2008; Wang et al., 2013), 2B-FLXHR-LIDAR (Henderson et al., 2013; L'Ecuyer et al., 2008) and the 2B-CWC-RVOD products (Leinonen et al., 2016). The 2B-CLDCLASS-LIDAR (hereby known as 2BCL) combines the advantages of LIDAR and RADAR to get an understanding of cloud vertical profile, cloud classification and cloud properties such as cloud layer top or cloud top height (CTH), cloud base height (CBH) and cloud phase (CPH). The 2B-FLXHR-LIDAR (hereby known as 2BFL) retrieval product, provides the broadband fluxes and heating rates from the CloudSat-CALIOP observations, which are useful in understanding the flux rates and optical depth of the clouds. The data products are available throughout the day for CTH, however, the revisit time is 16 days due to its orbit. The L2 data product with a spatial resolution of 5 km for 2015 is used for this comparison. For the climatology of 7 years (2009-2016), L3 monthly data from CALIOP, CAL_LID_L3_GEWEX_Cloud-Standard-V1 is used.



## 2.2 Moderate Resolution Imaging Spectroradiometer (MODIS)

The AQUA satellite, with a MODIS sensor onboard, was launched in 2002 and is also part of the A-train. The instrument has 36 spectral bands (wavelengths 0.4–14.4 $\mu$m). Every five minutes, the MODIS sensor scans an area of 2330 $\text{km}^2$. The MODIS-AQUA L3 monthly data product CLDPROP_M3_MODIS_Aqua (Platnick et al., 2019) is used for the analysis of L3 data. The data includes cloud properties such as CTH, cloud fraction (CFC), cloud optical depth/thickness (COD/COT) and cloud effective radius (CER). The MYD06_L2 (hereby known as MYD06; Platnick et al., 2015) contains data products retrieved from the MODIS instrument on the AQUA platform. The L2 data consists of 1 km resolution pixels, with cloud microphysical properties and 5 km resolution for all other retrieved properties. The MYD06 1 km retrieval, classifies pixels as cloudy and partially cloudy. We have considered clear, cloudy and partially cloudy for our analysis. The MYD06 collection uses the $CO_2$ slicing method and the Infrared Window Approach (IRW) to retrieve cloud top properties (Menzel et al., 2015).

## 2.3 Advanced Very-High-Resolution Radiometer

The AVHRR is a broadband radiometer with five or six spectral channels in the visible, near IR and IR. The instrument is a cross-tracking scanning radiometer with a resolution of 1.1 km. The AVHRR instrument was first launched in late 1978, and has since been operated continuously onboard NASA Polar Operational Environmental Satellites (POES), and more recently onboard the European MetOp platforms. The latest AVHRR has six channels, out of which only five can be transmitted at any given time. The NOAA-19 satellite is used for the L2 analysis.

### 2.3.1 AVHRR CMSAF

CLARA-A3 (The CMSAF Cloud, Albedo and Surface Radiation dataset from AVHRR data – third edition) is a climate data record derived from the polar-orbiting AVHRR instrument, of cloud, surface albedo, and surface radiation budget products for the period 1979-2024 (45 years; Karlsson et al., 2023c). The major advantage of this dataset is the global coverage and the period available for analysis. The CLARA-A3 edition (Karlsson et al., 2023b) contains retrieved cloud properties such as CTH, CPH and cloud water path (CWP). The CLARA-3 dataset is available for instantaneous, daily, and monthly periods and was released in mid-2023. The resolution of the data used in the analysis is 5 km data. The CLARA A3 uses an artificial neural network (ANN), trained on collocated AVHRR-CC data (Håkansson et al., 2018) to obtain cloud-top properties. In the case of cloud microphysics, traditional retrieval methods look-up tables (LUTs) are created based on the Nakajima-King approach Nakajima and King (1990).

### 2.3.2 AVHRR PATMOS

The AVHRR PATMOS provides a climate record of atmospheric cloud properties and brightness temperatures retrieved by merging the information from AVHRR, and the collocated High-resolution Infra-Red Sounder (HIRS, Foster et al., 2023). The algorithm was developed by the National Oceanic and Atmospheric Administration (NOAA) and the Cooperative Institute for Meteorological Satellite Studies (CIMSS). The climate data record includes 44 years of data (1979-2023), with a resolution



of 0.1°x 0.1° grid globally (Foster et al., 2023). The data set includes CTH, CER, COD, and ice water path (IWP). Multiple algorithms are referenced in the product, in particular the Advanced Baseline Imager Cloud Height Algorithm (ACHA; Calvert and Pavolonis, 2010), MODIS $CO_2$ slicing algorithm (Menzel et al., 2008), and Daytime Cloud Optical and Microphysical Properties (DCOMP; Walther et al., 2013). The ACHA CTH algorithm in this study uses 11, $12\mu$m observations along with the

155 split-window approach (Heidinger and Pavolonis, 2009), and data from the AVHRR Extended (CLAVAR-x). When AVHRR is combined with HIRS information, the $CO_2$ CTH can be determined.

## 2.4 Validation of polar satellites over the Southern Ocean

There have been numerous validation studies which we summarise here. Common to most studies, the validation results are generally reported for cloud globally and not as a function of specific regions. The validation of these data products over the

160 SO has been mostly performed using monthly products.

The MODIS products have been compared with CALIOP in several studies. Holz et al. (2008) found that CTH is underestimated by MODIS by 1.4 $\pm$ 2.9 km globally, when compared with the 1 km CALIOP product, and in the case of the 5 km CALIOP product, the difference is -2.6 $\pm$ 3.9 km. Yang et al. (2021) compared the CTHs of MODIS AQUA, MODIS TERRA, CALIOP, CloudSat and the Advanced Himawari Imager on Himawari-8 (HW8), with ground-based Ka-band zenith

radar (KAZR), at the Semi-Arid Climate and Environment Observatory of Lanzhou University (SACOL) site in NW China. They observed the minimum mean CTH differences between passive satellite sensors and KAZR are -0.35 km (-0.88 km for multilayer) for MODIS Terra, -0.87 km (-1.58 km for multilayer) for MODIS Aqua, -0.69 km (-1.70 km for multilayer) for Himawari-8 (HW8).

Karlsson and Johansson (2013) compared the CLARA-A1 (The CMSAF, Cloud, Albedo and Surface Radiation dataset from

170 AVHRR, data-edition one) dataset with CALIOP observations, and found that the overall bias in the CTH changes with the optical depth of the cloud with biases increasing with COD decreasing (< 0.35). The study also inferred that the CTHs of high clouds (> 8 km) are underestimated and boundary layer clouds (<2 km) are overestimated. Additionally, the research highlighted a substantial underestimation of cloudiness in polar regions, particularly during polar winter.

Foster et al. (2023) found that AVHRR PATMOS Version 6, has improved the performance of cloud property retrievals,

especially around the polar region from the previous version. The cloud fraction increased by 3% from Version 5 to Version 6, although the variation and seasonality of cloud optical depth across the satellite data decreased.

Karlsson and Devasthale (2018) evaluated four global cloud climate records, the International Satellite Cloud Climatology Project (ISCCP), European Space Agency (ESA) Climate Change Initiative (CCI) V3, CLARA-A2 and PATMOS-X V5, against each other and a CALIPSO dataset. The study uncovered large biases in cloud cover, notably in the southern polar

region, with major differences between the PATMOS-X and CLARA-A2 datasets in the cloud fraction. The PATMOS-X gives similar cloud amounts when compared to CALIOP observations. Furthermore, indications of orbital drift phenomena were detected in the PATMOS-X dataset, which was linked to the changing solar zenith and solar azimuth angle and surface temperatures resulting in increased cloud amount detection.



Chang et al. (2010) studied all the retrieval methods for CTHs from the Twelfth Geostationary Operational Environmental
Satellite (GOES-12) data, and found that there exists a negative bias of 2.5 km ±1 km for all CTHs retrieved as two-layer
and 4.5 km ± 2 km for cases with more than two layers. Sourdeval et al. (2016) found that the IWP retrieval quality can be
impacted by multilayer clouds and proposed an IWP retrieval method for multilayer conditions. They statistically analysed the
results to conclude that the new method is successful for retrieving IWPs ranging from 0.5 to 1000 g m$^{-2}$.

### 2.5 Multilayer clouds and detection

Mace et al. (2009) used the CloudSat-CALIPSO product to show that mid- and low-level multilayer clouds are present in
more than half of the scenes over the SO (along the ocean polar front), which is more often than their Arctic counterparts.
Similar studies, have found that ignoring the multilayer structure of clouds can lead to errors in the retrieval of cloud properties
such as cloud top pressure, cloud optical depth, and cloud fraction (Chen et al., 2000; Heidinger and Pavolonis, 2005; Joiner
et al., 2010). L'Ecuyer et al. (2019) found that the multilayer clouds contribute to enhancing LW radiation by 10.4 W/m$^2$ and
reducing SW radiation by 22.3 W/m$^2$ and they are often misclassified as single-layer thick mid-clouds leading to global CRE
of -17.1 W/m$^2$. The study also confirmed that the discrepancy in radiative fluxes, indicates that clouds with similar top of the
atmosphere radiative signatures can have varying impacts at the surface and on atmospheric heating.

Several multilayer cloud detection algorithms for global satellite instruments, with visible to infrared bands have been
developed (Baum et al., 1994; Joiner et al., 2010; Kawamoto et al., 2002). In general, the multilayer detection algorithms are
limited to thin-cirrus over low clouds, ocean areas or two-layer retrieval systems (Joiner et al., 2010; Sun-Mack et al., 2006).
MODIS, AVHRR, CERES have used CALIOP datasets to train decision tree algorithms to refine the multilayer cloud detection
algorithms (Li et al., 2011; Marchant et al., 2020; Sun-Mack et al., 2006; Simpsom et al., 2001).

In the multilayer detection flag for MODIS MYDO6 (Level-2 data; Marchant et al., 2020), the multilayer classification
agreement with CloudSat-CALIOP merged product is only 34% (Wang et al., 2016). New Artificial intelligence (AI) methods
have brought the accuracy for multilayer detection to around 60% for geostationary satellites such as Himawari (Li et al., 2022;
Ritman et al., 2022; Tan et al., 2022; Ritman et al., 2022). In this analysis we use the multilayer detection algorithm described
in Ritman et al. (2022).

## 3 Methodology

The comparison of the satellite cloud datasets has two main components. Firstly, we compare the Level 3 (L3) monthly cloud
products. Secondly, we analyse the Level 2 (L2) instantaneous cloud property data. Level 3 cloud products (CTH, CFC, CPH)
are compared for 7 years, from 2009-2016 for CALIOP (Global Energy and Water Cycle Experiment, GEWEX cloud prod-
uct; NASA/LARC/SD/ASDC, 2019), AVHRR CMSAF (CMSAF CLARA A3) and MODIS (CLDPROP_D3_MODIS_Aqua).
Note that the AVHRR PATMOS product does not have a L3 product and is therefore not included in the L3 analysis. In order to
further understand the origin of the retrieval bias in passive satellite cloud data, a comparison of Level 2 retrieved cloud prop-
erties from the satellite datasets is performed for one year (2015). The role of multilayer clouds in this bias is evaluated, along



with the role of sea ice. The study area selected for the comparison is 40°S to 70°S latitude and 180°E to 180°W longitude over the SO. The following sections describe in detail the methods used for the comparison and analysis of L3 and L2 data.

## 3.1 Level 3 monthly data comparison

In this analysis, the passive sensor type (flavour) of CALIOP L3 data (GEWEX product) is considered for CTH. The ECMWF
Reanalysis v5 (ERA-5, Hersbach et al., 2020) sea surface temperature (SST) data was analysed to understand if the surface temperature conditions were influencing the clouds during the time period. Seasonal climatology analysis was performed by considering the austral spring (September, October, November, hereby SON), summer (December, January, February, hereby DJF), autumn (March, April, May, hereby MAM) and winter (June, July, August, hereby JJA). Each seasonal composite was resampled onto a regular 0.25°x 0.25°resolution grid for all three sensors. The cloud properties analysed were cloud top height,
cloud fraction and fraction of liquid clouds. The comparison was conducted using the correlation of the passive sensors against the active sensor, along with statistical error metrics such as mean bias error (MBE), mean absolute error (MAE) and root mean square error (RMSE). The results and discussion for this analysis are described in section 4.1.

## 3.2 Level 2 orbit comparison

The L2 passive sensor data for 2015 from AVHRR CMSAF (CLARA A3), AVHRR PATMOS (Version 6), and MODIS AQUA
(MYD06 Collection 6) were collocated with active sensor data from the merged CloudSat-CALIOP retrieval. 4000 CloudSat-CALIOP granules were collocated with 20,000 MYD06, 365 AVHRR CMSAF and 365 AVHRR PATMOS granules. Each granule contained the cloud products such as CTH, COD, CER, CPH, latitude, longitude and common time of collocations. The dataset details are given in Table 1.

The CloudSat-CALIOP products 2BCL, 2BFL and 2BCR, were collocated with passive sensors, pixel-by-pixel for a time
matching of ±1min for MODIS MYDO6 and ±3min for AVHRR (Karlsson and Johansson, 2013) and ±5 km horizontal resolution. Figure 1 shows the collocation process for MYD06 and CloudSat-CALIOP data. The active sensors and passive sensor collocated datasets for AVHRR CMSAF (AC), AVHRR PATMOS (AP) and MODIS (MYD) were again time and geolocation matched against each other to obtain collocations for all four sensors for the year 2015 (Figure 1b). In order for a pixel to be considered a daytime pixel, the solar zenith angle has to be in between a range of 35°and 80°. A solar zenith angle
of < 80 degrees is considered day, 80–100°is considered twilight, and > 100°is considered night (Karlsson et al., 2016). We observed that our datasets, MYD and AC, had solar zenith angles ranging from 35 to 84°, while AP had a range up to 100°. Hence, a solar zenith angle mask with a range of 35°and 80°ensures a like-for-like comparison for the collocated datasets.

In order to understand the multilayer cloud influence, a simple empirical multilayer detection (Ritman et al., 2022) mask developed from 2BCL using the difference between two layers and cloud classification was applied to the collocated data. The
multilayer cloud mask condition uses the coarsest altitude resolution of 0.3 km, with a minimum gap to define layers of 0.1 km. The cloud layer in the L2 data was first analysed for cloud conditions and classified accordingly as no clouds, quality issues, or cloud layers. The number of cloud layers was assessed using the mask condition and classified as multilayer or single-layer



clouds. After applying the multilayer detection mask to the collocated pixels, cloud properties were analysed for the multilayer and single-layer conditions.

The cloud properties from the collocated three passive sensors (MYD06, AC, AP) and the active sensor (CloudSat-CALIOP) were examined and evaluated against each other. Vertical profiles of case studies with retrieved cloud properties (CTH, COD, CER, and CPH) were also investigated to understand the sensor retrievals. The passive sensor CTH and CFC were validated statistically with the active sensor using all the data from 2015. Furthermore, the cloud properties retrieval of COD and CER were compared as a function of the thermodynamic phase (CPH) and multilayer flag. A sea ice flag obtained from AVHRR

PATMOS data, was used to further understand the potential effects of sea ice on the retrievals. The evaluation of cloud mask performance skill scores for active and passive sensors was conducted using the Kuiper skill score (KSS; Hanssen and Kuipers, 1965). True positive rates (TPR) and False positive rates (FPR) were calculated using the KSS formula as shown in the equation

$$KSS = TPR - FPR = \frac{tp}{(tp + fn)} - \frac{fp}{(fp + tn)}, \tag{1}$$

where $tp$ is the number of true positives, $fp$ is the number of false positives, $fn$ is the number of false negatives, and $tn$ is the number of true negatives. A KSS of 1 indicates a high skill score and zero indicates no skill.

Each data set has different definitions of cloud mask and cloud mask uncertainty. The cloud mask for AP has categories including clear sky (0), probably clear (1), probably cloudy (2) and cloudy (3). This was converted to clear (0), encompassing clear sky and probably clear, and cloudy (1), combining probably cloudy and cloudy, for comparison with the other satellite

cloud masks. Similarly, MYD06 cloud mask, confidently clear (11), probably clear (10), uncertain (01) and cloudy (00) were classified into cloudy (0) and clear (1). Meanwhile, the AC had only two classifications: clear (0) or cloudy (1).

We analysed approximately 1.2 million pixels for each sensor in the L2 data. Upon applying the multilayer mask, we observed that approximately 25% (300000) of the collocated data consisted of multilayer scenes across the SO. Figure 2 shows the final collocated swaths used in the analysis.

The following sections discuss the results for the monthly data and pixel-wise data comparison for the passive and active sensors.

## 4  Results and Discussion

### 4.1  L3 monthly seasonal comparison

The L3 monthly data cloud properties for three sensors: AVHRR CMSAF (CLARA A3), MODIS (CLDPROP _D3_MODIS_Aqua);

and CALIOP (CAL_LID_L3_GEWEX_Cloud), were compared for 7 years (2009-2016). The dataset details used for comparison are given in Table 1.



### 4.1.1 Cloud Top Height

Figure 3 shows the difference in CTH for the top most layer between the active and passive sensors, alongside the SST for the full period and by season. When the data is analysed as a function of latitude band in the ocean region from $62°$ S to $70°$ S, both passive sensors generally overestimate the CTHs compared with the active sensor CTHs, except during winter (JJA), when only MODIS overestimates the CTH. This region is typically covered by sea ice during this time of year. The SST also shows variation during corresponding season with a low SST in winter and a high SST in summer. However, there does not seem to be a strong correlation with the differences. In the poles over the land area irrespective of the seasons, AVHRR CMSAF tends to underestimate the CTHs, while MODIS tends to overestimate the CTH. As per Noel et al. (2018), the cloud amount detected by the active sensors and passive sensors varies overall when comparing L3 monthly data. As CALIOP is an active sensor, it is more sensitive to high-thin clouds (CTH > 8 km) and the vertical distribution of the clouds. The presence of atmospheric temperature inversions may cause the passive sensor to either underestimate or overestimate the CTHs in the boundary layer (Marchand and Ackerman, 2010). Thus, factors such as the presence of multilayer clouds, differences in sensor sensitivity, differences in sensor resolution, and differences in retrieval algorithms may account for some of the poor agreement.

Table 2 summarises the comparisons, with mean CTH values in Table 2a and the statistics for the comparison with CALIOP in Table 2b by season and overall. It can be observed that the active sensor correlation with the passive sensors for the overall monthly CTHs is low with values ranging from 0.35 for CALIOP-AVHRR CMSAF (C-A) and 0.32 for CALIOP-MODIS (C-M). The CTH difference between the C-A sensors for DJF ranges from -1 to 1 km for the study domain, while for C-M it ranges from 0 to 4 km. The MBE for passive with the active sensor is 0.46 km for C-A and 1.12 km for C-M. The RMSE for C-A is 2.33 km; for C-M, the value is 2.60 km; and for A-M, it is 1.23 km, which is consistent with both correlation and MBE trend. Overall, the MBE between the MODIS-AVHRR CMSAF (A-M) passive sensors is larger for the CLARA-A3 dataset (0.95 km), but the correlation is better (0.67). Overall, there is a positive mean bias error for the passive sensors AVHRR CMSAF and MODIS CTHs.

The seasonal comparison of the C-M data over the course of a year shows that the austral autumn (SON) has the weakest correlation (0.28) and MBE (1.25 km). Meanwhile, the inverse is true for the austral summer (DJF) with a correlation of 0.36 and MBE of 1.05 km with > 1 km, positive bias over the mid-latitudes, and smaller bias ($\pm 0.5$ km) over the high-latitudes. Another intriguing observation is that in austral winter (JJA), MODIS overestimates (> 1 km) the CTHs in the high-latitude and underestimates (> 1.5 km) in the mid-latitudes with an MBE of 1.25 km. The seasonal comparison of CTHs for C-A shows that the correlation is highest in the austral spring (SON) for C-A (0.34) and lowest in the summer (DJF -0.15). AVHRR CMSAF tends to overestimate the CTHs in the mid-latitudes and underestimate them in the high-latitudes during winter, with an overall MBE of 0.25 km. In all other seasons, AVHRR CMSAF tends to overestimate the CTHs across the region, with the exception of Antarctica.

The passive sensor comparison shows that DJF has the highest correlation at 0.68, whereas JJA has the lowest at 0.45. Additionally, DJF exhibits the lowest mean bias error of 0.09 km for C-A among all seasons and 0.98 km for C-M. This may be due to the seasonality of cloud cover over the SO, as JJA tends to have more high-clouds (Bromwich et al., 2012) and passive




sensors fail to detect the cloud due to poor sensor sensitivity to optically thin high clouds. The results indicate that the AVHRR
CMSAF generally overestimates the CTHs compared to CALIOP. Conversely, MODIS tends to underestimate the CTHs.

### 4.1.2 Cloud Fraction

The comparison of CFC for the active sensor and passive sensor data overall and seasonally is shown in Figure 4. The overall
7-year mean CFC values for the SO were 0.82 for CALIOP, 0.86 for MODIS, and 0.85 for AVHRR CMSAF. This is consistent
with the findings of the review study conducted by Bromwich et al. (2012). The cloud fraction for all sensors, both passive
and active, has more cloud cover over the mid-latitudes (40°S to 60°S) than over the high-latitudes (> 60°S). In general, the
cloud cover decreases (< 70%) towards very high-latitudes near the coast of Antarctica for all sensors, which is also consistent
with findings of Bromwich et al. (2012). This decrease, especially in winter, is likely due to the presence of sea ice near the
Antarctic coast. This inhibits the flux of water vapour (surface evaporation), as presented in studies by Frey et al. (2018) and
Wall et al. (2017). The high cloudiness for this area can be attributed to the frequent synoptic-scale and mesoscale depressions
and intense cyclonic activity around the Antarctic Continent (Carrasco et al., 2003; King and Turner, 1997; Simmonds et al.,
2003). In general, the passive sensors have higher cloud coverage than CALIOP observations for the higher latitude.

The detailed mean and statistics for the CFC comparison are given in Table 3. It's intriguing to note that the correlation
between the passive sensors (AVHRR CMSAF and MODIS) and the active sensor (CALIOP) is low (0.52 and 0.48) over
the whole period. On the other hand, the correlation between the passive sensors is higher (0.87), which suggests that the L3
passive retrievals may have a different systematic bias. Bromwich et al. (2012) also observed a difference in the cloud cover
for the passive and active sensors, they attributed this to the spatial resolution issues of the active sensor when compared to
the passive sensor. The results of this analysis indicate that the active sensor cloud cover was less than the passive sensors, by
approximately 4% for MODIS and AVHRR CMSAF. Note that we are comparing against the CALIOP passive sensor adjusted
data set.

We found that CFC values are slightly lower in JJA (winter), with a mean of 0.83 and 0.82, respectively, compared to DJF
(summer) when we compared AVHRR CMSAF and MODIS L3 data. In the case of MODIS, the high latitude CFC for JJA
is approximately 60%, the lowest amongst all the sensors. This can be attributed to the presence of sea ice. For the active
sensor, JJA has the lowest CFC (0.81). It's also observed that there is a notable difference in cloud cover over land for different
passive sensors. MODIS overestimates the cloud cover over land; on the other hand, AVHRR CMSAF underestimates the cloud
cover compared to the active sensors. In the RMSE and MAE seasonal analysis for all sensor comparisons, it was found that
JJA has the highest value for both amongst all seasons. However, for the MBE, it's the opposite, with JJA having the lowest
value (≈0.01 km) for all sensors. The correlation also follows the same trend as MBE errors: low in JJA and high in DJF. For
comparisons among the passive sensors, the agreement of pixels identified as cloudy by both sensors is more consistent, and
the root mean square error (RMSE) is low for all seasons. All of the comparison's seasonal findings are consistent with the
conclusions of Bromwich et al. (2012).





The results indicate that CALIOP detects fewer clouds than the passive sensors. This could be due to a difference in resolution or in the approach taken to make the L3 data set comparable with passive sensors. The total cloud cover for the SO was found to be around 86% (MODIS), 85% (AVHRR CMSAF) for passive sensors and 82% for CALIOP.

### 4.1.3 Cloud fraction for liquid clouds

The overall cloud fraction of liquid water clouds (hereafter CFL) and seasonal distributions are illustrated in this Figure 5. For the entire year, CALIOP, AVHRR CMSAF, and MODIS show the presence of liquid clouds at around 50%. In terms of the correlation for the CFLs, the passive sensor comparison is better (0.71) than the active sensor comparison (0.37 C-A and 0.44 C-M). Lachlan-Cope (2010) observed that over Antarctica, ice clouds are more prevalent and along the coast of Antarctica, mixed-phase clouds are more prevalent. Other studies also observed that mixed-phase clouds are prevalent over the SO (Ahn et al., 2017; Huang et al., 2012; Mace et al., 2021) as they are usually misclassified more over the SO. From Figure 5, we can conclude that AVHRR CMSAF is identifying more liquid clouds over the high-latitudes than other products, while other passive products classify the clouds as ice clouds or mixed-phase clouds. In the A-M seven-year monthly comparison, the overall correlation stands at a higher level (0.71) compared to both passive-active comparisons. The comparison's mean and statistical analysis are given in Table 4. The comparison shows that AVHRR CMSAF primarily overestimates the CFL, while MODIS tends to underestimate it when compared to CALIOP. However, around the high-latitudes, towards the coast of Antarctica the AVHRR CMSAF underestimates the CFL when compare to CALIOP. In the case of seasonal comparisons, the correlations for passive sensors against active sensors are low overall in all seasons. For all the sensors, the CFL is consistently higher in DJF and the lowest in JJA. When we look at the MBE, AVHRR CMSAF has a negative bias for all seasons, while MODIS has a positive bias for all seasons compared to CALIOP. AVHRR CMSAF exhibits the smallest bias in JJA and the highest in DJF, while MODIS displays the smallest bias in DJF and the highest in JJA. This suggests a discrepancy in cloud classification during the liquid phase between the sensors in DJF. This disagreement can be attributed to the presence of supercooled liquid in DJF (Huang et al., 2015; Bodas-Salcedo et al., 2016). The C-A seasonal comparison reveals that the AVHRR CMSAF overestimates the liquid cloud cover across the study region in DJF and JJA, with the exception of Antarctica, where it underestimates the amount of liquid cloud. However, in SON and MAM towards the latitudes > 55°S, both sensors underestimate the CFL. The presence of sea ice explains this difference in SON and MAM. In summary, the comparison of the L3 data for the CTH shows a bias in the retrieval between passive sensors, underestimating the CTH by around 1 km compared to the active sensor: the mean bias error was 0.96 km (C-A) and 1.12 km (C-M). The study of L3 CFC and CFL revealed that AC overestimates the amount of clouds compared to observations made by MODIS and CALIOP, with the overestimation increasing during the winter months.



## 4.2 Results for Level 2 pixel-wise comparison

### 4.3 Case study

Figure 6 shows the swath for the (11μm) brightness temperature from MODIS (MYD) and merged CloudSat-CALIOP track,
for a Case Study on 10-01-2015 at 22:50 UTC. The collocated data for CTH, COD, CER, and CPH for the swath are illustrated
in Figure 7. The Figure displays MODIS (MYD) values in red, AVHRR PATMOS (AP) in orange, AVHRR CMSAF (AC) in
black, and the CloudSat-CALIOP merged data (hereby known as CC) in blue. The top panel shows the CTH for all sensors
(AC, AP, MYD, and CC), as well as the vertical profile of COD from the CC dataset (2BFL). The CTH for CC at a COD limit
of 0.5 is also shown (cyan) to illustrate the potential differences in optical depth sensitivity between the passive and active
sensors. The second panel illustrates the COD for the four sensors, along with the multilayer mask derived from CC, with red
indicating a multilayer and yellow indicating a single-layer. The COD shown on each product uses the passive sensors' 1.6
μm channel. We chose this channel because it is common to all passive sensors. The third panel displays the CER for passive
sensors, as well as the sea ice mask used in the AP retrieval, with cyan indicating sea ice, red for snow, and violet for no snow
or ice detected. The bottom panel shows the cloud phase distribution for the clouds retrieved by various sensors, with red as
clear sky, blue as water phase, purple as mixed phase, orange as ice phase, brown as supercooled liquid, pink as undetermined
(CC and MYD), and grey as undetermined after phase retrieval of MYD06.

The case study shows 61°S to 70°S and 120°W to 135°W. This example illustrates the highly variable nature of the cloud
field across the Southern Ocean. Our discussion focuses on four sub-sections: patchy, optically thin mid-level clouds over
optically thick boundary layer clouds (Figure 7a[i]); multilayer clouds (Figure 7a[ii]); thick frontal clouds (Figure 7a[iii]); and
thick upper-level clouds over thick low-level clouds (Figure 7a[iv]). In the first panel, the CTH ranges from 0 to 12 km and
shows substantial differences between the CTH of passive and active sensors, with both underestimation and overestimation
by the passive sensors. Figure 7a[i] shows an optically thin patchy mid-level cloud over an optically thick boundary layer of
clouds, in a region defined as sea ice by the AP sea ice flag. The MODIS retrieval fails to detect the mid-level cloud and
underestimates the CTH. However, the AC detects the mid-level cloud and as it is optically thin over a lower-layered cloud,
it places the CTH between the two layers. In the case of AP retrieval, due to the presence of sea ice or a flaw in the retrieval
logic, it overestimates the CTH and places the height near the troposphere layer, which is relatively low at this latitude. While
the COD of MYD, AC, and CC are similar, the COD of AP is significantly lower for a large section, possibly due to its
misclassification as ice. The CC COD values may be lower due to the presence of an optically thick boundary layer cloud and
attenuation. In the CER comparison for passive sensors, there is general agreement when it comes to single-layer cloud scenes
and greater differences in the presence of multilayer clouds.

The second subsection (Figure 7a[ii]) shows a multilayer cloud structure with a cirrus cloud and boundary layer clouds. In
this region, the passive sensors underestimate the CTHs by 4 km to 6 km for MYD, 3 km to 5 km for AC, and approximately 5
to 7 km for AP, when compared to the CTHs retrieved by the active sensors. This clearly demonstrates that the passive sensors,
regardless of retrieval methods, generally in sensitive to the optically thin cirrus cloud.





The next subsection (Figure 7a[iii]), an optically and geometrically thick layer of clouds shows the passive sensors MYD and AC overestimating the CTHs by 1 km to 2 km and AP underestimates the CTH by 1 to 3 km. This difference could be due to the different Numerical weather prediction (NWP) reanalyses, used for temperature profiles, which can vary substantially according to altitudes, particularly at high latitudes and difficulty in identifying the tropopause(Wu et al., 2012; Müller et al., 2018).

When a thick upper-level cloud is on top of a thick low-level cloud (Figure 7a[iv]), both MYD and AC can accurately retreive the top layer CTHs. However, when the cloud becomes optically thin, the differences between the CTHs for MYD and AC get bigger. The AP's sensitivity to the cloud layers is low, leading it to miss the top layer and only detecting optically thick lower layer clouds.

       In the second panel (Figure 7b), the cloud optical depths for each retrieval are compared. The COD of CC (2BFL) exhibits
high variability when compared to the COD of passive sensors. The COD has large differences among the sensors. We can see that, where there is a multilayer structure (Figure 7a[ii]and [iv]), CC COD is substantially higher than the other sensors. In single-layer areas, the COD of MYD and AC are more in line with the active sensor measurements. The AP COD varies throughout the scene, with a significant difference from other passive and active sensor measurements. The second panel displays the multilayer mask that CC developed. It identifies the areas with multilayer clouds quite well. The optical thickness
of the topmost layer cloud at 67.5°S is around 0.01 to 0.1, and the convective cloud has an optical thickness ranging from 1 to 10, and the multilayer mask successfully identifies it as a multilayer cloud.

       The third panel (Figure 7c) shows the CER compared for the 1.6 $\mu$m channel retrievals for passive sensors and the sea ice flag. The CER values of AC and AP range from 0–40 microns, and those of MODIS range from 0–60 microns. The CER values from different instruments show little agreement, especially the AP CER, which frequently deviates from the norm. When there
are single liquid layer clouds, the sensors produce reasonable results in CER observations. When multilayer clouds are present or there is an optically thick ice cloud, the CER disagreement is largest in the sensors. MODIS shows the largest differences in the presence of optically thick clouds or multilayer clouds. Additionally, the CER observations reveal missing data in the case of AC that are not evident in other cloud properties. The panel displays the sea ice flag from AP, revealing the presence of sea ice over half of the scene.

The cloud thermodynamic phase is shown in the fourth panel (Figure 7d). For CPH, each retrieval algorithm uses a slightly different phase classification scheme. All sensors share ice and water as common classes, with a clear sky for MYD, a supercooled liquid for AP, and a mixed phase for CC. The CC phase was retrieved from the cloud top. The agreement between the sensors overall is poor, but in the case of a single-layer structure, most sensors except AP agree on liquid retrieval between 69.5°S to 68.5°S. The discrepancy of AP's misclassification of phase here (69.5°S to 68.5°S), is likely due to AP identifying
CTH as a high cloud rather than a boundary layer cloud. AP classifies the majority of the clouds as ice clouds and supercooled liquid clouds, whereas AC classifies the clouds as ice or water clouds. MYD classifies the clouds as water, ice, or undetermined, and the active sensor CC allocates the clouds as water, mixed, and ice. The differences indicate interdependence in cloud property retrieval, such as CTH for AP. In the case of multilayer clouds, the phase of the cloud detected by the sensors is taken as the phase of the top layer, and in the case of passive sensors, often thin cirrus ice clouds are missed.





To summarise, we observe significant variations in the retrieved cloud properties in the presence of multilayer clouds, with passive sensors underestimating the retrieved cloud properties (CTH, COD) compared to active sensors. The dependency on the optical depth of the topmost cloud layer, the penetrative property, and the sensitivity of passive sensors are seen in this case study.

The case study demonstrates that cloud property retrievals differ based on cloud type, retrieval algorithm, and sensor sensitivity. The case study indicates that single-layer boundary layer clouds have smaller biases, while high-level clouds have larger biases in cloud property retrievals. Multilayer clouds exhibit the greatest biases among all cloud types, as passive sensors are generally less sensitive to high optically thin clouds.

## 4.4  L2 cloud mask

The cloud mask products from the passive sensors are evaluated against the CloudSat-CALIPSO product (Table 5). The true positive rate (TPR) is similar for AVHRR CMSAF (AC 0.94) and AVHRR PATMOS (AP 0.95) and lesser for MODIS (MYD 0.77). The false positive rate (FPR) vary considerably among the products, 0.22 for AC, 0.07 for MYD, and 0.51 for AP. The Kuiper Skill scores are relatively similar, 0.71 for AC, 0.70 for MYD, except for AP, 0.42. The higher resolution of MODIS and AVHRR CMSAF may contribute to their higher KSS compared to AVHRR PATMOS.

According to Karlsson et al. (2023a), when the global cloud masks of the CLARA A3 and CALIPSO (5 km cloud product) were compared, for the years 2006–2015, the overall hit rate was 0.82 and the KSS score was 0.68. The hit rate for high latitudes in both hemispheres was 0.85, and the KSS score was 0.69 and for polar regions, the hit rate was 0.69, and the KSS score was 0.49. The KSS score from our analysis comparing AC cloud mask and CC is 0.70, which is similar but the hit rate is higher (0.94). These differences can be attributed to changes in the period, the study area considered, and the active sensor data considered. Furthermore, differences in cloud masking algorithms and validation methods may contribute to the variations in scores. The hit rate for MYD and CC comparison is the lowest (0.77); on the other hand, the FPR is also the lowest amongst the sensors (0.07). It was observed that the hit rate and KSS score was influenced by the presence of sea ice (Figure 9).

## 4.5  L2 cloud top height comparison

Figure 8 shows a 2D histogram plot of passive satellite CTH with active (CC) CTH for the year 2015. A summary of the comparison of the average CTH with and without a COD limit > 0.5 is given in Table 6.

The average CTH of CC is higher without the COD limit, indicating that the high thin cloud is removed when this threshold is applied. In general the average height of passive sensors is higher than the CC with CTH threshold >0.5 and lower than the CTH without threshold. Analysis of the full dataset (2015) reveals mixed agreement between passive and active sensors, with AC (5.46 km) demonstrating better agreement than CC CTH without threshold (5.12 km), and AP (4.22 km) and MYD (4.12 km) demonstrating better agreement with CC CTH with COD > 0.5 (3.91 km). For both CC CTH (with and without COD > 0.5), single-layer clouds (3.41 km and 3.84 km), passive sensors exhibit good agreement.

Multilayer cloud identification was applied using a mask derived from CC CTH. The results for the multilayer cloud are unsurprisingly less accurate and inconsistent. The AC CTH (7.18 km) shows better agreement with the CC without the threshold





(8.4 km), indicating that it is more sensitive to high thin clouds than MYD (5.42 km) and AP (5.32 km), which show better agreement with the CC CTH COD > 0.5 threshold (4.81 km).

Figure 8 shows a 2D histogram comparison of passive satellite CTH with active (CC) CTH for 2015. The CC CTH plotted was CTH with a COD > 0.5. Each row shows the results for a single passive satellite: AC is the top row (Figure 8 a,b,c), MYD is the second row (Figure 8 d,e,f) and AP is the bottom row (Figure 8 g,h,i), and each column shows the result, from left to right: the full dataset (2015), single-layer clouds, and multilayer clouds.

    A number of features become apparent in Figure 8 The MYD and AP plots display a horizontal band, signifying the passive
sensor's underestimation of the CTH, which aligns with the earlier analysis. In general, the CTH of low clouds are overestimated by the passive sensors. The passive sensors underestimate the CTH of high clouds in MYD and AP. An explanation for the overestimation of low CTH is the effect of atmospheric temperature inversion, where the cloud brightness temperature taken by the passive sensor from the NWP reanalyses profile might be for a lower height than the actual CTH (Figure 7a[i]).For clouds that have underestimated the CTH, there are two possible reasons for this. Firstly, the clouds are thin and the retrieval
does not properly account for the extinction of the cloud and a contribution to the TOA radiance from the surface (Fu et al., 2017). Secondly, the passive sensor IR channels penetrate approximately an optical depth into the clouds, as seen clearly in case study Figure 7a[ii].

    Additionally, the differing performance of the passive data sets is in part due to different algorithm approaches. The AC CTH has been derived by training a neural network with collocations of CALIOP cloud top heights. While the MYD retrievals uses
a physical model based on $CO_2$ slicing and IR window approach (Menzel et al., 2015) for the retrieval of CTH. In AP, CTHs are retrieved using physical models, the ACHA algorithm and the $CO_2$ slicing method (Menzel et al., 2008). As the results of the $CO_2$ slicing method were poor, we have considered only the ACHA method CTHs. The ACHA methods employs a combination of IR channel observations and the split-window approach (Heidinger and Pavolonis, 2009), to retrieve the CTHs.

    The analysis of L2 CTH properties has shown significant differences between products. There are differences not only
495 between retrievals using the same instrument (AP and AC), but also between retrieval algorithms and between different instruments (MYD). The agreement between all sensors was reasonable for cloud top heights in the case of single-layer scenes. However, scenes classified as multilayer exhibited a lower level of agreement. The general findings on CTH are in agreement with previous studies. (Hollmann and Wetterdienst, 2020; Yang et al., 2021). Mitra et al. (2021) observed the maximum CTH uncertainty for MODIS (L2) for a single-layer unbroken low cloud of $0.540 \pm 0.690$ km. In the MYD06 collection, CTH bias
with CALIOP is reduced to 0.197 km for low-level boundary water clouds (Baum et al., 2012).

    In our comparison of the L2 collocate data of active and passive sensors, the mean bias in each sensor for CTH varies with MODIS (MYD06) 1.36 km, AVHRR CMSAF (CLARA A3) 1.07 km, and AVHRR PATMOS (V6) 1.21 km. When it comes to multilayer classified pixels, the passive sensors underestimate the CTHs. Given that the total number of multilayer identified scenes represents one-fourth of the total scene, the bias is significant. Overall, the AC neural network retrievals performed the
505 best, delivering higher sensitivity to thin clouds and performing significantly better for multilayer and high clouds.





## 4.6 L2 Cloud Effective Radius Comparison

In this section, we looked at the CER of the passive sensors as a function of CPH, to see how the values changed depending on the thermodynamic phase and the presence of single or multilayer clouds. The passive sensors categorise thermodynamic phases into various classifications according to their retrieval algorithms. The categories utilised by the sensors are presented in Table 7. A comparison between the different sensors AC, MYD, and AP, is illustrated in Figure 9, over a year, and for SL, and ML cases. Table 8 shows the statistical comparison for water, ice, and supercooled liquid phases.

The AC CER with respect to CPH for the whole year shows similar values for all thermodynamic phases. It can be observed that the liquid identified phase and supercooled liquid identified phase have a mean at around 14 $\mu$m for all cases, yearly, SL, and ML. In this analysis CPH classification of water, fog, and supercooled liquid are considered liquid. Ice, cirrus, and overlap are considered ice for the general CPH classification. When the general CPH classification is considered, it was found that for SL cases more liquid clouds were present than ML cases, and ice was more prevalent for ML cases. Another interesting observation is the strong presence of supercooled liquid in AC phase distribution with a mean of 14.62 $\mu$m for full year cases. For AC SL cases, supercooled liquid and liquid clouds are predominant, and for AC ML cases, its overlap and cirrus clouds are more prevalent.

The MYD (Figure 9) CER shows that the majority of the pixels phase as liquid/water clouds with a mean of 15.59 $\mu$m. In the case of multilayer clouds the mean is 16.38 $\mu$m. CER classified as ice clouds have a mean of 34.72 $\mu$m for SL and a mean of 26.54 $\mu$m for ML clouds. MYD has a phase classification specified as "undetermined liquid retrieval" (hereby ULQ), where liquid cloud retrieval was attempted but failed to classify as liquid cloud. Previously, this was classified as a mixed class in MYD collection 5 (Platnick et al., 2015). The ULQ has a mean of 29.32 $\mu$m. The presence of liquid clouds is more common for SL cases than ML cases. In the case of ML clouds, more ice clouds are present in the CPH with a peak at 25 $\mu$m.

AP has the majority of its CER classified as ice phase for full year, SL, and ML cases. In AP, the CER classified as supercooled liquid is present predominantly in the liquid phase and has a mean of 14.46 $\mu$m and has a higher frequency of occurrence in SL cases, compared to water phase. The ice phase has two peaks in the SL cases and full-year cases, one at 15 $\mu$m and one at 35 $\mu$m, and has a mean of 19.53 $\mu$m and 20.31 $\mu$m. The ML cases has a single peak with a mean of 19.23 $\mu$m. Another interesting feature is the presence of supercooled liquid in all cases, with the mean approximately 15 $\mu$m.

The results show that the CER distribution varies significantly between sensors for different thermodynamic phases. In the case of ML clouds, the dominant phase is ice. The presence of supercooled liquid is significant in the SO for both ML and SL clouds for both AC and AP retrievals, which is in line with previous findings (Huang et al., 2015; Bodas-Salcedo et al., 2016). There is a need to identify the clouds either as a mixed-phase or supercooled liquid class in passive sensor classification. In the case of AP CER, the ice phase clouds have bimodal distribution, which may indicates the misclassification of phase (Figure 9). The presence of sea ice also contributes to the misidentification as seen in the Case Study(Figure 7d[iv]). In general, the CER for the passive sensors agree when it is single-layer clouds and has large disagreement between sensors in the presence of multilayer clouds. From our analysis, it is evident that multilayer clouds are a major factor in phase misidentification which is





inline with Yost et al. (2023). Yost et al. (2023) indicated in their study that multilayer clouds are one of the major contributors
to phase misclassification.

## 4.7 L2 cloud property comparison as a function of surface type

In this section, we investigate the distribution of cloud properties as a function of sea ice coverage. The sea ice flag used in
AVHRR PATMOS retrieval was applied to all the collocated pixels. The distributions of the cloud properties (CER, CTH, and
COD) were analysed as a function of sea ice coverage for latitudes > 60°S, as shown in Figure 10. The narrow latitude range
was selected to reduce differences in microphysics caused by different meteorological regimes. The MODIS (MYD) property
is shown in red, AVHRR CMSAF in black, and AP in yellow. Cloud properties such as COD, CER, and CTH are compared
for sea ice detected and non-sea ice detected, and the mean and median are given in Table 9. The mean COD for AC was 25.98
and 19.19, for MYD it was 23.30 and 20.46, and for AP it was 28.30 and 17.7 for sea ice and non-sea ice cases, respectively.
The mean optical depth of the retrieved pixels increased in the presence of sea ice for all sensors. The results disagree with the
findings from Frey et al. (2018) and Palm et al. (2010). They show that COD is generally higher over ocean than over sea ice.
It is hypothesised that the high albedo of the sea ice causes a systematic positive bias in the retrieval algorithm. CER and CTH
also showed similar positive bias but the bias was much smaller than the differences observed for COD (Table 9; ≈1 $\mu$m for
CER and ≈ 0.3 km for CTH).

The seasonality of sea ice presence was also examined to understand the influence of sea ice on the sensor's observations.
We observed that the disagreement between the retrievals is higher in MAM and lowest in DJF when there is no sea ice present.
In the case of MYD, the COD is less than 50 for all seasons, and for AP, less than 80. For all seasons, the range for CER is
between 0-35 for AP, 5-65 for MYD, and 5-45 for AC. On the other hand, for CTH, the range varies according to the season,
with the presence of low clouds (< 2 km) high for MYD and AP in JJA and all sensors in SON and AC. In DJF and MAM, the
amount of mid-level clouds (3-6 km) was higher, while both MYD and AP had a higher concentration of low CTHs (< 2 km).
It can be concluded that in the case of CTH and CER, the seasonality is not an important factor, while for COD, the seasonality
and presence of sea ice are important factors in the derivation of the product.

## 5 Summary and Conclusion

In this study we have compared and validated the cloud property retrievals from merged CloudSat-CALIOP (CC) and MODIS-
AQUA (MYD06), AVHRR CMSAF (CLARA A3), and AVHRR PATMOS (V6) over the SO for monthly (L3) and instanta-
neous (L2) data. The results showed significant differences in cloud top height, cloud fraction (cloud mask), optical depth,
thermodynamic phase, and cloud effective radius.

For the Level 3 comparison, when compared to CALIOP CTHs for the top layer, we found that MODIS generally tends
to underestimate the CTHs and AVHRR CMSAF tends to overestimate the CTHs. The monthly observations of the CTHs
revealed significant biases and poor correlation within the dataset. The observed cloud fraction for passive sensors was higher
than for active sensors, especially towards the mid-latitudes (40°S to 60°S). In comparison to CALIOP CFC, the AVHRR





CMSAF CFC was > 20% and the MODIS CFC was > 10%. On the other hand, over the high latitudes (60°S to 70°), MODIS CFC was < 10% compared to CALIOP CFC. We observed that over Antarctica, CALIOP CFC was > 10% of MODIS CFC and > 20% of AVHRR CMSAF. When comparing the cloud fraction for liquid clouds with CALIOP, MODIS has an overall lower cloud cover, while AVHRR CMSAF has a lower cloud cover over the land and a higher cloud cover over mid-latitudes.

During the seasonal analysis, we observed that austral summer (JFA) has lower biases and better correlation than other seasons, while winter has higher biases and poorer correlation for CTHs comparison. It can be concluded that over the Antarctic coast CALIOP L3 cloud properties retrievals, have a positive bias with both MODIS and AVHRR CMSAF. However, over the mid-latitude, AVHRR CMSAF overestimates the cloud properties when compared to CALIOP.

An extensive analysis of L2 passive sensor (AVHRR CMSAF, MODIS, and AVHRR PATMOS) observations against active

sensor (CLOUDSAT-CALIOP merged) observations, was carried out in order to identify the factors contributing to the bias in cloud property retrievals. The presence of multilayer, sea ice concentration, and cloud mask were some of the factors analysed.

In the case of CTH analysis, a mean absolute bias of 0.65 km (AVHRR CMSAF), 1.03 km (MODIS), and 1.31 km (AVHRR PATMOS) was observed for single-layer cloud scenes cases. This mean bias increased to 1.86 km (AVHRR CMSAF), 3.22 km (MODIS), and 3.34 km (AVHRR PATMOS) for multilayered cloud scenes. Hence, we can conclude that the passive sensor

MBE against the active sensor for multilayer cases is 3 to 5 times the corresponding MBE for single-layer cases and 2 times for the overall year. The study demonstrates that multilayer clouds contribute significantly to the biases in CTH retrieval. The bias can also be attributed to the retrieval algorithm differences, COD of the layers, and sensitivity of sensors to the cloud.

The second major finding of this study ,the cloud mask comparison revealed that only around two-thirds of passive sensors observations agree with active sensor observations and in the case of AVHRR PATMOS the agreement is lower. More than half

of the observations for AVHRR PATMOS (KSS=0.43) show disagreement, likely due to its different spatial resolution from other sensors. In the case of sea ice-classified pixels, the KSS rates are lower, indicating the influence of sea ice on the cloud observation.

The cloud phase comparison would have been easier if there was a consistent phase definition between all the data sets. In the comparison of CER, it was observed that the disagreement between the passive sensors was greater in presence of multilayer

clouds. In single-layer cases, the liquid water phase dominates for MODIS, while the supercooled liquid phase for AVHRR CMSAF. The AVHRR PATMOS shows more ice phase for single-layer clouds than other passive sensors, with two peaks in the CER at 10 and 30 microns. The analysis of CER, COD, and CTH in relation to the sea ice mask revealed that the presence of sea ice plays an important role in uncertainties in COD.

The comparison and validation conclude that all passive sensors showed cloud property differences when compared with

CloudSat-CALIOP and when compared with each other, both negative and positive. The AVHRR CMSAF performs well throughout the CTH retrievals, but in multilayer scenarios, the correlation is lower. Similarly, MODIS is generally good at retrieving CTH, but in multilayer clouds, it has stronger biases when compared to AVHRR CMSAF. The AVHRR PATMOS sensors exhibited the lowest correlation and the highest bias in both the overall CTH and the multilayer and single-layer cases. The agreement between the cloud thermodynamics phase and the active sensor is reduced for AVHRR PATMOS. Although





MODIS has more channels and hence more information content compared to other retrievals, it is not better than AVHRR CMSAF with neural network retrievals. The AVHRR PATMOS has the poorest retrieval among the passive sensors.

The bias is affected by the multilayer structure of the clouds, the sensitivity of the retrieval algorithm, problems with temperature profile inversion, incorrect identification and classification of the thermodynamic phase, and the presence of supercooled liquid. The agreement for single-layer cloud retrieval is higher than the agreement for multilayer classified cloud retrieval.

This suggests that we need to invest more effort into improving multilayer retrieval issues, especially over the SO where the cloud properties are very different from those in other areas. Since reanalysis and climate models utilise these passive sensor retrievals.



*Data availability.* Publicly available data was analysed in this study. The data for the MODIS Level 3 product is available at the following DOI: http://dx.doi.org/10.5067/MODIS/CLDPROP_M3_MODIS_Aqua.011. The MODIS Level 2 product data is available at the following DOI:http://dx.doi.org/10.5067/MODIS/MYD06_L2.061. The AVHRR PATMOS Level 2 data is available at the DOI: https://doi.org/10.7289/V5X9287S. Data for AVHRR CMSAF Level 2 and Level 3 are available at the DOI: http://dx.doi.org/10.5676/EUM_SAF_CM/CLARA_AVHRR/V003. For the CloudSat-CALIOP merged dataset, the data is available at http://www.cloudsat.cira.colostate.edu/. The CALIOP Level 3 data is available at https://doi.org/10.5067/CALIOP/CALIPSO/LID_L3_GEWEX_Cloud-Standard-V1-00

*Author contributions.* AAK led the study, wrote code, analysed and interpretation of results. CP contributed to the methodology, code development and interpretation of results, and SS advised on science and interpretation of results. DR contributed to code development. All authors contributed to the science aspects. AAK prepared the manuscript with contributions from all authors.

*Competing interests.* The authors declare that they have no conflict of interest.

*Acknowledgements.* Mathilde Ritman for the CloudSat -CALIOP multilayer mask. This research project was undertaken with the assistance of resources and services from the National Computational Infrastructure (NCI), Australia.



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





**Figure 1** The flowchart of a) Collocation for MODIS/AVHRR Level 2 data and CloudSat-CALIOP b) Collocation for all sensors.

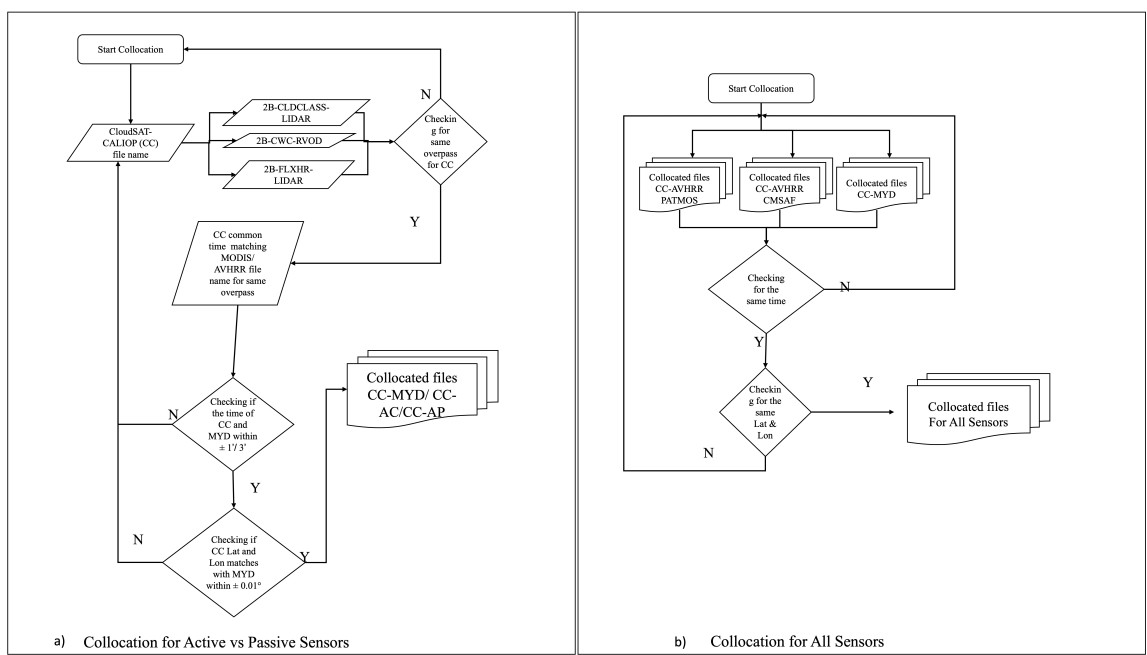

**Figure 2** The distribution of collocated Pixels for all 4 sensors. The collocation of MODIS and CloudSat-CALIOP (MODIS_CC) is shown in red, AVHRR CMSAF and CloudSat-CALIOP (AVHRRCMSAF_CC) is shown in blue and AVHRR PATMOS and CloudSat-CALIOP (AVHRRPATMOS_CC) is shown in green.

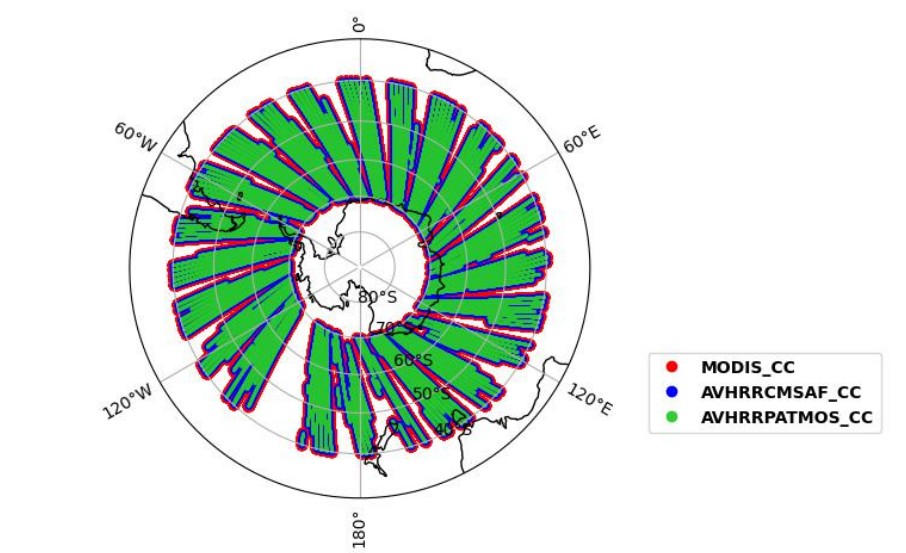





**Figure 3** 7-year (2009-2016) seasonal data comparison of Cloud Top Heights (CTH), over the Southern Ocean, for Level 3 data retrieved from AVHRR CMSAF (CLARA A3), MODIS AQUA (CLDPROP_D3_MODIS_Aqua) and CALIOP (CAL_LID_L3_GEWEX_Cloud). The left vertical panel shows the difference in CTHs for CALIOP and MODIS, the middle panel CALIOP and AVHRR CMSAF and the rightmost panel shows the SST from ERA-5 data for the same region.







**Figure 4** 7-year (2009-2016) seasonal data comparison of Cloud Fraction (CFC), over the Southern Ocean, for Level 3 data retrieved from AVHRR CMSAF (CLARA A3), MODIS AQUA (CLDPROP_D3_MODIS_Aqua) and CALIOP (CAL_LID_L3_GEWEX_Cloud). The left vertical panel shows the difference in CFCs for CALIOP and MODIS, the middle panel CALIOP and AVHRR CMSAF and the rightmost panel CALIOP CFC for the same region.





**Figure 5** 7-year (2009-2016) seasonal data comparison of Liquid Phase Clouds Fraction (CFL), over the Southern Ocean, for Level 3 data retrieved from AVHRR CMSAF (CLARA A3), MODIS AQUA (CLDPROP_D3_MODIS_Aqua) and CALIOP (CAL_LID_L3_GEWEX_Cloud.) The left vertical panel shows the difference in CFLs for CALIOP and MODIS, the middle panel CALIOP and AVHRR CMSAF and the rightmost panel shows CALIOP CFL for the same region.





**Figure 6** Swath for brightness temperature retrieved from MODIS (MYD) and merged CloudSat-CALIOP (CC) for Case study on 10-01-2015 at 22:50 UTC.

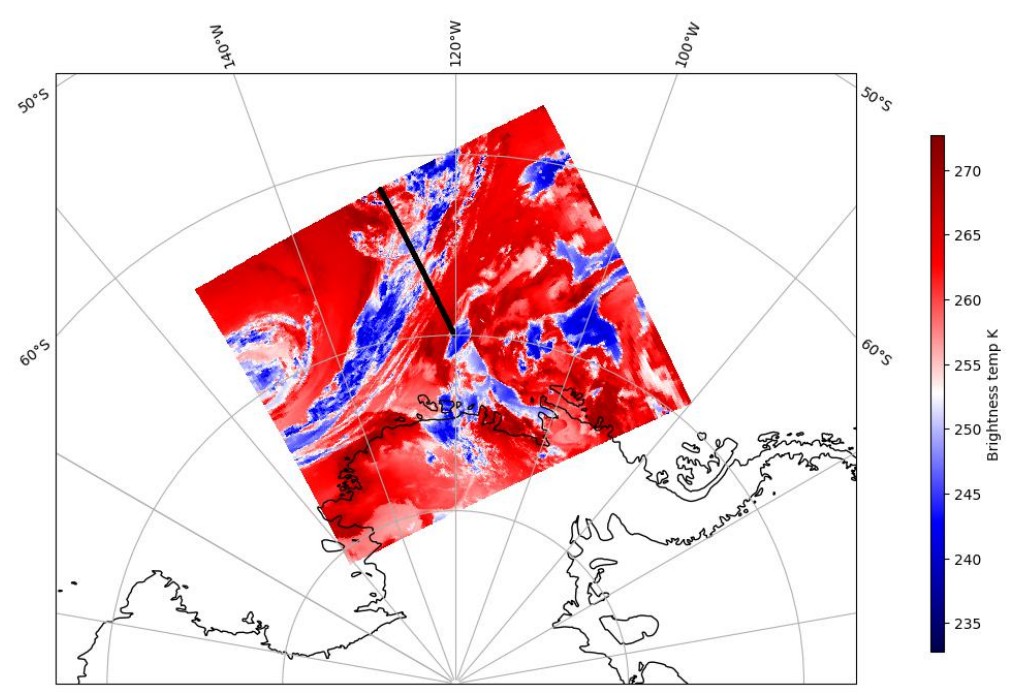



**Figure 7** Vertical profiles for cloud properties Cloud Top Height (CTH; Top panel), Cloud Optical Depth (COD; Second Panel), Cloud Effective Radius (CER) ; third panel) and Cloud Phase (CPH; Bottom Panel) retrieved from collocated data of AVHRR CMSAF (AC), AVHRR PATMOS (AP), MODIS (MYD) and merged CloudSat-CALIOP (CC) for Case study on 10-01-2015 at 22:50 UTC.





**Figure 8** Collocated Level 2 MODIS (top), AVHRR CMSAF (middle), AVHRR PATMOS (bottom) and CLOUDSAT-CALIOP merged data analysed for Cloud Top Height (CTH), over the Southern Ocean, for 2015 (left) shows the Joint 2D Histogram for CTHs, the 2D histograms for multilayer (right) and single-layer (centre) masked collocated data.

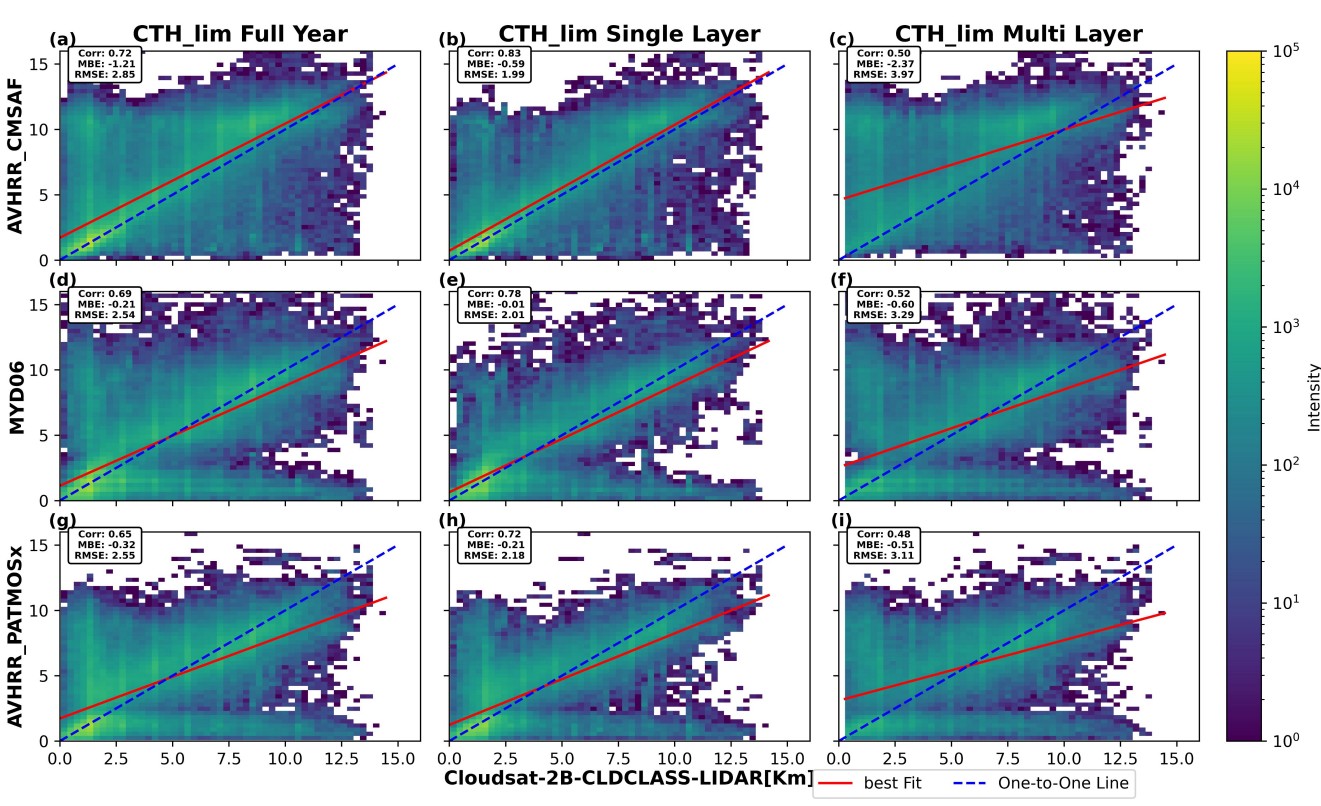



**Figure 9** Cloud Effective Radius (CER) distribution according to the cloud phase for the collocated data for the year 2015, over the Southern Ocean, for all passive sensors.

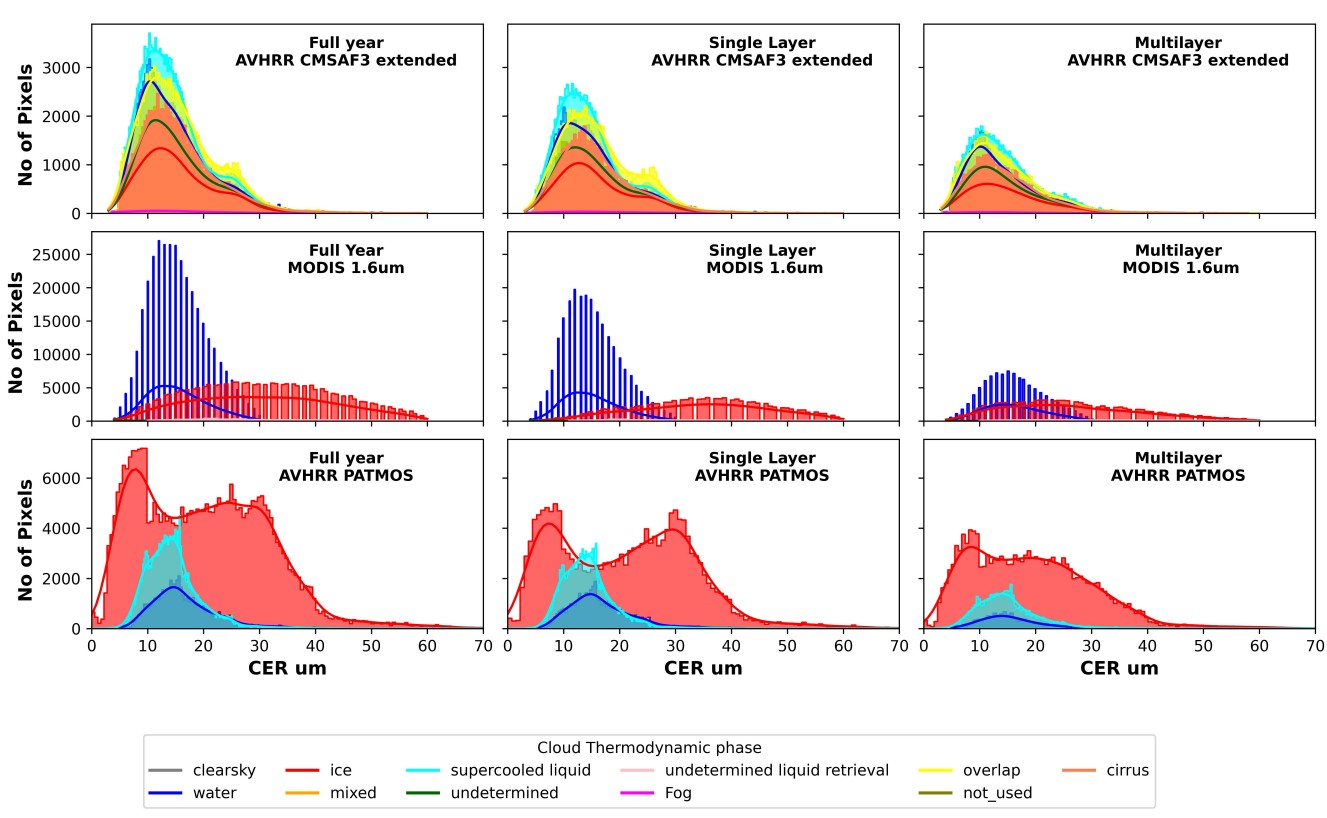



**Figure 10** Cloud Optical Depth (COD), Cloud Effective Radius (CER) and Cloud Top Height (CTH) as a function of Sea ice Flag from AVHRR PATMOS for all passive sensors, over the Southern Ocean. MODIS (MYD) is shown in red, AVHRR CMSAF (AC) is shown in black and AVHRR PATMOS (AP) is shown in yellow.

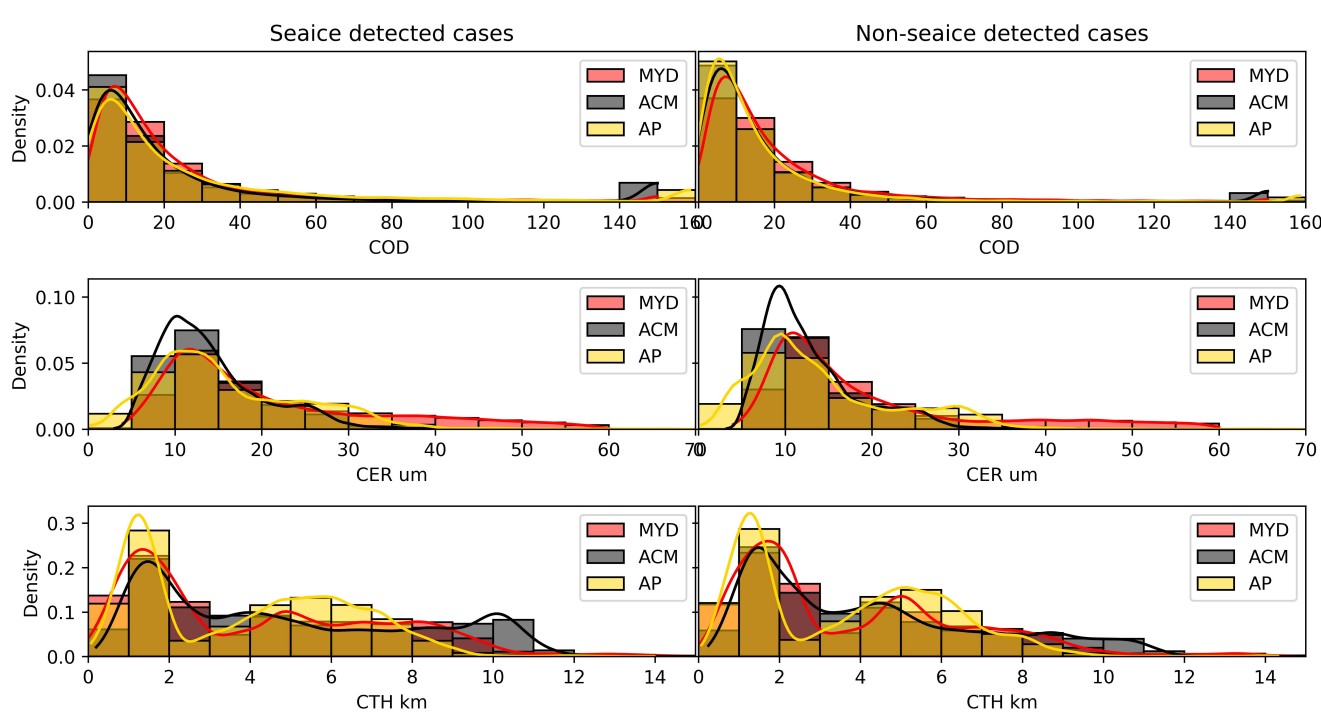





| | CloudSat CALIOP | CALIOP | MODIS | AVHRR CMSAF | AVHRR PATMOS |
|---|---|---|---|---|---|
| Level 3 product | - | CALIOP GEWEX | CLDPROP_M3_MODIS_Aqua | CLARA-A ed.3.0 | - |
| CTH | - | Cloud_Top_Altitude_Passive | Cloud_Top_Height | Cloud Top Height | - |
| CFC | - | Cloud_Amount_Mean_Passive | Cloud_Fraction | Cloud Fraction | - |
| CPHL | - | Water_Cloud_Amount_Mean_Passive | Cloud_Retrieval_Fraction_Liquid | Cloud Fraction for liquid clouds | - |
| Level 2 product | 2B-CLDCLASS-LIDAR (2BCL) 2B-FLXHR-LIDAR (2BFL) | - | MYD06_L2 | CLARA-A ed.3.0 | PATMOS-x versions 6.0 |
| CTH | CloudLayerTop (2BCL) | - | Cloud Top Height | Cloud Top Height | ACHA CTH |
| COD | COD (2BFL) | - | 1.6$\mu$m COD | 1.6$\mu$m COD | 1.6$\mu$m COD DCOMP |
| CER | - | - | 1.6$\mu$m CER | 1.6$\mu$m CER | 1.6$\mu$m CER |
| CPH | CloudPhase (2BCL) | - | Cloud top phase | Cloud top phase extended | Cloud top phase |
| CMK | Cloud Mask | - | Cloud Mask | Cloud Mask | Cloud Mask |

**Table 1.** Passive and active sensors Data and variables used for the Analysis.

| Mean Values of CTH | | | |
|---|---|---|---|
| Seasons | CALIOP (km) | AVHRR CMSAF (km) | MODIS (km) |
| 2009-2016 | 5.37 | 5.21 | 4.25 |
| Spring | 5.29 | 5.19 | 4.31 |
| Summer | 4.97 | 4.90 | 3.92 |
| Autumn | 5.39 | 5.17 | 4.14 |
| Winter | 5.8 | 5.55 | 4.60 |

(a)

| Sensor | CALIOP vs AVHRR CMSAF (km) | | | | CALIOP vs MODIS (km) | | | | AVHRR CMSAF vs MODIS (km) | | | |
|---|---|---|---|---|---|---|---|---|---|---|---|---|
| Season | RMSE | MAE | MBE | Correlation | RMSE | MAE | MBE | Correlation | RMSE | MAE | MBE | Correlation |
| 2009-2016 | 2.35 | 1.89 | 0.46 | 0.36 | 2.60 | 2.07 | 1.12 | 0.32 | 0.98 | 0.85 | 0.66 | 0.68 |
| Spring | 2.28 | 1.83 | 0.39 | 0.36 | 2.48 | 1.97 | 0.98 | 0.32 | 0.91 | 0.78 | 0.59 | 0.67 |
| Summer | 2.24 | 1.77 | 0.36 | 0.34 | 2.43 | 1.90 | 1.05 | 0.36 | 0.92 | 0.80 | 0.68 | 0.70 |
| Autumn | 2.43 | 1.95 | 0.52 | 0.31 | 2.70 | 2.14 | 1.25 | 0.28 | 1.05 | 0.91 | 0.73 | 0.57 |
| Winter | 2.66 | 2.16 | 0.56 | 0.14 | 2.77 | 2.23 | 1.20 | 0.28 | 1.27 | 1.05 | 0.64 | 0.42 |

(b)

**Table 2.** a) Mean values of Cloud Top Height (CTH) over the Southern Ocean and b) Statistics for 7 years (2009-2016) monthly data comparison of CTH for Level 3 data retrieved from AVHRR CMSAF (CLARA A3), MODIS AQUA (CLDPROP_D3_MODIS_Aqua) and CALIOP (CAL_LID_L3_GEWEX_Cloud.)



| Mean Values of CFC | | | |
|---|---|---|---|
| Seasons | CALIOP | AVHRR CMSAF | MODIS |
| 2009-2016 | 0.82 | 0.85 | 0.86 |
| Spring | 0.82 | 0.85 | 0.87 |
| Summer | 0.83 | 0.87 | 0.88 |
| Autumn | 0.83 | 0.86 | 0.87 |
| Winter | 0.81 | 0.83 | 0.82 |

(a)

| Sensor | CALIOP vs AVHRR CMSAF | | | | CALIOP vs MODIS | | | | AVHRR CMSAF vs MODIS | | | |
|---|---|---|---|---|---|---|---|---|---|---|---|---|
| Seasons | RMSE | MAE | MBE | Correlation | RMSE | MAE | MBE | Correlation | RMSE | MAE | MBE | Correlation |
| 2009-2016 | 0.17 | 0.12 | -0.03 | 0.52 | 0.18 | 0.13 | -0.04 | 0.48 | 0.07 | 0.05 | -0.01 | 0.87 |
| Spring | 0.17 | 0.12 | -0.03 | 0.56 | 0.17 | 0.12 | -0.04 | 0.54 | 0.07 | 0.05 | -0.01 | 0.85 |
| Summer | 0.17 | 0.12 | -0.04 | 0.55 | 0.17 | 0.12 | -0.05 | 0.58 | 0.06 | 0.04 | -0.02 | 0.89 |
| Autumn | 0.17 | 0.12 | -0.03 | 0.50 | 0.18 | 0.12 | -0.04 | 0.48 | 0.06 | 0.05 | -0.01 | 0.90 |
| Winter | 0.19 | 0.14 | -0.02 | 0.40 | 0.19 | 0.14 | -0.01 | 0.42 | 0.11 | 0.08 | 0.01 | 0.74 |

(b)

**Table 3.** a) Mean value of Cloud Fraction (CFC) over the Southern Ocean and b) Statistics for 7 years (2009-2016) monthly data comparison of CFC for Level 3 data retrieved from AVHRR CMSAF (CLARA A3), MODIS AQUA (CLDPROP_D3_MODIS_Aqua) and CALIOP (CAL_LID_L3_GEWEX_Cloud.)



| Mean Values of CFL | | | |
|---|---|---|---|
| Seasons | CALIOP | AVHRR CMSAF | MODIS |
| 2009-2016 | 0.49 | 0.53 | 0.45 |
| Spring | 0.47 | 0.49 | 0.43 |
| Summer | 0.54 | 0.63 | 0.54 |
| Autumn | 0.49 | 0.53 | 0.47 |
| Winter | 0.43 | 0.45 | 0.36 |

(a)

| Sensor | CALIOP vs AVHRR CMSAF | | | | CALIOP vs MODIS | | | | AVHRR CMSAF vs MODIS | | | |
|---|---|---|---|---|---|---|---|---|---|---|---|---|
| Season | RMSE | MAE | MBE | Correlation | RMSE | MAE | MBE | Correlation | RMSE | MAE | MBE | Correlation |
| 2009-2016 | 0.23 | 0.19 | -0.04 | 0.37 | 0.22 | 0.18 | 0.03 | 0.44 | 0.14 | 0.10 | 0.07 | 0.71 |
| Spring | 0.23 | 0.18 | -0.02 | 0.39 | 0.22 | 0.18 | 0.04 | 0.46 | 0.12 | 0.09 | 0.07 | 0.76 |
| Summer | 0.24 | 0.19 | -0.09 | 0.36 | 0.21 | 0.17 | 0.01 | 0.47 | 0.14 | 0.11 | 0.09 | 0.65 |
| Autumn | 0.24 | 0.19 | -0.03 | 0.30 | 0.23 | 0.18 | 0.02 | 0.38 | 0.13 | 0.10 | 0.05 | 0.60 |
| Winter | 0.24 | 0.19 | -0.02 | 0.21 | 0.24 | 0.19 | 0.07 | 0.32 | 0.17 | 0.13 | 0.09 | 0.41 |

(b)

**Table 4.** a) Mean Liquid Phase Clouds Fraction (CFL) values over the Southern Ocean and b) Statistics for 7 years (2009-2016) monthly data comparison of CFL for Level 3 data retrieved from AVHRR CMSAF (CLARA A3), MODIS AQUA (CLDPROP_D3_MODIS_Aqua) and CALIOP (CAL_LID_L3_GEWEX_Cloud.)

| Sensors | True Positive Rate (TPR)/Hit rate | False Positive Rate (FPR) | Kuiper Skill Score (KSS) |
|---|---|---|---|
| CloudSat-CALIOP vs AVHRR CMSAF | 0.94 | 0.22 | 0.71 |
| CloudSat-CALIOP vs MODIS | 0.77 | 0.07 | 0.70 |
| CloudSat-CALIOP vs AVHRR PATMOS | 0.95 | 0.51 | 0.43 |

**Table 5.** Cloud mask performance for 2015 over the Southern Ocean comparison for Level 2 AVHRR CMSAF, AVHRR PATMOS and MODIS against active sensor CloudSat-CALIOP (CC)

.





| Mean Values of CTHs | | | | |
|---|---|---|---|---|
| | CloudSat_CALIOP (km) > 0.5 COD | CloudSat_CALIOP (km) without COD limit | AVHRR CMSAF | MODIS | AVHRR PATMOS (km) |
| 2015 | 3.91 | 5.46 | 5.12 | 4.12 | 4.22 |
| Single-layer cases | 3.41 | 3.84 | 4.00 | 3.42 | 3.63 |
| Multilayer Cases | 4.81 | 8.40 | 7.18 | 5.42 | 5.32 |

(a)

| Statistics | CC vs AVHRR CMSAF | | | | CC vs MODIS | | | | CC vs AVHRR PATMOS | | | |
|---|---|---|---|---|---|---|---|---|---|---|---|---|
| | RMSE (km) | MAE (km) | MBE (km) | Correlation | RMSE (km) | MAE (km) | MBE (km) | Correlation | RMSE (km) | MAE (km) | MBE (km) | Correlation |
| 2015 | 2.85 | 1.76 | -1.21 | 0.72 | 2.54 | 1.56 | -0.21 | 0.69 | 2.55 | 1.67 | -0.32 | 0.65 |
| SL Cases | 1.99 | 1.12 | -0.59 | 0.83 | 2.01 | 1.18 | -0.01 | 0.78 | 2.18 | 1.37 | -0.21 | 0.72 |
| ML cases | 3.97 | 2.94 | -2.37 | 0.50 | 3.29 | 2.23 | -0.60 | 0.52 | 3.11 | 2.21 | -0.51 | 0.48 |

(b)

| Statistics | CC vs AVHRR CMSAF | | | | CC vs MODIS | | | | CC vs AVHRR PATMOS | | | |
|---|---|---|---|---|---|---|---|---|---|---|---|---|
| | RMSE (km) | MAE (km) | MBE (km) | Correlation | RMSE (km) | MAE (km) | MBE (km) | Correlation | RMSE (km) | MAE (km) | MBE (km) | Correlation |
| 2015 | 2.00 | 1.08 | 0.33 | 0.86 | 2.90 | 1.80 | 1.33 | 0.75 | 3.05 | 2.03 | 1.23 | 0.69 |
| SL cases | 1.26 | 0.65 | -0.16 | 0.93 | 1.68 | 1.03 | 0.42 | 0.88 | 2.03 | 1.31 | 0.21 | 0.80 |
| ML Cases | 2.89 | 1.86 | 1.22 | 0.64 | 4.31 | 3.22 | 2.99 | 0.53 | 4.34 | 3.34 | 3.08 | 0.44 |

(c)

**Table 6.** a) Mean Cloud top height (CTH) values for 2015 over the SO for Level 2 CTHs data MODIS (MYD), CloudSat-CALIOP (CC), AVHRR CMSAF (AC) and AVHRR PATMOS (AP). Statistics of the comparison of CC CTH with the COD limit > 0.5 is shown in b) and c) is for CC CTH without the COD limit.

| Cloud Thermodynamic Phase Classifications | | | |
|---|---|---|---|
| CLOUDSAT CALIPSO | MODIS | AVHRR CMSAF exteneded | AVHRR PATMOS |
| Undetrmined | Clear sky | Clear sky | Clear sky |
| Water | Water | Not used | Water |
| Ice | Ice | Fog | Supercooled liquid |
| Mixed | Undetermined | Water | Mixed |
| | Undetermined liquid retrieval | Supercooled liquid | Ice |
| | | Ice | Undetermined |
| | | Overlap | |
| | | Cirrus | |

**Table 7.** Cloud Thermodynamic Phase classification for Level 2 AVHRR CMSAF (AC), AVHRR PATMOS (AP), MODIS (MYD) and CloudSat-CALIOP (CC).





| Cloud Effective Radius | | | | | | | | |
|---|---|---|---|---|---|---|---|---|
| **Phase** | **Sensors** | Full year | | single-layer | | Multi Layer | |
| | | Mean | Median | Mean | Median | Mean | Median |
| **Water** | AC | 14.75 | 13.37 | 14.01 | 13.45 | 14.19 | 12.66 |
| | MYD | 15.59 | 15.00 | 15.21 | 15.00 | 16.38 | 16.00 |
| | AP | 16.72 | 15.51 | 16.01 | 15.68 | 16.26 | 14.79 |
| **Ice** | AC | 15.54 | 14.19 | 14.82 | 14.80 | 14.50 | 13.06 |
| | MYD | 30.90 | 30.00 | 30.68 | 35.00 | 26.54 | 25.00 |
| | AP | 20.32 | 19.57 | 19.51 | 21.53 | 19.23 | 18.08 |
| **Supercooled Liquid** | AC | 14.68 | 13.34 | 13.98 | 13.60 | 14.24 | 12.86 |
| | MYD | nan | nan | nan | nan | nan | nan |
| | AP | 14.62 | 13.96 | 14.46 | 13.87 | 15.11 | 14.28 |

**Table 8.** Mean and Median Cloud Effective Radius (CER) comparison for 2015 over the Southern Ocean, for water, ice and supercooled liquid phases from Level 2 AVHRR CMSAF (AC), AVHRR PATMOS (AP), MODIS (MYD) and CloudSat-CALIOP (CC).

| | COD | | | | CER | | | | CTH | | | |
|---|---|---|---|---|---|---|---|---|---|---|---|---|
| | Sea ice | | Non Seaice | | Sea ice | | Non Seaice | | Sea ice | | Non Seaice | |
| **Sensors** | Mean | Median | Mean | median | Mean | Median | Mean | median | Mean | Median | Mean | median |
| MYD | 23.30 | 13.0 | 20.46 | 13.0 | 20.82 | 16.0 | 19.45 | 15.00 | 4.15 | 3.3 | 3.87 | 2.75 |
| ACM | 25.98 | 11.46 | 19.19 | 10.35 | 14.25 | 12.66 | 13.01 | 11.32 | 4.91 | 4.17 | 4.27 | 3.56 |
| AP | 28.30 | 13.31 | 17.37 | 9.952 | 16.38 | 13.89 | 14.55 | 11.94 | 3.85 | 3.95 | 3.67 | 3.79 |

**Table 9.** Mean and Median values of Cloud Optical Depth (COD), Cloud Effective Radius (CER) and Cloud Top Height (CTH) over sea and over sea ice for Level 2 AVHRR CMSAF, AVHRR PATMOS and MODIS pixels at latitudes ($> 60°$S).