# Peer review of "Validation and comparison of cloud properties retrieved from passive satellites over the Southern Ocean"

_EGUsphere, 2025_

## Author Comment (AC1)

**General Replies to Comments**

Thank you for the valuable reviews and comments on the paper. We have amended the manuscript with regard to the issues raised following an extensive quality check and thorough re-evaluation. The abstract has also been updated accordingly. A brief summary of the key changes is given below.

All reviewers were concerned that the AVHRR data had not been properly collocated. The collocation method was thoroughly checked and found to be accurate; please refer to the appendix for further details.

However, Reviewer 1 identified an issue when a PATMOS swath ran across a calendar day: the initial day's record was erroneously being written onto the following day. This only occurs for a relatively small fraction of the total number of swaths (~5%), but these swaths have now been excluded from the analysis. The omission of these data did not qualitatively change the results or conclusions.

Figures 6, 7, 8, and 9 have been updated to reflect the changes made by excluding AVHRR PATMOS swaths after 22 UTC. Similarly, Tables 5, 6, and 8 have also been revised.

Unfortunately, the original case study chosen was one of these erroneous AVHRR swaths. As such, we have replaced the original case study from 10th January 2015 with a new one, 19th October 2015. This change in case study had no impact on the results or conclusions.

Please find the updated key changes, figures, and tables below.

In the general section, reviewers comments are included in *italics* with the original sentence in *'italics'* and changes in blue. The "Line #" is for the new modified manuscript.

**General key amendments to the manuscript.**

Listed below are the amendments to the manuscript according to the comments from Reviewer 1 (R1), Reviewer 2 (R2) and Community comments (CC1).

- Lines 64, 766 [R2]: The reference has been replaced with the suggested reference.

  *'Minnis et al., 2011'* with *'Minnis et al., 2020'*

  Minnis, P., Sun-Mack, S., Young, D., Heck, P., Garber, D., and Chen, Y.: CERES edition-2 cloud property retrievals using TRMM VIRS and Terra and Aqua MODIS data-Part I: Algorithms, IEEE Transactions on Geoscience and Remote Sensing, 49, 4374–4400, https://doi.org/10.1109/TGRS.2011.2144601, 2011.

  Minnis, P., Sun-Mack, S., Chen, Y., Chang, F.-L., Yost, C. R., Smith, W. L., Heck, P. W., Arduini, R. F., Bedka, S. T., Yi, Y., et al.: CERES MODIS cloud product retrievals for edition 4—Part I: Algorithm changes, IEEE Transactions on Geoscience and Remote Sensing, 59, 2744–2780, 2020.

- In Line 140 [R1, R2]: *'NASA'* has been replaced with *'NOAA'*

- Line 142 [R1]: R1 noted *'It says here NOAA-19 is used for the L2 comparisons, but in Table 1 it says the cloud optical properties being compared (COD and CER) are derived from the 1.6-micron channel. Channel 3ab on the AVHRR can either measure the 1.6 or 3.75, but not at once. NOAA-19 is switched to 3b, meaning it measures the 3.75, so it shouldn't be possible to have 1.6-micron properties for CMSAF and PATMOS-x.'*

Thank you for pointing this out. The 1.6 µm  CER retrievals have been replaced with 3.7 µm retrievals in all analyses.

Figure 9 and tables 9 and 10 have been updated along with all mention of the 1.6 µm channel. This has resulted in changes in MODIS results. These changes are highlighted and discussed later in this document.

*'The COD shown on each product uses the passive sensors' 1.6 µm channel.',*

 Changed to

'The COD was derived using the 3.7 µm channel, as it is common to all passive sensors' Line (390).

| | CloudSat CALIOP | CALIOP | MODIS | AVHRR CMSAF | AVHRR PATMOS |
|---|---|---|---|---|---|
| Level 3 product | - | CALIOP GEWEX | CLDPROP_M3_MODIS_Aqua | CLARA-A ed.3.0 | - |
| CTH | - | Cloud_Top_Altitude_Passive | Cloud_Top_Height | Cloud Top Height | - |
| CFC | - | Cloud_Amount_Mean_Passive | Cloud_Fraction | Cloud Fraction | - |
| CPHL | - | Water_Cloud_Amount_Mean_Passive | Cloud_Retrieval_Fraction_Liquid | Cloud Fraction for liquid clouds | - |
| Level 2 product | 2B-CLDCLASS-LIDAR (2BCL) 2B-FLXHR-LIDAR (2BFL) | - | MYD06_L2 | CLARA-A ed.3.0 | PATMOS-x versions 6.0 |
| CTH | CloudLayerTop (2BCL) | - | Cloud Top Height | Cloud Top Height | ACHA CTH |
| COD | COD (2BFL) | - | 1.6µm COD | 1.6µm COD | 1.6µm COD DCOMP |
| CER | - | - | 1.6µm CER | 1.6µm CER | 1.6µm CER |
| CPH | CloudPhase (2BCL) | - | Cloud top phase | Cloud top phase extended | Cloud top phase |
| CMK | Cloud Mask | - | Cloud Mask | Cloud Mask | Cloud Mask |

*Table 1. Passive and active sensors Data and variables used for the Analysis.*

was changed to

| | CloudSat CALIOP | CALIOP | MODIS | AVHRR CMSAF | AVHRR PATMOS |
|---|---|---|---|---|---|
| Level 3 product | - | CALIOP GEWEX | CLDPROP_M3_MODIS_Aqua | CLARA-A ed.3.0 | - |
| CTH | - | Cloud_Top_Altitude_Passive | Cloud_Top_Height | Cloud Top Height | - |
| CFC | - | Cloud_Amount_Mean_Passive | Cloud_Fraction | Cloud Fraction | - |
| CPHL | - | Water_Cloud_Amount_Mean_Passive | Cloud_Retrieval_Fraction_Liquid | Cloud Fraction for liquid clouds | - |
| Level 2 product | 2B-CLDCLASS-LIDAR (2BCL) 2B-FLXHR-LIDAR (2BFL) | - | MYD06_L2 | CLARA-A ed.3.0 | PATMOS-x versions 6.0 |
| CTH | CloudLayerTop (2BCL) | - | Cloud Top Height | Cloud Top Height | ACHA CTH |
| COD | COD (2BFL) | - | 3.7µm COD | 3.7µm COD | 3.7µm COD DCOMP |
| CER | - | - | 3.7µm CER | 3.7µm CER | 3.7µm CER |
| CPH | CloudPhase (2BCL) | - | Cloud top phase | Cloud top phase extended | Cloud top phase |
| CMK | Cloud Mask | - | Cloud Mask | Cloud Mask | Cloud Mask |

Table 1. Passive and active sensors Data and variables used for the Analysis.

- Line 154 [R1,CC1]: *PATMOS* has been replaced with PATMOS-x for clarity throughout the manuscript.
-
- Line 161 [R1,CC1]: R1 and CC1 pointed out *"For version 6.0 ACHA uses 11 and 13.3 channels, not 11 and 12."*

The sentence *'The ACHA CTH algorithm in this study uses 11, 12µm observations along with'* was changed to 'The ACHA CTH algorithm in this study uses the 11 and 13.3 µm observations along with'

- Line 237 [R1] : R1 pointed out the confusion over the term granule. *"I'm a little confused using the term 'granules' here. THE AVHRR GAC data is reported in ascending and descending orbits. Is that the meaning of granule here, or is it the global L2b data?"*

  The sentences, *'4000 CloudSat-CALIOP granules were collocated with 20,000 MYD06, 365 AVHRR CMSAF and 365 AVHRR PATMOS-x granules. Each granule contained the cloud products such as CTH, COD, CER, CPH, latitude, longitude and common time of collocations.'* was changed to

  ' 4000 CloudSat-CALIOP orbits were collocated with 20,000 MYD06 granules, 365 AVHRR CMSAF and 365 AVHRR PATMOS-x global ascending orbit files. Each collocated file contained the cloud products such as CTH, COD, CER, CPH, latitude, longitude and common time of collocations.' for clarity.

- Line 274 [R2] :R2 commented *" It isn't clear what bin the uncertain pixels were put. Please explain."*

  *'Similarly, MYD06 cloud mask, confidently clear (11), probably clear (10), uncertain (01) and cloudy (00) were classified into cloudy (0) and clear (1)'* has been modified to

  'Similarly, MYD06 cloud masks were classified as cloudy (0), both uncertain (01) and cloudy (00) and as clear (1), both confidently clear (11) and probably clear (10). '

- Line 298-306 [R2]: R2 commented on *"Check to see that the values in the text match those in the table. I think there are some discrepancies"*. Mismatched values from text reviewed and corrected accordingly and is highlighted below.

  *'Table 2 summarises the comparisons, with mean CTH values in Table 2a and the statistics for the comparison with CALIOP in Table 2b by season and overall. It can be observed that the active sensor correlation with the passive sensors for the overall monthly CTHs is low with values ranging from 0.35 for CALIOP-AVHRR CMSAF (C-A) and 0.32 for CALIOP-MODIS (CM). The CTH difference between the C-A sensors for DJF ranges from -1 to 1 km for the study domain, while for C-M it ranges from 0 to 4 km. The MBE for passive with the active sensor is 0.46 km for C-A and 1.12 km for C-M. The RMSE for C-A is 2.33 km; for C-M, the value is 2.60 km; and for A-M, it is 1.23 km, which is consistent with both correlation and MBE trend. Overall, the MBE between the MODIS-AVHRR CMSAF (A-M) passive sensors is larger for the CLARA-A3 dataset (0.95 km), but the correlation is better (0.67). Overall, there is a positive mean bias error for the passive sensors AVHRR CMSAF and MODIS CTHs.'*

  has been modified to

  'Table 2 summarises the comparisons, with mean CTH values in Table 2a and the statistics for the comparison with CALIOP in Table 2b by season and overall. It can be

observed that the active sensor correlation with the passive sensors for the overall monthly CTHs is low with values ranging from 0.36 for CALIOP-AVHRR CMSAF (C-A) and 0.32 for CALIOP-MODIS (CM). The CTH difference between the C-A sensors for DJF ranges from -1 to 1 km for the study domain, while for C-M it ranges from 0 to 4 km. The MBE for passive with the active sensor is 0.46 km for C-A and 1.12 km for C-M. The RMSE for C-A is 2.35 km; for C-M, the value is 2.60 km; and for A-M, it is 0.98 km, which is consistent with both correlation and MBE trend. Overall, the MBE between the MODIS-AVHRR CMSAF (A-M) passive sensors is larger for the CLARA-A3 dataset (0.98 km), but the correlation is better (0.68). Overall, there is a positive mean bias error for the passive sensors AVHRR CMSAF and MODIS CTHs.'

- Line 355 [R2]: removed *"this"* from the sentence

  *'The overall cloud fraction of liquid water clouds (hereafter CFL) and seasonal distributions are illustrated in this Figure 5.'*

  has been modified to

  'The overall cloud fraction of liquid water clouds (hereafter CFL) and seasonal distributions are illustrated in Figure 5.'

- Line 360 [R2]: Redundant sentence *'In the A-M seven-year monthly comparison, the overall correlation stands at a higher level (0.71) compared to both passive-active comparisons.'* has been removed.
- Line 390 [R2]: changed *'uses'* to *'was derived using'.*
- Line 409 [R1]: The Spelling typo *'retreive'* was corrected *to retrieve.*
- Line 438 [R1]: R1 commented, *"CMSAF and PATMOS-x both use AVHRR GAC data so they shouldn't have different spatial resolutions for L2 products. The sampling may be different, but that's not the same thing."*
  Hence, the sentence, *'The higher resolution of MODIS and AVHRR CMSAF may contribute to their higher KSS compared to AVHRR PATMOS.'* was amended to "The higher resolution of MODIS may contribute to the higher KSS compared to AVHRR PATMOS-x."
- Line 467-469 [R2]: R2 commented, *"CTH 'overestimates' are due to inversions, where the BT matches the profile temperature at a 'lower' height. Do you mean higher height?"*

  *'An explanation for the overestimation of low CTH is the effect of atmospheric temperature inversion, where the cloud brightness temperature taken by the passive sensor from the NWP reanalyses profile might be for a lower height than the actual CTH (Figure 7a[i]).'*

  Changed to

  An explanation for the overestimation of low CTH is the presence of a sharp temperature inversion that provides two solutions for the same IR temperature from the NWP reanalyse profile, one above and one below the actual CTH (Figure 7a[iii], Dong et al., 2008). A new reference was added to the line.

  And references added to References section in Line 638

Dong, X., Minnis, P., Xi, B., Sun-Mack, S., and Chen, Y.: Comparison of CERES-MODIS stratus cloud properties with ground-based measurements at the DOE ARM Southern Great Plains site, Journal of Geophysical Research: Atmospheres, 113, 2008.

- Line 493- 529[R2]: R2 commented *"Section lacks discussion regarding difference across algorithms with respect to CER magnitudes (i.e. why is MODIS ice Re so much higher than the others), and CER distributions (i.e. why does AC ice look like AC water, unlike AP and MODIS)"*

The discussion has been expanded and changed as part of the change in the MODIS retrieval channel (3.7 µm to 1.6 µm) as below. When the MODIS retrieval was changed to 3.7 µm, the values of CER became similar to other passive sensors, and the spread of the CER decreased: Line 530-560.

4.2.4 L2 Cloud Effective Radius Comparison

This section compares CER from the passive sensors as a function of CPH and the presence of single or multilayer clouds. The passive sensors categorise thermodynamic phase into various classifications according to their retrieval algorithms; for example, MYD has five phase categories, while AC has eight, including a supercooled water class. The categories used by each dataset are presented in Table 7. In the results presented, the number of classes was reduced: classifications of water, fog, and supercooled liquid are grouped as liquid; ice, cirrus, and overlap are treated as ice. When this generalised CPH classification is used, more liquid-phase clouds are found in SL cases, while ice dominates in ML cases. A comparison between the different sensors AC, MYD, and AP is illustrated in Figure 9 for the year 2015, stratified for SL and ML cases, and summarised in Table 8.

The liquid CER probability density functions (PDFs) are in broad agreement across the three sensors for all cases. The mean values for liquid clouds are approximately 14 µm for AC, 13 µm for MYD, and 17 µm for AP. The higher CER values from AP suggest potential misclassification of liquid clouds as ice clouds as the proportion of ice clouds in AP is over five times higher than that of liquid clouds. The liquid phase occurs more frequently in SL clouds than ML clouds across all sensors, with similar mean values for both cases (Table 8).

In contrast, the CER comparisons for ice-phase clouds reveal notable differences across the sensors. The overall mean for AC is 15.57 µm, while MYD and AP report means of 20.90 µm and 20.40 µm, respectively. AP also exhibits a distinct bimodal CER distribution with peaks at approximately 10 µm and 30 µm, particularly evident in SL ice-phase retrievals. ML cases show more similar CER distributions across the sensors. The SL–ML CER difference is relatively small for AC ($\approx 1$µm), modest for AP ($\approx 2$ µm), and larger for MYD ($\approx 5$ µm). Sensitivity to ice crystal habit selection and the corresponding LUT structure also influences retrievals. MYD uses scattering functions derived from the General Habit Mixture (GHM) developed by Baum et al. (2005a, b) and Yang et al. (2013), implemented as band-specific LUTs (Platnick et al., 2016; Amarasinghe et al., 16 2017). The ice habitat selection includes severely roughened aggregates, hollow columns, bullet rosettes, plates, and droxtals. AC uses radiative transfer simulations based on narrower droplet size distributions and an ice crystal model including severely roughened aggregated solid columns, hollow columns, and bullet rosettes (Yang et al., 2013; Baum et al., 2011). AP

uses LUTs from Baum/Yang scattering models with reduced spectral dimensionality , and its ice habit selection includes roughened bullet rosettes, aggregates, and hollow columns (Walther and Heidinger, 2012; Heidinger and Foster, 2021). These distinctions in habit selection and LUT configuration influence the consistency and comparability of retrieved cloud microphysical properties across sensors.

Supercooled liquid clouds are consistently retrieved by both AC and AP, particularly in SL regimes. AC reports a substantial supercooled component in full-year statistics, with a mean CER of 14.65 µm, due to its extended phase classification (Karlsson et al., 2023a). AP also distinguishes supercooled liquid from warm liquid, frequently identifying it in SL clouds, with a slightly higher mean CER of 17.20 µm. MYD lacks a dedicated supercooled category but includes an "undetermined liquid" retrieval class (ULQ; mean 19.71 µm), likely encompassing supercooled or glaciating clouds not clearly resolved by its phase algorithm (Platnick et al., 2016). Both AC and AP indicate persistent supercooled liquid presence over the Southern Ocean, consistent with prior observational studies (Huang et al., 2015; Bodas-Salcedo et al., 2016).

The differences observed between ML and SL cases suggest that multilayer clouds influence retrieval outcomes. Yost et al. (2023) had similar findings and demonstrated that multilayer conditions, especially non-opaque clouds over liquid, are among the leading sources of phase retrieval uncertainty in passive satellite observations.

Figure 9 replaced from

*Figure 9 Cloud Effective Radius (CER) distribution according to the cloud phase for the collocated data for the year 2015, over the Southern Ocean, for all passive sensors.*

[Figure]

To

Figure 9 Cloud Effective Radius (CER) distribution according to the cloud phase for the collocated data for the year 2015, over the Southern Ocean, for all passive sensor

[Figure]

*Table 8*

*From*

| **Cloud Effective Radius** | | | | | | | |
|---|---|---|---|---|---|---|---|
| **Phase** | **Sensors** | Full year | | single-layer | | Multi Layer | |
| | | Mean | Median | Mean | Median | Mean | Median |
| **Water** | AC | 14.75 | 13.37 | 14.01 | 13.45 | 14.19 | 12.66 |
| | MYD | 15.59 | 15.00 | 15.21 | 15.00 | 16.38 | 16.00 |
| | AP | 16.72 | 15.51 | 16.01 | 15.68 | 16.26 | 14.79 |
| **Ice** | AC | 15.54 | 14.19 | 14.82 | 14.80 | 14.50 | 13.06 |
| | MYD | 30.90 | 30.00 | 30.68 | 35.00 | 26.54 | 25.00 |
| | AP | 20.32 | 19.57 | 19.51 | 21.53 | 19.23 | 18.08 |
| **Supercooled Liquid** | AC | 14.68 | 13.34 | 13.98 | 13.60 | 14.24 | 12.86 |
| | MYD | nan | nan | nan | nan | nan | nan |
| | AP | 14.62 | 13.96 | 14.46 | 13.87 | 15.11 | 14.28 |

*Table 8. Mean and Median Cloud Effective Radius (CER) comparison for 2015 over the Southern Ocean, for water, ice and supercooled liquid phases from Level 2 AVHRR CMSAF (AC), AVHRR PATMOS-x (AP), MODIS (MYD) and CloudSat-CALIOP (CC).*

*To*

| Cloud Effective Radius [$\mu$m] | | | | | | | | |
|---|---|---|---|---|---|---|---|---|
| **Phase** | **Sensors** | Full year | | single-layer | | Multi Layer | |
| | | Mean | Median | Mean | Median | Mean | Median |
| **Liquid** | AC | 14.75 | 13.36 | 14.81 | 13.45 | 14.16 | 12.59 |
| | MYD | 13.28 | 13.00 | 13.38 | 13.00 | 13.07 | 12.00 |
| | AP | 17.20 | 15.51 | 17.35 | 15.95 | 16.61 | 15.03 |
| **Ice** | AC | 15.57 | 14.24 | 16.20 | 14.84 | 14.53 | 13.15 |
| | MYD | 20.90 | 21.00 | 23.36 | 23.00 | 18.33 | 18.00 |
| | AP | 20.40 | 19.67 | 21.39 | 21.74 | 19.26 | 18.09 |
| **Supercooled Liquid** | AC | 14.65 | 13.31 | 14.91 | 13.59 | 14.18 | 12.78 |
| | MYD | nan | nan | nan | nan | nan | nan |
| | AP | 14.70 | 13.97 | 14.54 | 13.88 | 15.21 | 14.33 |

Table 8. Mean and Median Cloud Effective Radius (CER) comparison for 2015 over the Southern Ocean, for water, ice and supercooled liquid phases from Level 2 AVHRR CMSAF (AC), AVHRR PATMOS-x (AP), MODIS (MYD) and CloudSat-CALIOP (CC).

- Line 569-572 [R2]: Changed > to greater and < to lesser in text as per reviewer comments
  *'In comparison to CALIOP CFC, the AVHRR CMSAF CFC was > 20% and the MODIS CFC was > 10%. On the other hand, over the high latitudes (60◦S to 70◦ S), MODIS CFC was < 10% compared to CALIOP CFC. We observed that over Antarctica, CALIOP CFC was > 10% of MODIS CFC and > 20% of AVHRR CMSAF.'* to ' In comparison to CALIOP CFC, the AVHRR CMSAF CFC was 20% greater and the MODIS CFC was 10% greater. On the other hand, over the high latitudes (60◦S to 70◦), MODIS CFC was 10% less compared to CALIOP CFC. We observed that over Antarctica, CALIOP CFC was 10% greater than MODIS CFC and 20% greater than AVHRR CMSAF.'

- Line 551-580 [R1]:R1 commented *'It says here NOAA-19 is used for the L2 comparisons, but in Table 1 it says the cloud optical properties being compared (COD and CER) are derived from the 1.6-micron channel. Channel 3ab on the AVHRR can either measure the 1.6 or 3.75, but not at once. NOAA-19 is switched to 3b, meaning it measures the 3.75, so it shouldn't be possible to have 1.6-micron properties for CMSAF and PATMOS-x.*

Thank you for pointing this out; the section was changed due to the 1.6$\mu$m CER retrievals being replaced with 3.7$\mu$m CER retrievals in the analyses as below. Figure 10 and table 9 were also replaced: Line 530-560.

4.2.5 L2 cloud property comparison as a function of surface type

In this section, we investigate the distribution of cloud properties as a function of sea ice coverage. The sea ice flag in AVHRR PATMOS-x(AP) retrieval was used to define the presence of sea ice, based on fractional sea ice coverage from NOAA/NCEP Climate Forecast System Reanalysis (CFSR) data (Foster et al., 2023). The distributions of the cloud properties CER, CTH, and COD were analysed as a function of sea ice coverage for latitudes > 60◦S, as shown in Figure 10. The narrow latitude range was selected to reduce the differences in microphysics caused by different meteorological regimes. The MYD cloud properties are shown in red, AC in black, and AP in yellow. Cloud properties such as

COD, CER, and CTH are compared for sea ice detected and non-sea ice detected; the mean and median are summarised in Table 9.

The mean COD for AC was 33.9 and 14.4; for MYD it was 24.1 and 18.6; and for AP it was 41.7 and 12.7 for sea ice and non-sea ice cases, respectively. The optical depth of the retrieved pixels increased in the presence of sea ice for all sensors. The results disagree with the findings of Frey et al. (2018) and Palm et al. (2010), which concluded that clouds were more frequent, and cloud optical depth tends to be higher over ice free regions. While the disagreement could arise from differences in latitude range and the limited temporal scope of our dataset compared to the multi-year analyses Frey et al. (2018) and Palm et al. (2010). It is hypothesised that the difference is due to two effects both originating in flaws of the retrieval algorithms: 1 - sea ice (with no associated cloud) is misidentified as cloud; this could be the cause of the spike in observations with greater than 100 COD over sea ice, and 2 - the albedo of sea ice used in the retrieval algorithms introducing a systematic positive COD 17 bias. It is difficult to define an accurate sea ice albedo; a systematic underestimation of the albedo could cause the optical depth of clouds to be overestimated.

In the comparison of CTH and CER between sea ice and non-sea ice cases, the differences for AP are small (0.1 µm for CER and -0.01 km for CTH). The MYD and AC products also show small positive differences in CER ($\approx$1 µm) over sea ice. For non sea ice cases, the CTHs are 0.05 km (MYD) and 0.6 km (AC) lower than sea ice cases. Although CER and CTH exhibit positive differences, these were much smaller than the differences observed for COD.

The seasonality of the COD, CER, and CTH anomalies over sea ice was also examined. We observed that the disagreement between retrievals over sea ice and non sea ice is higher in JJA, where sea ice cases show (not shown here) lower COD and CTH values. COD values remain below 50 for AC and MYD and below 80 for AP across all seasons, with generally lower values over sea ice in winter (JJA). CER distributions are relatively consistent across seasons, ranging from 0–35 µm for AP, 5–65 µm for MYD, and 5–45 µm for AC. Seasonal variation is more evident in CTH: low clouds (less than 2 km) dominate for MYD and AP in JJA and for all three sensors in SON, while DJF and MAM show a greater presence of mid-level clouds (3–6 km), though low clouds remain prominent for MYD and AP. It can be concluded that in the case of CTH and CER, the seasonality is not an important factor, while for COD, the seasonality and presence of sea ice are important factors in the retrieval of the product.

Figure 10 replaced from

*Figure 10 Cloud Optical Depth (COD), Cloud Effective Radius (CER) and Cloud Top Height (CTH) as a function of sea ice Flag from AVHRR PATMOS-x for all passive sensors, over the Southern Ocean less than 60OS. MODIS (MYD) is shown in red, AVHRR CMSAF (AC) is shown in black and AVHRR PATMOS-x (AP) is shown in yellow.*

[Figure]

To

Figure 10 Cloud Optical Depth (COD), Cloud Effective Radius (CER) and Cloud Top Height (CTH) as a function of Sea ice Flag from AVHRR PATMOS-x for all passive sensors, over the Southern Ocean less than 60 degrees S. MODIS (MYD) is shown in red, AVHRR CMSAF (AC) is shown in black and AVHRR PATMOS-x (AP) is shown in yellow.

[Figure]

*Table 9*

*From*

| Sensors | COD | | | | CER | | | | CTH | | | |
|---------|-----|-----|-----|-----|-----|-----|-----|-----|-----|-----|-----|-----|
| | Sea ice | | Non Seaice | | Sea ice | | Non Seaice | | Sea ice | | Non Seaice | |
| | Mean | Median | Mean | median | Mean | Median | Mean | median | Mean | Median | Mean | median |
| MYD | 23.30 | 13.0 | 20.46 | 13.0 | 20.82 | 16.0 | 19.45 | 15.00 | 4.15 | 3.3 | 3.87 | 2.75 |
| ACM | 25.98 | 11.46 | 19.19 | 10.35 | 14.25 | 12.66 | 13.01 | 11.32 | 4.91 | 4.17 | 4.27 | 3.56 |
| AP | 28.30 | 13.31 | 17.37 | 9.952 | 16.38 | 13.89 | 14.55 | 11.94 | 3.85 | 3.95 | 3.67 | 3.79 |

*Table 9.Mean and Median values of Cloud Optical Depth (COD), Cloud Effective Radius (CER) and Cloud Top Height (CTH) over sea and over sea ice for Level 2 AVHRR CMSAF, AVHRR PATMOS-x and MODIS pixels at latitudes (> 60◦ S).*

*To*

| | COD | | | | CER [$\mu$m] | | | | CTH [km] | | | |
|---|---|---|---|---|---|---|---|---|---|---|---|---|
| | Sea ice | | Non Seaice | | Sea ice | | Non Seaice | | Sea ice | | Non Seaice | |
| **Sensors** | Mean | Median | Mean | median | Mean | Median | Mean | median | Mean | Median | Mean | median |
| MYD | 24.08 | 13.00 | 18.55 | 12.00 | 15.92 | 13.00 | 14.46 | 12.00 | 4.02 | 3.30 | 3.93 | 2.80 |
| ACM | 33.89 | 11.46 | 14.44 | 9.17 | 14.02 | 12.25 | 13.01 | 11.91 | 5.07 | 4.52 | 4.40 | 3.60 |
| AP | 41.71 | 24.28 | 12.70 | 8.37 | 15.64 | 13.37 | 15.54 | 12.94 | 3.74 | 3.62 | 3.75 | 3.85 |

Table 9. Mean and Median values of Cloud Optical Depth (COD), Cloud Effective Radius (CER) and Cloud Top Height (CTH) over sea and over sea ice for Level 2 AVHRR CMSAF, AVHRR PATMOS-x and MODIS pixels at latitudes (> 60◦ S).

- Line 603 [R1]:R1 pointed out *''The AVHRR PATMOS sensors exhibited the lowest correlation and the highest bias in both the overall CTH and the multilayer and single-layer cases.' The numbers reported in Figure 8 don't seem to support this conclusion. The PATMOS-x RMSE and MBE is lower than MYD and CMSAF for the multi-layer cases and lower than CMSAF for all cases.'*

  The sentence was amended to exclude the incorrect conclusion. Hence *'The AVHRR PATMOS sensors exhibited the lowest correlation and the highest bias in both the overall CTH and the multilayer and single-layer cases.'* amended to Line 603 - 'The AVHRR PATMOS-x retrievals exhibited the lowest correlation and the highest bias in both the overall CTH and the multilayer and single-layer cases.'

- Line 610 [R2]: R2 commented incomplete sentence. Hence, removed *'Since reanalysis and climate models utilise these passive sensor retrievals.'*

- The case study portion was amended in response to Reviewer 1's notice concerning a flaw in the AVHRR PATMOS algorithm, which intermittently substitutes observations from the preceding day. The initial case study was impacted by this issue and subsequently withdrawn. Furthermore, all case studies with observation hours exceeding 22:00 UTC were omitted to prevent similar occurrences. This time threshold resulted in a reduction in only 5% of the collocations analysed in this study. Hence, Figures 6, 7, 8, 9 and 10 and Tables 5, 6, 8 and 9 have been amended with the changes. The changes did not significantly affect the overall results, aside from minor variations in decimal values.The following are the text changes made in the Manuscript.

  Line 254: Line was added in Methodology 'While analysing the case study data, it was found that AP data intermittently substitutes observations from the preceding day in case of missing orbits towards the end of the day. Hence, all case studies after 22:00 UTC were discarded from the whole year's analyses.'

  Line 379 : The entire section was changed as below.

  4.2.1 Case study

Figure 6 shows the 11 µm brightness temperature swaths for MODIS and CTH swath for MODIS (MYD), AVHRR PATMOS-x (AP) and AVHRR CMSAF (AC) and in black the merged CloudSat-CALIOP track for the 19-10-2015 at 08:37 UTC. Although the general pattern of CTH is similar between the three passive sensors, there are some significant differences. Although all three sensors see a similar high cloud pattern around 50◦S, AC retrieves the CTH around 11 km, while MYD and AP are around 9 km. AP has a significant region of cloud with CTHs ≈ 5 km at 40◦S and 80◦E, while for the same region, AC and MYD have low clouds (≈ 2 km).

Figure 7 shows a curtain plot for the CC track and the corresponding CTH, COD, CER, and CPH for each instrument. The top panel shows the CTH for all passive sensors (AC, AP, MYD, and CC), as well as the vertical profile of COD from the CC dataset (2BFL). The CTH for CC at a COD limit of 0.5 is also shown (cyan) to illustrate the potential differences in optical depth sensitivity between the passive and active sensors. The second panel illustrates the COD for the four sensors, along with the multilayer mask derived from CC. The COD was derived using the 3.7 µm channel as it is common to all passive sensors. The third panel displays the CER, as well as the sea ice mask used in the AP retrieval. The bottom panel shows the cloud phase for each instrument; for CC the phase is the phase detected at the top of the cloud. This example illustrates the highly variable nature of the cloud field across the Southern Ocean. Our discussion focuses on three sub-sections: multilayer clouds (Figure 7a[i]); thick upper-level clouds over thick low-level clouds (Figure 7a[ii]); and optically thick boundary layer clouds (Figure 7a[iii]).

In the first panel, the CTH ranges from 0 to 12 km and shows substantial differences between the CTH of passive and active sensors, with both underestimation and overestimation by passive sensors compared to CC. Figure 7a[i] shows a multilayer cloud structure with an optically thin high-level cloud over optically thick mid-level clouds. MODIS and AP detect the highlevel cloud, however since it is optically thin over an optically thick lower layer cloud, it places the CTH at an effective CTH between the upper and lower layers. For the AC retrieval, the CTH is similar to the CC height. In Figure 7a[ii], AC produces higher CTH values than CC, MYD yields lower values, and AP's results are closest to those of CC. This difference could be due to the different numerical weather prediction (NWP) reanalyses used for temperature profiles, which can vary substantially, particularly at high altitudes and high latitudes, and the difficulty in identifying the inversion at the tropopause (Wu et al., 2012; Müller et al., 2018).

The section Figure 7a[iii] around 41.5◦S shows a region of boundary layer clouds. AC does a good job of retrieving the boundary layer cloud, while AP boundary layer CTH can show large over estimations, and MYD occasionally over and underestimates. Boundary layer clouds can be difficult to estimate, particularly when temperature inversions are present, as they can present multiple solutions when using the thermal characteristics to retrieve CTH (Huang et al., 2012; Wall et al., 2017; Truong et al., 2020). AP misclassifies the phase between 41.5◦S to 40◦S which may explain the high CTH retrieval and highlights some internal dependency in the retrieval approach.

The second panel displays the CC multilayer mask and identifies the areas with multilayer clouds well. The COD for each retrieval is also compared in the second panel. The COD of CC (2BFL) exhibits high variability when compared to the COD of the passive sensors, particularly for ML clouds. CC COD is substantially higher than that of the other sensors. The COD has large differences among the passive sensors and active sensors in the presence of a multilayer structure (Figure 7a[i] and [ii]); In single-layer areas, the COD of MYD, AP and AC are more in agreement with the active sensor measurements. The results align with the product statement for CC COD (Henderson and L'Ecuyer, 2023) which states that the COD is similar to MYD for single layer and varies with multilayer clouds. The third panel (Figure 7c) shows the CER comparison and the sea ice flag. While the general pattern of the CER from the passive sensors is in agreement, the values, particularly where ice clouds are detected are quite different, AP CER are higher than MYD and AC CER retrievals in general.

The cloud phase is shown in the fourth panel (Figure 7d). For CPH, each retrieval algorithm uses a slightly different phase classification scheme. The CC retrievals classify the phase as mixed, liquid or ice phase on the basis of cloud top and base temperatures, radar reflectivity and integrated attenuated backscattering coefficient across all the cloud layers in the vertical column (Wang, 2019). For the passive sensors (MYD, AP and AC), the CPH is the phase detected at the top of the cloud. In the case study, all sensors share ice and liquid as common classes, with an additional clear sky class for MYD, a supercooled liquid class for AP, and a mixed phase class for CC. Further classification of the phase can be seen in 4.2.4. The agreement between the passive sensors overall is good with occasional differences; for example, at 43◦ S, AC and MYD classify the cloud phase as liquid while CC and AP classify it as ice, with cloud optical depth indicating a thin upper layer.

[revised manuscript text omitted]

- Figure 2 [R1]: R1 commented, *'It seems strange to me that the collocation coverage for CMSAF and PATMOS-x are different since they are derived using the same AVHRR measurements.'*

The Figure 2 caption was amended to reflect the collocation similarity as below

*'Figure 2 The distribution of collocated Pixels for all 4 sensors. The collocation of MODIS and CloudSat-CALIOP (MODIS_CC) is shown in red, AVHRR CMSAF and CloudSat-CALIOP (AVHRRCMSAF_CC) is shown in blue and AVHRR PATMOS-x and CloudSat-CALIOP (AVHRRPATMOS_CC) is shown in green.'*

To

'Figure 2 The distribution of collocated pixels for all 4 sensors. The collocation of MODIS and CloudSat-CALIOP (MODIS_CC) is shown in red, AVHRR CMSAF and CloudSat-CALIOP (AVHRRCMSAF_CC) is shown in blue and AVHRR PATMOS-x and CloudSat-CALIOP (AVHRRPATMOS_CC) is shown in green. The colours used in the plot to indicate the same data selection for each sensor are different but overlapping locations make them appear visually similar.'

**Reply to individual comments from Reviewer 1 (RC1)**

*The Southern Ocean is an important area of cloud research and this is a worthwhile inter-comparison. The background in the Introduction is also quite thorough and does a good job of framing this study in a larger context. There are some inconsistencies with the description of the datasets, particularly PATMOS-x, and the collocation process used in Figure 7. In order to have confidence in the results of this inter-comparison these need to be addressed, and at least some of the analysis will need to be redone. Specific comments below.*

We thank the reviewer for their valuable comments and corrections.

**Comment 1:** *Line 133: The POES satellites were operated by NOAA so I believe this would typically be NOAA Polar Operational Environmental Satellites (POES).*

The changes have been made as per requested.

- Line 133 *'NASA'* has been replaced with *'NOAA'* in Line 140.

**Comment 2:** *Line 135: It says here NOAA-19 is used for the L2 comparisons, but in Table 1 it says the cloud optical properties being compared (COD and CER) are derived from the 1.6-micron channel. Channel 3ab on the AVHRR can either measure the 1.6 or 3.75, but not at once. NOAA-19 is switched to 3b, meaning it measures the 3.75, so it shouldn't be possible to have 1.6-micron properties for CMSAF and PATMOS-x.*

The MODIS (COD and CER) from 1.6 to 3.7. The analysis and results are modified accordingly, and Figure 9 and Tables 1 and 8 were amended. Section 4.2.4 was also amended.

- As per comments, the 1.6 µm channel has been replaced with 3.7µm in all analyses. Subsequently, Figure 9 and Tables 9 and 10 have been updated along with all mention of 1.6µm channel.

  *'The COD shown on each product uses the passive sensors' 1.6 µm channel.',*

  Changed to

  'The COD was derived using the 3.7 µm channel, as it is common to all passive sensors' Line (390).

| | CloudSat CALIOP | CALIOP | MODIS | AVHRR CMSAF | AVHRR PATMOS |
|---|---|---|---|---|---|
| **Level 3 product** | - | CALIOP GEWEX | CLDPROP_M3_MODIS_Aqua | CLARA-A ed.3.0 | - |
| **CTH** | - | Cloud_Top_Altitude_Passive | Cloud_Top_Height | Cloud Top Height | - |
| **CFC** | - | Cloud_Amount_Mean_Passive | Cloud_Fraction | Cloud Fraction | - |
| **CPHL** | - | Water_Cloud_Amount_Mean_Passive | Cloud_Retrieval_Fraction_Liquid | Cloud Fraction for liquid clouds | - |
| **Level 2 product** | 2B-CLDCLASS-LIDAR (2BCL) 2B-FLXHR-LIDAR (2BFL) | - | MYD06_L2 | CLARA-A ed.3.0 | PATMOS-x versions 6.0 |
| **CTH** | CloudLayerTop (2BCL) | - | Cloud Top Height | Cloud Top Height | ACHA CTH |
| **COD** | COD (2BFL) | - | 1.6µm COD | 1.6µm COD | 1.6µm COD DCOMP |
| **CER** | - | - | 1.6µm CER | 1.6µm CER | 1.6µm CER |
| **CPH** | CloudPhase (2BCL) | - | Cloud top phase | Cloud top phase extended | Cloud top phase |
| **CMK** | Cloud Mask | - | Cloud Mask | Cloud Mask | Cloud Mask |

*Table 1. Passive and active sensors Data and variables used for the Analysis.*

Changed to

| | CloudSat CALIOP | CALIOP | MODIS | AVHRR CMSAF | AVHRR PATMOS |
|---|---|---|---|---|---|
| **Level 3 product** | - | CALIOP GEWEX | CLDPROP_M3_MODIS_Aqua | CLARA-A ed.3.0 | - |
| **CTH** | - | Cloud_Top_Altitude_Passive | Cloud_Top_Height | Cloud Top Height | - |
| **CFC** | - | Cloud_Amount_Mean_Passive | Cloud_Fraction | Cloud Fraction | - |
| **CPHL** | - | Water_Cloud_Amount_Mean_Passive | Cloud_Retrieval_Fraction_Liquid | Cloud Fraction for liquid clouds | - |
| **Level 2 product** | 2B-CLDCLASS-LIDAR (2BCL) 2B-FLXHR-LIDAR (2BFL) | | MYD06_L2 | CLARA-A ed.3.0 | PATMOS-x versions 6.0 |
| **CTH** | CloudLayerTop (2BCL) | - | Cloud Top Height | Cloud Top Height | ACHA CTH |
| **COD** | COD (2BFL) | - | 3.7μm COD | 3.7μm COD | 3.7μm COD DCOMP |
| **CER** | - | - | 3.7μm CER | 3.7μm CER | 3.7μm CER |
| **CPH** | CloudPhase (2BCL) | - | Cloud top phase | Cloud top phase extended | Cloud top phase |
| **CMK** | Cloud Mask | - | Cloud Mask | Cloud Mask | Cloud Mask |

Table 1. Passive and active sensors Data and variables used for the Analysis.

- Two sections also have been expanded and changed as part of the change in the MODIS retrieval channel change (3.7 µm to 1.6 µm) as below: Line 493-529 and Line 530-560.

4.2.4 L2 Cloud Effective Radius Comparison

This section compares CER from the passive sensors as a function of CPH and the presence of single or multilayer clouds. The passive sensors categorise thermodynamic phase into various classifications according to their retrieval algorithms; for example, MYD has five phase categories, while AC has eight, including a supercooled water class. The categories used by each dataset are presented in Table 7. In the results presented, the number of classes was reduced: classifications of water, fog, and supercooled liquid are grouped as liquid; ice, cirrus, and overlap are treated as ice. When this generalised CPH classification is used, more liquid-phase clouds are found in SL cases, while ice dominates in ML cases. A comparison between the different sensors AC, MYD, and AP is illustrated in Figure 9 for the year 2015, stratified for SL and ML cases, and summarised in Table 8.

The liquid CER probability density functions (PDFs) are in broad agreement across the three sensors for all cases. The mean values for liquid clouds are approximately 14 µm for AC, 13 µm for MYD, and 17 µm for AP. The higher CER values from AP suggest potential misclassification of liquid clouds as ice clouds as the proportion of ice clouds in AP is over five times higher than that of liquid clouds. The liquid phase occurs more frequently in SL clouds than ML clouds across all sensors, with similar mean values for both cases (Table 8).

In contrast, the CER comparisons for ice-phase clouds reveal notable differences across the sensors. The overall mean for AC is 15.57 µm, while MYD and AP report means of 20.90 µm and 20.40 µm, respectively. AP also exhibits a distinct bimodal CER distribution with peaks at approximately 10 µm and 30 µm, particularly evident in SL ice-phase retrievals. ML cases show more similar CER distributions across the sensors. The SL–ML CER difference is relatively small for AC (≈ 1µm), modest for AP (≈ 2 µm), and larger for MYD (≈ 5 µm). Sensitivity to ice crystal habit selection and the corresponding LUT structure also influences retrievals. MYD uses scattering functions derived from the General Habit Mixture (GHM) developed by Baum et al. (2005a, b) and Yang et al. (2013), implemented as band-specific LUTs (Platnick et al., 2016; Amarasinghe et al., 16 2017). The ice habitat selection includes severely roughened aggregates, hollow columns, bullet rosettes, plates,

and droxtals. AC uses radiative transfer simulations based on narrower droplet size distributions and an ice crystal model including severely roughened aggregated solid columns, hollow columns, and bullet rosettes (Yang et al., 2013; Baum et al., 2011). AP uses LUTs from Baum/Yang scattering models with reduced spectral dimensionality , and its ice habit selection includes roughened bullet rosettes, aggregates, and hollow columns (Walther and Heidinger, 2012; Heidinger and Foster, 2021). These distinctions in habit selection and LUT configuration influence the consistency and comparability of retrieved cloud microphysical properties across sensors.

Supercooled liquid clouds are consistently retrieved by both AC and AP, particularly in SL regimes. AC reports a substantial supercooled component in full-year statistics, with a mean CER of 14.65 µm, due to its extended phase classification (Karlsson et al., 2023a). AP also distinguishes supercooled liquid from warm liquid, frequently identifying it in SL clouds, with a slightly higher mean CER of 17.20 µm. MYD lacks a dedicated supercooled category but includes an "undetermined liquid" retrieval class (ULQ; mean 19.71 µm), likely encompassing supercooled or glaciating clouds not clearly resolved by its phase algorithm (Platnick et al., 2016). Both AC and AP indicate persistent supercooled liquid presence over the Southern Ocean, consistent with prior observational studies (Huang et al., 2015; Bodas-Salcedo et al., 2016).

The differences observed between ML and SL cases suggest that multilayer clouds influence retrieval outcomes. Yost et al. (2023) had similar findings and demonstrated that multilayer conditions, especially non-opaque clouds over liquid, are among the leading sources of phase retrieval uncertainty in passive satellite observations.

Figure 9 replaced from

*Figure 9 Cloud Effective Radius (CER) distribution according to the cloud phase for the collocated data for the year 2015, over the Southern Ocean, for all passive sensors.*

[Figure]

To

[Figure]

Figure 9 Cloud Effective Radius (CER) distribution according to the cloud phase for the collocated data for the year 2015, over the Southern Ocean, for all passive sensors.

*Table 8*

*From*

| Cloud Effective Radius | | | | | | | |
|---|---|---|---|---|---|---|---|
| **Phase** | **Sensors** | Full year | | single-layer | | Multi Layer | |
| | | Mean | Median | Mean | Median | Mean | Median |
| **Water** | AC | 14.75 | 13.37 | 14.01 | 13.45 | 14.19 | 12.66 |
| | MYD | 15.59 | 15.00 | 15.21 | 15.00 | 16.38 | 16.00 |
| | AP | 16.72 | 15.51 | 16.01 | 15.68 | 16.26 | 14.79 |
| **Ice** | AC | 15.54 | 14.19 | 14.82 | 14.80 | 14.50 | 13.06 |
| | MYD | 30.90 | 30.00 | 30.68 | 35.00 | 26.54 | 25.00 |
| | AP | 20.32 | 19.57 | 19.51 | 21.53 | 19.23 | 18.08 |
| **Supercooled Liquid** | AC | 14.68 | 13.34 | 13.98 | 13.60 | 14.24 | 12.86 |
| | MYD | nan | nan | nan | nan | nan | nan |
| | AP | 14.62 | 13.96 | 14.46 | 13.87 | 15.11 | 14.28 |

*Table 8. Mean and Median Cloud Effective Radius (CER) comparison for 2015 over the Southern Ocean, for water, ice and supercooled liquid phases from Level 2 AVHRR CMSAF (AC), AVHRR PATMOS-x (AP), MODIS (MYD) and CloudSat-CALIOP (CC).*

*To*

| Cloud Effective Radius [$\mu$m] | | | | | | | |
|---|---|---|---|---|---|---|---|
| **Phase** | **Sensors** | Full year | | single-layer | | Multi Layer | |
| | | Mean | Median | Mean | Median | Mean | Median |
| **Liquid** | AC | 14.75 | 13.36 | 14.81 | 13.45 | 14.16 | 12.59 |
| | MYD | 13.28 | 13.00 | 13.38 | 13.00 | 13.07 | 12.00 |
| | AP | 17.20 | 15.51 | 17.35 | 15.95 | 16.61 | 15.03 |
| **Ice** | AC | 15.57 | 14.24 | 16.20 | 14.84 | 14.53 | 13.15 |
| | MYD | 20.90 | 21.00 | 23.36 | 23.00 | 18.33 | 18.00 |
| | AP | 20.40 | 19.67 | 21.39 | 21.74 | 19.26 | 18.09 |
| **Supercooled Liquid** | AC | 14.65 | 13.31 | 14.91 | 13.59 | 14.18 | 12.78 |
| | MYD | nan | nan | nan | nan | nan | nan |
| | AP | 14.70 | 13.97 | 14.54 | 13.88 | 15.21 | 14.33 |

Table 8. Mean and Median Cloud Effective Radius (CER) comparison for 2015 over the Southern Ocean, for water, ice and supercooled liquid phases from Level 2 AVHRR CMSAF (AC), AVHRR PATMOS-x (AP), MODIS (MYD) and CloudSat-CALIOP (CC).

**4.2.5 L2 cloud property comparison as a function of surface type**

In this section, we investigate the distribution of cloud properties as a function of sea ice coverage. The sea ice flag in AVHRR PATMOS-x(AP) retrieval was used to define the presence of sea ice, based on fractional sea ice coverage from NOAA/NCEP Climate Forecast System Reanalysis (CFSR) data (Foster et al., 2023). The distributions of the cloud properties CER, CTH, and COD were analysed as a function of sea ice coverage for latitudes > 60∘S, as shown in Figure 10. The narrow latitude range was selected to reduce the differences in microphysics caused by different meteorological regimes. The MYD cloud properties are shown in red, AC in black, and AP in yellow. Cloud properties such as COD, CER, and CTH are compared for sea ice detected and non-sea ice detected; the mean and median are summarised in Table 9.

The mean COD for AC was 33.9 and 14.4; for MYD it was 24.1 and 18.6; and for AP it was 41.7 and 12.7 for sea ice and non-sea ice cases, respectively. The optical depth of the retrieved pixels increased in the presence of sea ice for all sensors. The results disagree with the findings of Frey et al. (2018) and Palm et al. (2010), which concluded that clouds were more frequent, and cloud optical depth tends to be higher over ice free regions. While the disagreement could arise from differences in latitude range and the limited temporal scope of our dataset compared to the multi-year analyses Frey et al. (2018) and Palm et al. (2010). It is hypothesised that the difference is due to two effects both originating in flaws of the retrieval algorithms: 1 - sea ice (with no associated cloud) is misidentified as cloud; this could be the cause of the spike in observations with greater than 100 COD over sea ice, and

2 - the albedo of sea ice used in the retrieval algorithms introducing a systematic positive COD 17 bias. It is difficult to define an accurate sea ice albedo; a systematic underestimation of the albedo could cause the optical depth of clouds to be overestimated.

In the comparison of CTH and CER between sea ice and non-sea ice cases, the differences for AP are small (0.1 µm for CER and -0.01 km for CTH). The MYD and AC products also show small positive differences in CER (≈1 µm) over sea ice. For non sea ice cases, the CTHs are 0.05 km (MYD) and 0.6 km (AC) lower than sea ice cases. Although CER and CTH exhibit positive differences, these were much smaller than the differences observed for COD.

The seasonality of the COD, CER, and CTH anomalies over sea ice was also examined. We observed that the disagreement between retrievals over sea ice and non sea ice is higher in JJA, where sea ice cases show (not shown here) lower COD and CTH values. COD values remain below 50 for AC and MYD and below 80 for AP across all seasons, with generally lower values over sea ice in winter (JJA). CER distributions are relatively consistent across seasons, ranging from 0–35 µm for AP, 5–65 µm for MYD, and 5–45 µm for AC. Seasonal variation is more evident in CTH: low clouds (less than 2 km) dominate for MYD and AP in JJA and for all three sensors in SON, while DJF and MAM show a greater presence of mid-level clouds (3–6 km), though low clouds remain prominent for MYD and AP. It can be concluded that in the case of CTH and CER, the seasonality is not an important factor, while for COD, the seasonality and presence of sea ice are important factors in the retrieval of the product.

Figure 10 replaced from

*Figure 10 Cloud Optical Depth (COD), Cloud Effective Radius (CER) and Cloud Top Height (CTH) as a function of sea ice Flag from AVHRR PATMOS-x for all passive sensors, over the Southern Ocean less than 60OS. MODIS (MYD) is shown in red, AVHRR CMSAF (AC) is shown in black and AVHRR PATMOS-x (AP) is shown in yellow.*

[Figure]

To

Figure 10 Cloud Optical Depth (COD), Cloud Effective Radius (CER) and Cloud Top Height (CTH) as a function of Sea ice Flag from AVHRR PATMOS-x for all passive sensors, over the Southern Ocean less than 60 degrees S. MODIS (MYD) is shown in red, AVHRR CMSAF (AC) is shown in black and AVHRR PATMOS-x (AP) is shown in yellow.

[Figure]

Table 9

From

| Sensors | COD | | | | CER | | | | CTH | | | |
|---|---|---|---|---|---|---|---|---|---|---|---|---|
| | Sea ice | | Non Seaice | | Sea ice | | Non Seaice | | Sea ice | | Non Seaice | |
| | Mean | Median | Mean | median | Mean | Median | Mean | median | Mean | Median | Mean | median |
| MYD | 23.30 | 13.0 | 20.46 | 13.0 | 20.82 | 16.0 | 19.45 | 15.00 | 4.15 | 3.3 | 3.87 | 2.75 |
| ACM | 25.98 | 11.46 | 19.19 | 10.35 | 14.25 | 12.66 | 13.01 | 11.32 | 4.91 | 4.17 | 4.27 | 3.56 |
| AP | 28.30 | 13.31 | 17.37 | 9.952 | 16.38 | 13.89 | 14.55 | 11.94 | 3.85 | 3.95 | 3.67 | 3.79 |

*Table 9. Mean and Median values of Cloud Optical Depth (COD), Cloud Effective Radius (CER) and Cloud Top Height (CTH) over sea and over sea ice for Level 2 AVHRR CMSAF, AVHRR PATMOS-x and MODIS pixels at latitudes (> 60∘ S).*

*To*

| Sensors | COD | | | | CER [$\mu m$] | | | | CTH [km] | | | |
| | Sea ice | | Non Seaice | | Sea ice | | Non Seaice | | Sea ice | | Non Seaice | |
| | Mean | Median | Mean | median | Mean | Median | Mean | median | Mean | Median | Mean | median |
|---|---|---|---|---|---|---|---|---|---|---|---|---|
| MYD | 24.08 | 13.00 | 18.55 | 12.00 | 15.92 | 13.00 | 14.46 | 12.00 | 4.02 | 3.30 | 3.93 | 2.80 |
| ACM | 33.89 | 11.46 | 14.44 | 9.17 | 14.02 | 12.25 | 13.01 | 11.91 | 5.07 | 4.52 | 4.40 | 3.60 |
| AP | 41.71 | 24.28 | 12.70 | 8.37 | 15.64 | 13.37 | 15.54 | 12.94 | 3.74 | 3.62 | 3.75 | 3.85 |

Table 9. Mean and Median values of Cloud Optical Depth (COD), Cloud Effective Radius (CER) and Cloud Top Height (CTH) over sea and over sea ice for Level 2 AVHRR CMSAF, AVHRR PATMOS-x and MODIS pixels at latitudes (> 60◦ S).

**Comment 3:** *Line 146: Here and elsewhere PATMOS should be PATMOS-x*

*'PATMOS'* was changed to 'PATMOS-x' throughout the manuscript to maintain clarity.

**Comment 4:** *Line 154: For version 6.0 ACHA uses 11 and 13.3 channels, not 11 and 12.*

The comment was accepted and changed accordingly as below

The sentence *'The ACHA CTH algorithm in this study uses 11, 12μm observations along with the'* was changed to Line 161 'The ACHA CTH algorithm in this study uses 11, 13.3μm observations along with'

**Comment 5:** *Line 231: I'm a little confused using the term 'granules' here. THE AVHRR GAC data is reported in ascending and descending orbits. Is that the meaning of granule here, or is it the global L2b data?*

The *'granules'* term used for AVHRR here is in the context of the number of files. I have only taken ascending data not descending orbits. In order to provide further clarity the *'granules'* are changed accordingly to orbit files and global ascending orbit files as shown below.

The sentences, *'4000 CloudSat-CALIOP granules were collocated with 20,000 MYD06, 365 AVHRR CMSAF and 365 AVHRR PATMOS-x granules. Each granule contained the cloud products such as CTH, COD, CER, CPH, latitude, longitude and common time of collocations.'* was changed to

' 4000 CloudSat-CALIOP orbit granules were collocated with 20,000 MYD06 granules, 365 AVHRR CMSAF and 365 AVHRR PATMOS-x global ascending orbit files. Each file contained the cloud products such as CTH, COD, CER, CPH, latitude, longitude and common time of collocations.', line 237.

**Comment 6:** *Line 410: retreive -> retrieve*

The spelling typo was changed.

The Spelling typo *'retreive'* was corrected *to retrieve.*

**Comment 7:** *Line 453: CMSAF and PATMOS-x both use AVHRR GAC data so they shouldn't have different spatial resolutions for L2 products. The sampling may be different, but that's not the same thing.*

We acknowledge the comment and have removed AVHRR CMSAF from the sentence. Hence, the sentence

*'The higher resolution of MODIS and AVHRR CMSAF may contribute to their higher KSS compared to AVHRR PATMOS.'*
was amended to
" The higher resolution of  MODIS may contribute to the higher KSS compared to AVHRR PATMOS-x.", Line 438.

**Comment 8:** *Line 603: 'The AVHRR PATMOS sensors exhibited the lowest correlation and the highest bias in both the overall CTH and the multilayer and single-layer cases.' The numbers reported in Figure 8 don't seem to support this conclusion. The PATMOS-x RMSE and MBE is lower than MYD and CMSAF for the multi-layer cases and lower than CMSAF for all cases.*

*'The AVHRR PATMOS sensors exhibited the lowest correlation and the highest bias in both the overall CTH and the multilayer and single-layer cases.'* Amended to 'The AVHRR PATMOS-x retrievals exhibited the lowest correlation in both the overall CTH and the multilayer and single-layer cases.'

**Comment 9:** *Figure 2. It seems strange to me that the collocation coverage for CMSAF and PATMOS-x are different since they are derived using the same AVHRR measurements.*

The collocation is not different; it's the same however the plots may have been confusing. I have amended the text in the figure to make this clearer. In order to show all satellites the color is highlighted. Please refer to Appendix Figure 1 for the 3D plot to show the collocation. To clarify this the figure label for Figure 2 has been modified as below

From

*'Figure 2 The distribution of collocated Pixels for all 4 sensors. The collocation of MODIS and CloudSat-CALIOP (MODIS_CC) is shown in red, AVHRR CMSAF and CloudSat-CALIOP (AVHRRCMSAF_CC) is shown in blue and AVHRR PATMOS-x and CloudSat-CALIOP (AVHRRPATMOS_CC) is shown in green.'*

To

'Figure 2 The distribution of collocated pixels for all 4 sensors. The collocation of MODIS and CloudSat-CALIOP (MODIS_CC) is shown in red, AVHRR CMSAF and CloudSat-CALIOP (AVHRRCMSAF_CC) is shown in blue and AVHRR PATMOS-x and CloudSat-CALIOP (AVHRRPATMOS_CC) is shown in green. The colours used in the plot to indicate the same data selection for each sensor are different but overlapping locations make them appear visually similar.'

**Comment 10:** *I suspect the odd PATMOS-x behavior in Figure 7 can be explained by a collocation problem due to an issue with how the PATMOS-x L2b global file is created. The L2b files are created by sampling orbits for a single day to a 0.1-degree global grid. Final orbits from the previous day are included because the orbit can cross from one day to another. So, if you are looking at the end of the day, and that orbit is missing, you can get a situation where the orbit used is from the previous day (this issue was addressed for recent file deliveries but still exists for most of the record). Below is a list of the orbits used in the NOAA-19 L2b ascending file from January 10th, 2015 (this is from the 'source' global attribute in the file). The pertinent files have been italicized. The start and end times in the last two files show a time gap, suggesting an orbit wasn't processed. This time gap coincides with the start and end times from the first file, which is from January 9th. This gap coincides with the reported collocation time of 22:50 UTC in Figure 7, suggesting the PATMOS-x orbit being analyzed was from a day previous than Cloudsat, MODIS, and CMSAF.*

*clavrx_NSS.GHRR.NP.D15009.S2214.E0001.B3051617.GC.hirs_avhrr_fusion.level2.nc*

clavrx_NSS.GHRR.NP.D15009.S2356.E0146.B3051718.GC.hirs_avhrr_fusion.level2.nc

clavrx_NSS.GHRR.NP.D15010.S0140.E0335.B3051819.SV.hirs_avhrr_fusion.level2.nc

clavrx_NSS.GHRR.NP.D15010.S0329.E0524.B3051920.WI.hirs_avhrr_fusion.level2.nc

clavrx_NSS.GHRR.NP.D15010.S0518.E0711.B3052021.WI.hirs_avhrr_fusion.level2.nc

clavrx_NSS.GHRR.NP.D15010.S0705.E0853.B3052122.WI.hirs_avhrr_fusion.level2.nc

clavrx_NSS.GHRR.NP.D15010.S0847.E1029.B3052223.GC.hirs_avhrr_fusion.level2.nc

clavrx_NSS.GHRR.NP.D15010.S1204.E1351.B3052425.GC.hirs_avhrr_fusion.level2.nc

clavrx_NSS.GHRR.NP.D15010.S1345.E1531.B3052526.GC.hirs_avhrr_fusion.level2.nc

clavrx_NSS.GHRR.NP.D15010.S1525.E1657.B3052627.WI.hirs_avhrr_fusion.level2.nc

clavrx_NSS.GHRR.NP.D15010.S1652.E1836.B3052728.WI.hirs_avhrr_fusion.level2.nc

clavrx_NSS.GHRR.NP.D15010.S1830.E2025.B3052729.GC.hirs_avhrr_fusion.level2.nc

*clavrx_NSS.GHRR.NP.D15010.S2019.E2208.B3052930.GC.hirs_avhrr_fusion.level2.nc*

*clavrx_NSS.GHRR.NP.D15010.S2344.E0134.B3053132.GC.hirs_avhrr_fusion.level2.nc*

*It's worth reiterating this is an issue with how the PATMOS-x L2b files are created and would be difficult for the authors to identify without looking at imagery to ensure the products are observing the same scenes. Regardless it brings into question the results of Figure 7, and potentially all of the L2 comparison results, depending on how frequently this issue occurs. This should be looked at more carefully.*

We have taken into consideration the time and longitude for the collocation. After careful consideration and checking the swaths we have found the reviewer comment to be correct. Hence we have replaced the Case study with a new swath on 19-10-2025 at 8:37 UTC. In order to avoid further such instances of previous swath issues of AVHRR PATMOS-x we have avoided all case studies after 22:00 UTC for all days. We only had around 5% of data affected by this exclusion. The overall analysis had very minute changes in decimal points and the final results were not affected. The changes are as follows:

Line (254) was added in Methodology 'While analysing the case study data, it was found that AP data intermittently substitutes observations from the preceding day in case of missing orbits towards the end of the day. Hence, all case studies after 22:00 UTC were discarded from the whole year's analyses.'

Line 379: The entire section was changed.

4.2.1 Case study

Figure 6 shows the 11 µm brightness temperature swaths for MODIS and CTH swath for MODIS (MYD), AVHRR PATMOS-x (AP) and AVHRR CMSAF (AC) and in black the merged CloudSat-CALIOP track for the 19-10-2015 at 08:37 UTC. Although the general pattern of CTH is similar between the three passive sensors, there are some significant differences. Although all three sensors see a similar high cloud pattern around 50∘S, AC retrieves the CTH around 11 km, while MYD and AP are around 9 km. AP has a significant region of cloud with CTHs ≈ 5 km at 40∘S and 80∘E, while for the same region, AC and MYD have low clouds (≈ 2 km).

Figure 7 shows a curtain plot for the CC track and the corresponding CTH, COD, CER, and CPH for each instrument. The top panel shows the CTH for all passive sensors (AC, AP, MYD, and CC), as well as the vertical profile of COD from the CC dataset (2BFL). The CTH for CC at a COD limit of 0.5 is also shown (cyan) to illustrate the potential differences in optical depth sensitivity between the passive and active sensors. The second panel illustrates the COD for the four sensors, along with the multilayer mask derived from CC. The COD was derived using the 3.7 µm channel as it is common to all passive sensors. The third panel displays the CER, as well as the sea ice mask used in the AP retrieval. The bottom panel shows the cloud phase for each instrument; for CC the phase is the phase detected at the top of the cloud. This example illustrates the highly variable nature of the cloud field across the Southern Ocean. Our discussion focuses on three sub-sections: multilayer clouds (Figure 7a[i]); thick upper-level clouds over thick low-level clouds (Figure 7a[ii]); and optically thick boundary layer clouds (Figure 7a[iii]).

In the first panel, the CTH ranges from 0 to 12 km and shows substantial differences between the CTH of passive and active sensors, with both underestimation and overestimation by passive sensors compared to CC. Figure 7a[i] shows a multilayer cloud structure with an optically thin high-level cloud over optically thick mid-level clouds. MODIS and AP detect the highlevel cloud, however since it is optically thin

over an optically thick lower layer cloud, it places the CTH at an effective CTH between the upper and lower layers. For the AC retrieval, the CTH is similar to the CC height. In Figure 7a[ii], AC produces higher CTH values than CC, MYD yields lower values, and AP's results are closest to those of CC. This difference could be due to the different numerical weather prediction (NWP) reanalyses used for temperature profiles, which can vary substantially, particularly at high altitudes and high latitudes, and the difficulty in identifying the inversion at the tropopause (Wu et al., 2012; Müller et al., 2018).

The section Figure 7a[iii] around 41.5∘S shows a region of boundary layer clouds. AC does a good job of retrieving the boundary layer cloud, while AP boundary layer CTH can show large over estimations, and MYD occasionally over and underestimates. Boundary layer clouds can be difficult to estimate, particularly when temperature inversions are present, as they can present multiple solutions when using the thermal characteristics to retrieve CTH (Huang et al., 2012; Wall et al., 2017; Truong et al., 2020). AP misclassifies the phase between 41.5∘S to 40∘S which may explain the high CTH retrieval and highlights some internal dependency in the retrieval approach.

The second panel displays the CC multilayer mask and identifies the areas with multilayer clouds well. The COD for each retrieval is also compared in the second panel. The COD of CC (2BFL) exhibits high variability when compared to the COD of the passive sensors, particularly for ML clouds. CC COD is substantially higher than that of the other sensors. The COD has large differences among the passive sensors and active sensors in the presence of a multilayer structure (Figure 7a[i] and [ii]); In single-layer areas, the COD of MYD, AP and AC are more in agreement with the active sensor measurements. The results align with the product statement for CC COD (Henderson and L'Ecuyer, 2023) which states that the COD is similar to MYD for single layer and varies with multilayer clouds. The third panel (Figure 7c) shows the CER comparison and the sea ice flag. While the general pattern of the CER from the passive sensors is in agreement, the values, particularly where ice clouds are detected are quite different, AP CER are higher than MYD and AC CER retrievals in general.

The cloud phase is shown in the fourth panel (Figure 7d). For CPH, each retrieval algorithm uses a slightly different phase classification scheme. The CC retrievals classify the phase as mixed, liquid or ice phase on the basis of cloud top and base temperatures, radar reflectivity and integrated attenuated backscattering coefficient across all the cloud layers in the vertical column (Wang, 2019). For the passive sensors (MYD, AP and AC), the CPH is the phase detected at the top of the cloud. In the case study, all sensors share ice and liquid as common classes, with an additional clear sky class for MYD, a supercooled liquid class for AP, and a mixed phase class for CC. Further classification of the phase can be seen in 4.2.4. The agreement between the passive sensors overall is good with occasional differences; for example, at 43∘ S, AC and MYD classify the cloud phase as liquid while CC and AP classify it as ice, with cloud optical depth indicating a thin upper layer.

To summarise, we observe significant variations in the retrieved cloud properties in the presence of multilayer clouds, with passive sensors both overestimating and underestimating the retrieved CTH compared to active sensors. The dependency on the optical depth on the topmost cloud layer, the penetrative property, and the sensitivity of passive sensors are seen in this case study. The case study demonstrates that cloud property retrievals differ based on cloud type, retrieval algorithm, and sensor sensitivity. Multilayer clouds often exhibit the greatest discrepancies.

Figure 6 changed from

*Figure 6 Swath for brightness temperature retrieved from MODIS (MYD) and merged CloudSat-CALIOP (CC) for Case study on 10-01-2015 at 22:50 UTC.*

[Figure]

To

Figure 6 Swath for a) brightness temperature retrieved from MODIS (MYD), b) Cloud top height retrieved from MODIS (MYD), c) AVHRR PATMOS-x and d) AVHRR CMSAF with swath plot of merged CloudSat-CALIOP (CC), shown in black, for Case study on 19-10-2015 at 08:37 UTC.

[Figure]

Figure 7 changed from

*Figure 7. Vertical profiles for cloud properties Cloud Top Height (CTH; Top panel), Cloud Optical Depth (COD; Second Panel), Cloud Effective Radius (CER) ; third panel) and Cloud Phase (CPH; Bottom Panel) retrieved from collocated data of AVHRR CMSAF (AC), AVHRR PATMOS (AP), MODIS (MYD) and merged CloudSat-CALIOP (CC) for Case study on 10-01-2015 at 22:50 UTC*

[Figure]

To

Figure 7 Vertical profiles for cloud properties Cloud Top Height (CTH; Top panel), Cloud Optical Depth (COD; Second Panel), Cloud Effective Radius (CER) ; third panel) and Cloud Phase (CPH; Bottom Panel) retrieved from collocated data of AVHRR CMSAF (AC), AVHRR PATMOS-x (AP), MODIS (MYD) and merged CloudSat-CALIOP (CC) for Case study on 19-10-2015 at 08:37 UTC.

[Figure]

Figure 8 replaced from

*Figure 8 Collocated Level 2 MODIS (top), AVHRR CMSAF (middle), AVHRR PATMOS-x (bottom) and CLOUDSATCALIOP merged data analysed for Cloud Top Height (CTH), over the Southern Ocean, for 2015 (left) shows the Joint 2D Histogram for CTHs, the 2D histograms for multilayer (right) and single-layer (centre) masked collocated data.*

To

[Figure]

Figure 8 Collocated Level 2 MODIS (top), AVHRR CMSAF (middle), AVHRR PATMOS-x (bottom) and CLOUDSATCALIOP merged data analysed for Cloud Top Height (CTH), over the Southern Ocean, for

2015 (left) shows the Joint 2D Histogram for CTHs, the 2D histograms for multilayer (right) and single-layer (centre) masked collocated data.

[Figure]

Table 5

From

| Sensors | True Positive Rate (TPR)/Hit rate | False Positive Rate (FPR) | Kuiper Skill Score (KSS) |
|---|---|---|---|
| CloudSat-CALIOP vs AVHRR CMSAF | 0.94 | 0.22 | 0.71 |
| CloudSat-CALIOP vs MODIS | 0.77 | 0.07 | 0.70 |
| CloudSat-CALIOP vs AVHRR PATMOS | 0.95 | 0.51 | 0.43 |

*Table 5. Cloud mask performance for 2015 over the Southern Ocean comparison for Level 2 AVHRR CMSAF, AVHRR PATMOS-x and MODIS against active sensor CloudSat-CALIOP (CC).*

To

| Sensors | True Positive Rate (TPR)/Hit rate | False Positive Rate (FPR) | Kuiper Skill Score (KSS) |
|---|---|---|---|
| CloudSat-CALIOP vs AVHRR CMSAF | 0.94 | 0.22 | 0.71 |
| CloudSat-CALIOP vs MODIS | 0.77 | 0.07 | 0.70 |
| CloudSat-CALIOP vs AVHRR PATMOS-x | 0.95 | 0.52 | 0.43 |

Table 5. Cloud mask performance for 2015 over the Southern Ocean comparison for Level 2 AVHRR CMSAF, AVHRR PATMOS-x and MODIS against active sensor CloudSat-CALIOP (CC).

Table 6

*From*

<table>
<tr><th colspan="6">Mean Values of CTHs</th></tr>
<tr><th></th><th>CloudSat_CALIOP (km) > 0.5 COD</th><th>CloudSat_CALIOP (km) without COD limit</th><th>AVHRR CMSAF</th><th>MODIS</th><th>AVHRR PATMOS (km)</th></tr>
<tr><td>2015</td><td>3.91</td><td>5.46</td><td>5.12</td><td>4.12</td><td>4.22</td></tr>
<tr><td>Single-layer cases</td><td>3.41</td><td>3.84</td><td>4.00</td><td>3.42</td><td>3.63</td></tr>
<tr><td>Multilayer Cases</td><td>4.81</td><td>8.40</td><td>7.18</td><td>5.42</td><td>5.32</td></tr>
</table>

(a)

<table>
<tr><th>Statistics</th><th colspan="4">CC vs AVHRR CMSAF</th><th colspan="4">CC vs MODIS</th><th colspan="4">CC vs AVHRR PATMOS</th></tr>
<tr><th></th><th>RMSE (km)</th><th>MAE (km)</th><th>MBE (km)</th><th>Correlation</th><th>RMSE (km)</th><th>MAE (km)</th><th>MBE (km)</th><th>Correlation</th><th>RMSE (km)</th><th>MAE (km)</th><th>MBE (km)</th><th>Correlation</th></tr>
<tr><td>2015</td><td>2.85</td><td>1.76</td><td>-1.21</td><td>0.72</td><td>2.54</td><td>1.56</td><td>-0.21</td><td>0.69</td><td>2.55</td><td>1.67</td><td>-0.32</td><td>0.65</td></tr>
<tr><td>SL Cases</td><td>1.99</td><td>1.12</td><td>-0.59</td><td>0.83</td><td>2.01</td><td>1.18</td><td>-0.01</td><td>0.78</td><td>2.18</td><td>1.37</td><td>-0.21</td><td>0.72</td></tr>
<tr><td>ML cases</td><td>3.97</td><td>2.94</td><td>-2.37</td><td>0.50</td><td>3.29</td><td>2.23</td><td>-0.60</td><td>0.52</td><td>3.11</td><td>2.21</td><td>-0.51</td><td>0.48</td></tr>
</table>

(b)

<table>
<tr><th>Statistics</th><th colspan="4">CC vs AVHRR CMSAF</th><th colspan="4">CC vs MODIS</th><th colspan="4">CC vs AVHRR PATMOS</th></tr>
<tr><th></th><th>RMSE (km)</th><th>MAE (km)</th><th>MBE (km)</th><th>Correlation</th><th>RMSE (km)</th><th>MAE (km)</th><th>MBE (km)</th><th>Correlation</th><th>RMSE (km)</th><th>MAE (km)</th><th>MBE (km)</th><th>Correlation</th></tr>
<tr><td>2015</td><td>2.00</td><td>1.08</td><td>0.33</td><td>0.86</td><td>2.90</td><td>1.80</td><td>1.33</td><td>0.75</td><td>3.05</td><td>2.03</td><td>1.23</td><td>0.69</td></tr>
<tr><td>SL cases</td><td>1.26</td><td>0.65</td><td>-0.16</td><td>0.93</td><td>1.68</td><td>1.03</td><td>0.42</td><td>0.88</td><td>2.03</td><td>1.31</td><td>0.21</td><td>0.80</td></tr>
<tr><td>ML Cases</td><td>2.89</td><td>1.86</td><td>1.22</td><td>0.64</td><td>4.31</td><td>3.22</td><td>2.99</td><td>0.53</td><td>4.34</td><td>3.34</td><td>3.08</td><td>0.44</td></tr>
</table>

(c)

*Table 6. a) Mean Cloud top height (CTH) values for 2015 over the SO for Level 2 CTHs data MODIS (MYD), CloudSat-CALIOP (CC), AVHRR CMSAF (AC) and AVHRR PATMOS-x (AP). Statistics of the comparison of CC CTH with the COD limit > 0.5 is shown in b) and c) is for CC CTH without the COD limit.*

*To*

<table>
<tr><th colspan="6">Mean Values of CTHs</th></tr>
<tr><th></th><th>CloudSat_CALIOP (km) > 0.5 COD</th><th>CloudSat_CALIOP (km) without COD limit</th><th>AVHRR CMSAF</th><th>MODIS</th><th>AVHRR PATMOS-x (km)</th></tr>
<tr><td>2015</td><td>3.92</td><td>5.48</td><td>5.15</td><td>4.12</td><td>4.25</td></tr>
<tr><td>Single-layer cases</td><td>3.43</td><td>3.86</td><td>4.00</td><td>3.42</td><td>3.63</td></tr>
<tr><td>Multilayer Cases</td><td>4.82</td><td>8.41</td><td>7.20</td><td>5.40</td><td>5.37</td></tr>
</table>

(a)

<table>
<tr><th>Statistics</th><th colspan="4">CC vs AVHRR CMSAF</th><th colspan="4">CC vs MODIS</th><th colspan="4">CC vs AVHRR PATMOS-x</th></tr>
<tr><th></th><th>RMSE (km)</th><th>MAE (km)</th><th>MBE (km)</th><th>Correlation</th><th>RMSE (km)</th><th>MAE (km)</th><th>MBE (km)</th><th>Correlation</th><th>RMSE (km)</th><th>MAE (km)</th><th>MBE (km)</th><th>Correlation</th></tr>
<tr><td>2015</td><td>2.87</td><td>1.77</td><td>-1.23</td><td>0.72</td><td>2.53</td><td>1.55</td><td>-0.19</td><td>0.69</td><td>2.53</td><td>1.65</td><td>-0.32</td><td>0.66</td></tr>
<tr><td>SL Cases</td><td>2.00</td><td>1.13</td><td>-0.59</td><td>0.83</td><td>2.00</td><td>1.18</td><td>0.01</td><td>0.78</td><td>2.16</td><td>1.35</td><td>-0.21</td><td>0.72</td></tr>
<tr><td>ML cases</td><td>3.98</td><td>2.94</td><td>-2.38</td><td>0.50</td><td>3.27</td><td>2.21</td><td>-0.58</td><td>0.52</td><td>3.09</td><td>2.19</td><td>-0.54</td><td>0.49</td></tr>
</table>

(b)

<table>
<tr><th>Statistics</th><th colspan="4">CC vs AVHRR CMSAF</th><th colspan="4">CC vs MODIS</th><th colspan="4">CC vs AVHRR PATMOS-x</th></tr>
<tr><th></th><th>RMSE (km)</th><th>MAE (km)</th><th>MBE (km)</th><th>Correlation</th><th>RMSE (km)</th><th>MAE (km)</th><th>MBE (km)</th><th>Correlation</th><th>RMSE (km)</th><th>MAE (km)</th><th>MBE (km)</th><th>Correlation</th></tr>
<tr><td>2015</td><td>2.00</td><td>1.08</td><td>0.33</td><td>0.86</td><td>2.90</td><td>1.81</td><td>1.36</td><td>0.75</td><td>3.01</td><td>2.00</td><td>1.23</td><td>0.70</td></tr>
<tr><td>SL cases</td><td>1.27</td><td>0.65</td><td>-0.16</td><td>0.93</td><td>1.67</td><td>1.02</td><td>0.44</td><td>0.88</td><td>1.97</td><td>1.28</td><td>0.22</td><td>0.81</td></tr>
<tr><td>ML Cases</td><td>2.88</td><td>1.85</td><td>1.21</td><td>0.64</td><td>4.31</td><td>3.23</td><td>3.01</td><td>0.54</td><td>4.29</td><td>3.30</td><td>3.05</td><td>0.45</td></tr>
</table>

(c)

Table 6. a) Mean Cloud top height (CTH) values for 2015 over the SO for Level 2 CTHs data MODIS (MYD), CloudSat-CALIOP (CC), AVHRR CMSAF (AC) and AVHRR PATMOS-x (AP). Statistics of the comparison of CC CTH with the COD limit > 0.5 is shown in b) and c) is for CC CTH without the COD limit.

*Table 8*

*From*

| Cloud Effective Radius | | | | | | | |
|---|---|---|---|---|---|---|---|
| **Phase** | **Sensors** | Full year | | single-layer | | Multi Layer | |
| | | Mean | Median | Mean | Median | Mean | Median |
| **Water** | AC | 14.75 | 13.37 | 14.01 | 13.45 | 14.19 | 12.66 |
| | MYD | 15.59 | 15.00 | 15.21 | 15.00 | 16.38 | 16.00 |
| | AP | 16.72 | 15.51 | 16.01 | 15.68 | 16.26 | 14.79 |
| **Ice** | AC | 15.54 | 14.19 | 14.82 | 14.80 | 14.50 | 13.06 |
| | MYD | 30.90 | 30.00 | 30.68 | 35.00 | 26.54 | 25.00 |
| | AP | 20.32 | 19.57 | 19.51 | 21.53 | 19.23 | 18.08 |
| **Supercooled Liquid** | AC | 14.68 | 13.34 | 13.98 | 13.60 | 14.24 | 12.86 |
| | MYD | nan | nan | nan | nan | nan | nan |
| | AP | 14.62 | 13.96 | 14.46 | 13.87 | 15.11 | 14.28 |

*Table 8. Mean and Median Cloud Effective Radius (CER) comparison for 2015 over the Southern Ocean, for water, ice and supercooled liquid phases from Level 2 AVHRR CMSAF (AC), AVHRR PATMOS-x (AP), MODIS (MYD) and CloudSat-CALIOP (CC).*

*To*

| Cloud Effective Radius [$\mu$m] | | | | | | | |
|---|---|---|---|---|---|---|---|
| **Phase** | **Sensors** | Full year | | single-layer | | Multi Layer | |
| | | Mean | Median | Mean | Median | Mean | Median |
| **Liquid** | AC | 14.75 | 13.36 | 14.81 | 13.45 | 14.16 | 12.59 |
| | MYD | 13.28 | 13.00 | 13.38 | 13.00 | 13.07 | 12.00 |
| | AP | 17.20 | 15.51 | 17.35 | 15.95 | 16.61 | 15.03 |
| **Ice** | AC | 15.57 | 14.24 | 16.20 | 14.84 | 14.53 | 13.15 |
| | MYD | 20.90 | 21.00 | 23.36 | 23.00 | 18.33 | 18.00 |
| | AP | 20.40 | 19.67 | 21.39 | 21.74 | 19.26 | 18.09 |
| **Supercooled Liquid** | AC | 14.65 | 13.31 | 14.91 | 13.59 | 14.18 | 12.78 |
| | MYD | nan | nan | nan | nan | nan | nan |
| | AP | 14.70 | 13.97 | 14.54 | 13.88 | 15.21 | 14.33 |

Table 8. Mean and Median Cloud Effective Radius (CER) comparison for 2015 over the Southern Ocean, for water, ice and supercooled liquid phases from Level 2 AVHRR CMSAF (AC), AVHRR PATMOS-x (AP), MODIS (MYD) and CloudSat-CALIOP (CC).

**Appendix**

**1. Collocation**

Below is a figure to demonstrate that all four sensors were collocated appropriately.

[Figure]

a)

b)

[Figure]

c)

**Appendix Figure 1:** a) Collocation of All sensors, b)Each sensor pixel distribution and c) 3 D plot for the collocation for each individual passive sensor with active sensor.

**2. Case Study Swaths for Clarifications**

The below figures show the case study swaths for the previous case study (09-01-2015; Appendix Figure 2) and the new case study (10-01-2015; Appendix Figure 3).

[Figure]

**Appendix Figure 2**: Cloud top height swaths of AVHRR PATMOS-x, AVHRR CMSAF and MODIS for 09-01-2015.

[Figure]

**Appendix Figure 3:** Cloud top height swaths of AVHRR PATMOS-x, AVHRR CMSAF and MODIS for 10-01-2015.

It's clearly visible that AVHRR PATMOS-x uses the previous day swath. Hence the case study was discarded.

**3. New Case Study Swaths for COD and CER**

[Figure]

**Appendix Figure 4:** Cloud optical depth swaths of a) MODIS, b) AVHRR PATMOS-x and c) AVHRR CMSAF for 19-10-2015.

[Figure]

**Appendix Figure 5:** Cloud effective radius swaths of a) MODIS, b) AVHRR PATMOS-x and c) AVHRR CMSAF and for 19-10-2015.

---

## Author Comment (AC2)

**General Replies to Comments**

Thank you for the valuable reviews and comments on the paper. We have amended the manuscript with regard to the issues raised following an extensive quality check and thorough re-evaluation. The abstract has also been updated accordingly. A brief summary of the key changes is given below.

All reviewers were concerned that the AVHRR data had not been properly collocated. The collocation method was thoroughly checked and found to be accurate.

However, Reviewer 1 identified an issue when a PATMOS swath ran across a calendar day: the initial day's record was erroneously being written onto the following day. This only occurs for a relatively small fraction of the total number of swaths (~5%), but these swaths have now been excluded from the analysis. The omission of these data did not qualitatively change the results or conclusions.

Figures 6, 7, 8, and 9 have been updated to reflect the changes made by excluding AVHRR PATMOS swaths after 22 UTC. Similarly, Tables 5, 6, and 8 have also been revised.

Unfortunately, the original case study chosen was one of these erroneous AVHRR swaths. As such, we have replaced the original case study from 10th January 2015 with a new one, 19th October 2015. This change in case study had no impact on the results or conclusions.

Please find the updated key changes, figures, and tables below.

In the general section, reviewers comments are included in *italics* with the original sentence in *'italics'* and changes in blue. The "Line #" is for the new modified manuscript.

**General key amendments to the manuscript.**

Listed below are the amendments to the manuscript according to the comments from Reviewer 1 (R1), Reviewer 2 (R2) and Community comments (CC1).

- Lines 64, 766 [R2]: The reference has been replaced with the suggested reference.

    *'Minnis et al., 2011'* with *'Minnis et al., 2020'*

    Minnis, P., Sun-Mack, S., Young, D., Heck, P., Garber, D., and Chen, Y.: CERES edition-2 cloud property retrievals using TRMM VIRS and Terra and Aqua MODIS data-Part I: Algorithms, IEEE Transactions on Geoscience and Remote Sensing, 49, 4374–4400, https://doi.org/10.1109/TGRS.2011.2144601, 2011.

    Minnis, P., Sun-Mack, S., Chen, Y., Chang, F.-L., Yost, C. R., Smith, W. L., Heck, P. W., Arduini, R. F., Bedka, S. T., Yi, Y., et al.: CERES MODIS cloud product retrievals for edition 4—Part I: Algorithm changes, IEEE Transactions on Geoscience and Remote Sensing, 59, 2744–2780, 2020.

- In Line 140 [R1, R2]: *'NASA'* has been replaced with 'NOAA'
- Line 142 [R1] : R1 noted *'It says here NOAA-19 is used for the L2 comparisons, but in Table 1 it says the cloud optical properties being compared (COD and CER) are derived from the 1.6-micron channel. Channel 3ab on the AVHRR can either measure the 1.6 or 3.75, but not at once. NOAA-19 is switched to 3b, meaning it measures the 3.75, so it shouldn't be possible to have 1.6-micron properties for CMSAF and PATMOS-x.'*

Thank you for pointing this out. The 1.6 µm CER retrievals have been replaced with 3.7 µm retrievals in all analyses.

Figure 9 and tables 9 and 10 have been updated along with all mention of the 1.6 µm channel. This has resulted in changes in MODIS results. These changes are highlighted and discussed later in this document.

*'The COD shown on each product uses the passive sensors' 1.6 µm channel.',*

Changed to

'The COD was derived using the 3.7 µm channel, as it is common to all passive sensors' Line (390).

| | CloudSat CALIOP | CALIOP | MODIS | AVHRR CMSAF | AVHRR PATMOS |
|---|---|---|---|---|---|
| Level 3 product | - | CALIOP GEWEX | CLDPROP_M3_MODIS_Aqua | CLARA-A ed.3.0 | - |
| CTH | - | Cloud_Top_Altitude_Passive | Cloud_Top_Height | Cloud Top Height | - |
| CFC | - | Cloud_Amount_Mean_Passive | Cloud_Fraction | Cloud Fraction | - |
| CPHL | - | Water_Cloud_Amount_Mean_Passive | Cloud_Retrieval_Fraction_Liquid | Cloud Fraction for liquid clouds | - |
| Level 2 product | 2B-CLDCLASS-LIDAR (2BCL) 2B-FLXHR-LIDAR (2BFL) | - | MYD06_L2 | CLARA-A ed.3.0 | PATMOS-x versions 6.0 |
| CTH | CloudLayerTop (2BCL) | - | Cloud Top Height | Cloud Top Height | ACHA CTH |
| COD | COD (2BFL) | - | 1.6µm COD | 1.6µm COD | 1.6µm COD DCOMP |
| CER | - | - | 1.6µm CER | 1.6µm CER | 1.6µm CER |
| CPH | CloudPhase (2BCL) | - | Cloud top phase | Cloud top phase extended | Cloud top phase |
| CMK | Cloud Mask | - | Cloud Mask | Cloud Mask | Cloud Mask |

*Table 1. Passive and active sensors Data and variables used for the Analysis.*

was changed to

| | CloudSat CALIOP | CALIOP | MODIS | AVHRR CMSAF | AVHRR PATMOS |
|---|---|---|---|---|---|
| Level 3 product | - | CALIOP GEWEX | CLDPROP_M3_MODIS_Aqua | CLARA-A ed.3.0 | - |
| CTH | - | Cloud_Top_Altitude_Passive | Cloud_Top_Height | Cloud Top Height | - |
| CFC | - | Cloud_Amount_Mean_Passive | Cloud_Fraction | Cloud Fraction | - |
| CPHL | - | Water_Cloud_Amount_Mean_Passive | Cloud_Retrieval_Fraction_Liquid | Cloud Fraction for liquid clouds | - |
| Level 2 product | 2B-CLDCLASS-LIDAR (2BCL) 2B-FLXHR-LIDAR (2BFL) | - | MYD06_L2 | CLARA-A ed.3.0 | PATMOS-x versions 6.0 |
| CTH | CloudLayerTop (2BCL) | - | Cloud Top Height | Cloud Top Height | ACHA CTH |
| COD | COD (2BFL) | - | 3.7µm COD | 3.7µm COD | 3.7µm COD DCOMP |
| CER | - | - | 3.7µm CER | 3.7µm CER | 3.7µm CER |
| CPH | CloudPhase (2BCL) | - | Cloud top phase | Cloud top phase extended | Cloud top phase |
| CMK | Cloud Mask | - | Cloud Mask | Cloud Mask | Cloud Mask |

Table 1. Passive and active sensors Data and variables used for the Analysis.

- Line 154 [R1,CC1]: *PATMOS* has been replaced with PATMOS-x for clarity throughout the manuscript.

- Line 161 [R1,CC1]: R1 and CC1 pointed out *"For version 6.0 ACHA uses 11 and 13.3 channels, not 11 and 12."*

  The sentence *'The ACHA CTH algorithm in this study uses 11, 12 μm observations along with'* was changed to 'The ACHA CTH algorithm in this study uses the 11 and 13.3 μm observations along with'

- Line 237 [R1] : R1 pointed out the confusion over the term granule. *"I'm a little confused using the term 'granules' here. THE AVHRR GAC data is reported in ascending and descending orbits. Is that the meaning of granule here, or is it the global L2b data?"*

  The sentences, *'4000 CloudSat-CALIOP granules were collocated with 20,000 MYD06, 365 AVHRR CMSAF and 365 AVHRR PATMOS-x granules. Each granule contained the cloud products such as CTH, COD, CER, CPH, latitude, longitude and common time of collocations.'* was changed to

  ' 4000 CloudSat-CALIOP orbits were collocated with 20,000 MYD06 granules, 365 AVHRR CMSAF and 365 AVHRR PATMOS-x global ascending orbit files. Each collocated file contained the cloud products such as CTH, COD, CER, CPH, latitude, longitude and common time of collocations.' for clarity.

- Line 274 [R2] :R2 commented *" It isn't clear what bin the uncertain pixels were put. Please explain."*

  *'Similarly, MYD06 cloud mask, confidently clear (11), probably clear (10), uncertain (01) and cloudy (00) were classified into cloudy (0) and clear (1)'* has been modified to

  'Similarly, MYD06 cloud masks were classified as cloudy (0), both uncertain (01) and cloudy (00) and as clear (1), both confidently clear (11) and probably clear (10).'

- Line 298-306 [R2]: R2 commented on *"Check to see that the values in the text match those in the table. I think there are some discrepancies"*. Mismatched values from text reviewed and corrected accordingly and is highlighted below.

  *'Table 2 summarises the comparisons, with mean CTH values in Table 2a and the statistics for the comparison with CALIOP in Table 2b by season and overall. It can be observed that the active sensor correlation with the passive sensors for the overall monthly CTHs is low with values ranging from 0.35 for CALIOP-AVHRR CMSAF (C-A) and 0.32 for CALIOP-MODIS (CM). The CTH difference between the C-A sensors for DJF ranges from -1 to 1 km for the study domain, while for C-M it ranges from 0 to 4 km. The MBE for passive with the active sensor is 0.46 km for C-A and 1.12 km for C-M. The RMSE for C-A is 2.33 km; for C-M, the value is 2.60 km; and for A-M, it is 1.23 km, which is consistent with both correlation and MBE trend. Overall, the MBE between the MODIS-AVHRR CMSAF (A-M) passive sensors is larger for the CLARA-A3 dataset (0.95 km), but the correlation is better (0.67). Overall, there is a positive mean bias error for the passive sensors AVHRR CMSAF and MODIS CTHs.'*

has been modified to

'Table 2 summarises the comparisons, with mean CTH values in Table 2a and the statistics for the comparison with CALIOP in Table 2b by season and overall. It can be observed that the active sensor correlation with the passive sensors for the overall monthly CTHs is low with values ranging from 0.36 for CALIOP-AVHRR CMSAF (C-A) and 0.32 for CALIOP-MODIS (CM). The CTH difference between the C-A sensors for DJF ranges from -1 to 1 km for the study domain, while for C-M it ranges from 0 to 4 km. The MBE for passive with the active sensor is 0.46 km for C-A and 1.12 km for C-M. The RMSE for C-A is 2.35 km; for C-M, the value is 2.60 km; and for A-M, it is 0.98 km, which is consistent with both correlation and MBE trend. Overall, the MBE between the MODIS-AVHRR CMSAF (A-M) passive sensors is larger for the CLARA-A3 dataset (0.98 km), but the correlation is better (0.68). Overall, there is a positive mean bias error for the passive sensors AVHRR CMSAF and MODIS CTHs.'

- Line 355 [R2]: removed *"this"* from the sentence

  *'The overall cloud fraction of liquid water clouds (hereafter CFL) and seasonal distributions are illustrated in this Figure 5.'*

  has been modified to

  'The overall cloud fraction of liquid water clouds (hereafter CFL) and seasonal distributions are illustrated in Figure 5.'

- Line 360 [R2]: Redundant sentence *'In the A-M seven-year monthly comparison, the overall correlation stands at a higher level (0.71) compared to both passive-active comparisons.'* has been removed.

- Line 390 [R2]: changed *'uses'* to *'was derived using'*.
- Line 409 [R1]: The Spelling typo *'retreive'* was corrected to *retrieve.*
- Line 438 [R1]: R1 commented, *"CMSAF and PATMOS-x both use AVHRR GAC data so they shouldn't have different spatial resolutions for L2 products. The sampling may be different, but that's not the same thing."*
  Hence, the sentence, *'The higher resolution of MODIS and AVHRR CMSAF may contribute to their higher KSS compared to AVHRR PATMOS.'* was amended to "The higher resolution of MODIS may contribute to the higher KSS compared to AVHRR PATMOS-x."

- Line 467-469 [R2]: R2 commented, *"CTH 'overestimates' are due to inversions, where the BT matches the profile temperature at a 'lower' height. Do you mean higher height?"*

  *'An explanation for the overestimation of low CTH is the effect of atmospheric temperature inversion, where the cloud brightness temperature taken by the passive sensor from the NWP reanalyses profile might be for a lower height than the actual CTH (Figure 7a[i]).'*

Changed to

An explanation for the overestimation of low CTH is the presence of a sharp temperature inversion that provides two solutions for the same IR temperature from the NWP reanalyse profile, one above and one below the actual CTH (Figure 7a[iii], Dong et al., 2008). A new reference was added to the line.

And references added to References section in Line 638

Dong, X., Minnis, P., Xi, B., Sun-Mack, S., and Chen, Y.: Comparison of CERES-MODIS stratus cloud properties with ground-based measurements at the DOE ARM Southern Great Plains site, Journal of Geophysical Research: Atmospheres, 113, 2008.

- Line 493- 529[R2]: R2 commented *"Section lacks discussion regarding difference across algorithms with respect to CER magnitudes (i.e. why is MODIS ice Re so much higher than the others), and CER distributions (i.e. why does AC ice look like AC water, unlike AP and MODIS)"*

The discussion has been expanded and changed as part of the change in the MODIS retrieval channel (3.7 µm to 1.6 µm) as below. When the MODIS retrieval was changed to 3.7 µm, the values of CER became similar to other passive sensors, and the spread of the CER decreased.

4.2.4 L2 Cloud Effective Radius Comparison

This section compares CER from the passive sensors as a function of CPH and the presence of single or multilayer clouds. The passive sensors categorise thermodynamic phase into various classifications according to their retrieval algorithms; for example, MYD has five phase categories, while AC has eight, including a supercooled water class. The categories used by each dataset are presented in Table 7. In the results presented, the number of classes was reduced: classifications of water, fog, and supercooled liquid are grouped as liquid; ice, cirrus, and overlap are treated as ice. When this generalised CPH classification is used, more liquid-phase clouds are found in SL cases, while ice dominates in ML cases. A comparison between the different sensors AC, MYD, and AP is illustrated in Figure 9 for the year 2015, stratified for SL and ML cases, and summarised in Table 8.

The liquid CER probability density functions (PDFs) are in broad agreement across the three sensors for all cases. The mean values for liquid clouds are approximately 14 µm for AC, 13 µm for MYD, and 17 µm for AP. The higher CER values from AP suggest potential misclassification of liquid clouds as ice clouds as the proportion of ice clouds in AP is over five times higher than that of liquid clouds. The liquid phase occurs more frequently in SL clouds than ML clouds across all sensors, with similar mean values for both cases (Table 8).

In contrast, the CER comparisons for ice-phase clouds reveal notable differences across the sensors. The overall mean for AC is 15.57 µm, while MYD and AP report means of 20.90 µm and 20.40 µm, respectively. AP also exhibits a distinct bimodal CER distribution with

peaks at approximately 10 µm and 30 µm, particularly evident in SL ice-phase retrievals. ML cases show more similar CER distributions across the sensors. The SL–ML CER difference is relatively small for AC ($\approx 1$µm), modest for AP ($\approx 2$ µm), and larger for MYD ($\approx 5$ µm). Sensitivity to ice crystal habit selection and the corresponding LUT structure also influences retrievals. MYD uses scattering functions derived from the General Habit Mixture (GHM) developed by Baum et al. (2005a, b) and Yang et al. (2013), implemented as band-specific LUTs (Platnick et al., 2016; Amarasinghe et al., 16 2017). The ice habitat selection includes severely roughened aggregates, hollow columns, bullet rosettes, plates, and droxtals. AC uses radiative transfer simulations based on narrower droplet size distributions and an ice crystal model including severely roughened aggregated solid columns, hollow columns, and bullet rosettes (Yang et al., 2013; Baum et al., 2011). AP uses LUTs from Baum/Yang scattering models with reduced spectral dimensionality , and its ice habit selection includes roughened bullet rosettes, aggregates, and hollow columns (Walther and Heidinger, 2012; Heidinger and Foster, 2021). These distinctions in habit selection and LUT configuration influence the consistency and comparability of retrieved cloud microphysical properties across sensors.

Supercooled liquid clouds are consistently retrieved by both AC and AP, particularly in SL regimes. AC reports a substantial supercooled component in full-year statistics, with a mean CER of 14.65 µm, due to its extended phase classification (Karlsson et al., 2023a). AP also distinguishes supercooled liquid from warm liquid, frequently identifying it in SL clouds, with a slightly higher mean CER of 17.20 µm. MYD lacks a dedicated supercooled category but includes an "undetermined liquid" retrieval class (ULQ; mean 19.71 µm), likely encompassing supercooled or glaciating clouds not clearly resolved by its phase algorithm (Platnick et al., 2016). Both AC and AP indicate persistent supercooled liquid presence over the Southern Ocean, consistent with prior observational studies (Huang et al., 2015; Bodas-Salcedo et al., 2016).

The differences observed between ML and SL cases suggest that multilayer clouds influence retrieval outcomes. Yost et al. (2023) had similar findings and demonstrated that multilayer conditions, especially non-opaque clouds over liquid, are among the leading sources of phase retrieval uncertainty in passive satellite observations.

Figure 9 replaced from

[Figure]

To

[Figure]

Figure 9 Cloud Effective Radius (CER) distribution according to the cloud phase for the collocated data for the year 2015, over the Southern Ocean, for all passive sensor

*Table 8*

| Cloud Effective Radius | | Full year | | single-layer | | Multi Layer | |
|---|---|---|---|---|---|---|---|
| **Phase** | **Sensors** | Mean | Median | Mean | Median | Mean | Median |
| **Water** | AC | 14.75 | 13.37 | 14.01 | 13.45 | 14.19 | 12.66 |
| | MYD | 15.59 | 15.00 | 15.21 | 15.00 | 16.38 | 16.00 |
| | AP | 16.72 | 15.51 | 16.01 | 15.68 | 16.26 | 14.79 |
| **Ice** | AC | 15.54 | 14.19 | 14.82 | 14.80 | 14.50 | 13.06 |
| | MYD | 30.90 | 30.00 | 30.68 | 35.00 | 26.54 | 25.00 |
| | AP | 20.32 | 19.57 | 19.51 | 21.53 | 19.23 | 18.08 |
| **Supercooled Liquid** | AC | 14.68 | 13.34 | 13.98 | 13.60 | 14.24 | 12.86 |
| | MYD | nan | nan | nan | nan | nan | nan |
| | AP | 14.62 | 13.96 | 14.46 | 13.87 | 15.11 | 14.28 |

*From*

*Table 8. Mean and Median Cloud Effective Radius (CER) comparison for 2015 over the Southern Ocean, for water, ice and supercooled liquid phases from Level 2 AVHRR CMSAF (AC), AVHRR PATMOS-x (AP), MODIS (MYD) and CloudSat-CALIOP (CC).*

*To*

| Cloud Effective Radius [$\mu$m] | | Full year | | single-layer | | Multi Layer | |
|---|---|---|---|---|---|---|---|
| **Phase** | **Sensors** | Mean | Median | Mean | Median | Mean | Median |
| **Liquid** | AC | 14.75 | 13.36 | 14.81 | 13.45 | 14.16 | 12.59 |
| | MYD | 13.28 | 13.00 | 13.38 | 13.00 | 13.07 | 12.00 |
| | AP | 17.20 | 15.51 | 17.35 | 15.95 | 16.61 | 15.03 |
| **Ice** | AC | 15.57 | 14.24 | 16.20 | 14.84 | 14.53 | 13.15 |
| | MYD | 20.90 | 21.00 | 23.36 | 23.00 | 18.33 | 18.00 |
| | AP | 20.40 | 19.67 | 21.39 | 21.74 | 19.26 | 18.09 |
| **Supercooled Liquid** | AC | 14.65 | 13.31 | 14.91 | 13.59 | 14.18 | 12.78 |
| | MYD | nan | nan | nan | nan | nan | nan |
| | AP | 14.70 | 13.97 | 14.54 | 13.88 | 15.21 | 14.33 |

Table 8. Mean and Median Cloud Effective Radius (CER) comparison for 2015 over the Southern Ocean, for water, ice and supercooled liquid phases from Level 2 AVHRR CMSAF (AC), AVHRR PATMOS-x (AP), MODIS (MYD) and CloudSat-CALIOP (CC).

- Line 569-572 [R2]: Changed > to greater and < to lesser in text as per reviewer comments
  *'In comparison to CALIOP CFC, the AVHRR CMSAF CFC was > 20% and the MODIS CFC was > 10%. On the other hand, over the high latitudes (60∘S to 70∘ S), MODIS CFC*

*was < 10% compared to CALIOP CFC. We observed that over Antarctica, CALIOP CFC was > 10% of MODIS CFC and > 20% of AVHRR CMSAF.'* to ' In comparison to CALIOP CFC, the AVHRR CMSAF CFC was 20% greater and the MODIS CFC was 10% greater. On the other hand, over the high latitudes (60∘S to 70∘), MODIS CFC was 10% less compared to CALIOP CFC. We observed that over Antarctica, CALIOP CFC was 10% greater than MODIS CFC and 20% greater than AVHRR CMSAF.'

- Line 551-580 [R1]:R1 commented *It says here NOAA-19 is used for the L2 comparisons, but in Table 1 it says the cloud optical properties being compared (COD and CER) are derived from the 1.6-micron channel. Channel 3ab on the AVHRR can either measure the 1.6 or 3.75, but not at once. NOAA-19 is switched to 3b, meaning it measures the 3.75, so it shouldn't be possible to have 1.6-micron properties for CMSAF and PATMOS-x.*

Thank you for pointing this out, the section was changed due to the 1.6 µm CER retrievals being replaced with 3.7 µm CER retrievals in the analyses as below. Figure 10 and table 9 were also replaced: Line 530-560.

4.2.5 L2 cloud property comparison as a function of surface type

In this section, we investigate the distribution of cloud properties as a function of sea ice coverage. The sea ice flag in AVHRR PATMOS-x(AP) retrieval was used to define the presence of sea ice, based on fractional sea ice coverage from NOAA/NCEP Climate Forecast System Reanalysis (CFSR) data (Foster et al., 2023). The distributions of the cloud properties CER, CTH, and COD were analysed as a function of sea ice coverage for latitudes > 60∘S, as shown in Figure 10. The narrow latitude range was selected to reduce the differences in microphysics caused by different meteorological regimes. The MYD cloud properties are shown in red, AC in black, and AP in yellow. Cloud properties such as COD, CER, and CTH are compared for sea ice detected and non-sea ice detected; the mean and median are summarised in Table 9.

The mean COD for AC was 33.9 and 14.4; for MYD it was 24.1 and 18.6; and for AP it was 41.7 and 12.7 for sea ice and non-sea ice cases, respectively. The optical depth of the retrieved pixels increased in the presence of sea ice for all sensors. The results disagree with the findings of Frey et al. (2018) and Palm et al. (2010), which concluded that clouds were more frequent, and cloud optical depth tends to be higher over ice free regions. While the disagreement could arise from differences in latitude range and the limited temporal scope of our dataset compared to the multi-year analyses Frey et al. (2018) and Palm et al. (2010). It is hypothesised that the difference is due to two effects both originating in flaws of the retrieval algorithms: 1 - sea ice (with no associated cloud) is misidentified as cloud; this could be the cause of the spike in observations with greater than 100 COD over sea ice, and 2 - the albedo of sea ice used in the retrieval algorithms introducing a systematic positive COD 17 bias. It is difficult to define an accurate sea ice albedo; a systematic underestimation of the albedo could cause the optical depth of clouds to be overestimated.

In the comparison of CTH and CER between sea ice and non-sea ice cases, the differences for AP are small (0.1 µm for CER and -0.01 km for CTH). The MYD and AC products also show small positive differences in CER (≈1 µm) over sea ice. For non sea ice cases, the CTHs are 0.05 km (MYD) and 0.6 km (AC) lower than sea ice cases. Although CER and CTH exhibit positive differences, these were much smaller than the differences observed for COD.

The seasonality of the COD, CER, and CTH anomalies over sea ice was also examined. We observed that the disagreement between retrievals over sea ice and non sea ice is higher in JJA, where sea ice cases show (not shown here) lower COD and CTH values. COD values remain below 50 for AC and MYD and below 80 for AP across all seasons, with generally lower values over sea ice in winter (JJA). CER distributions are relatively consistent across seasons, ranging from 0–35 µm for AP, 5–65 µm for MYD, and 5–45 µm for AC. Seasonal variation is more evident in CTH: low clouds (less than 2 km) dominate for MYD and AP in JJA and for all three sensors in SON, while DJF and MAM show a greater presence of mid-level clouds (3–6 km), though low clouds remain prominent for MYD and AP. It can be concluded that in the case of CTH and CER, the seasonality is not an important factor, while for COD, the seasonality and presence of sea ice are important factors in the retrieval of the product.

Figure 10 replaced from

*Figure 10 Cloud Optical Depth (COD), Cloud Effective Radius (CER) and Cloud Top Height (CTH) as a function of sea ice Flag from AVHRR PATMOS-x for all passive sensors, over the Southern Ocean less than 60OS. MODIS (MYD) is shown in red, AVHRR CMSAF (AC) is shown in black and AVHRR PATMOS-x (AP) is shown in yellow.*

[Figure]

To

Figure 10 Cloud Optical Depth (COD), Cloud Effective Radius (CER) and Cloud Top Height (CTH) as a function of Sea ice Flag from AVHRR PATMOS-x for all passive sensors, over the Southern Ocean less than 60 degrees S. MODIS (MYD) is shown in red, AVHRR CMSAF (AC) is shown in black and AVHRR PATMOS-x (AP) is shown in yellow.

[Figure]

*Table 9 From*

| | COD | | | | CER | | | | CTH | | | |
|---|---|---|---|---|---|---|---|---|---|---|---|---|
| | Sea ice | | Non Seaice | | Sea ice | | Non Seaice | | Sea ice | | Non Seaice | |
| **Sensors** | Mean | Median | Mean | median | Mean | Median | Mean | median | Mean | Median | Mean | median |
| MYD | 23.30 | 13.0 | 20.46 | 13.0 | 20.82 | 16.0 | 19.45 | 15.00 | 4.15 | 3.3 | 3.87 | 2.75 |
| ACM | 25.98 | 11.46 | 19.19 | 10.35 | 14.25 | 12.66 | 13.01 | 11.32 | 4.91 | 4.17 | 4.27 | 3.56 |
| AP | 28.30 | 13.31 | 17.37 | 9.952 | 16.38 | 13.89 | 14.55 | 11.94 | 3.85 | 3.95 | 3.67 | 3.79 |

*Table 9.Mean and Median values of Cloud Optical Depth (COD), Cloud Effective Radius (CER) and Cloud Top Height (CTH) over sea and over sea ice for Level 2 AVHRR CMSAF, AVHRR PATMOS-x and MODIS pixels at latitudes (> 60∘ S).*

*To*

| | COD | | | | CER [$\mu$m] | | | | CTH [km] | | | |
|---|---|---|---|---|---|---|---|---|---|---|---|---|
| | Sea ice | | Non Seaice | | Sea ice | | Non Seaice | | Sea ice | | Non Seaice | |
| **Sensors** | Mean | Median | Mean | median | Mean | Median | Mean | median | Mean | Median | Mean | median |
| MYD | 24.08 | 13.00 | 18.55 | 12.00 | 15.92 | 13.00 | 14.46 | 12.00 | 4.02 | 3.30 | 3.93 | 2.80 |
| ACM | 33.89 | 11.46 | 14.44 | 9.17 | 14.02 | 12.25 | 13.01 | 11.91 | 5.07 | 4.52 | 4.40 | 3.60 |
| AP | 41.71 | 24.28 | 12.70 | 8.37 | 15.64 | 13.37 | 15.54 | 12.94 | 3.74 | 3.62 | 3.75 | 3.85 |

Table 9. Mean and Median values of Cloud Optical Depth (COD), Cloud Effective Radius (CER) and Cloud Top Height (CTH) over sea and over sea ice for Level 2 AVHRR CMSAF, AVHRR PATMOS-x and MODIS pixels at latitudes (> 60∘ S).

- Line 603 [R1]:R1 pointed out *''The AVHRR PATMOS sensors exhibited the lowest correlation and the highest bias in both the overall CTH and the multilayer and single-layer cases.' The numbers reported in Figure 8 don't seem to support this conclusion. The PATMOS-x RMSE and MBE is lower than MYD and CMSAF for the multi-layer cases and lower than CMSAF for all cases.'*

  The sentence was amended to exclude the incorrect conclusion. Hence *'The AVHRR PATMOS sensors exhibited the lowest correlation and the highest bias in both the overall CTH and the multilayer and single-layer cases.'* amended to Line 603 - 'The AVHRR PATMOS-x retrievals exhibited the lowest correlation and the highest bias in both the overall CTH and the multilayer and single-layer cases.'

- Line 610 [R2]: R2 commented incomplete sentence. Hence, removed *'Since reanalysis and climate models utilise these passive sensor retrievals.'*

- The case study portion was amended in response to Reviewer 1's notice concerning a flaw in the AVHRR PATMOS algorithm, which intermittently substitutes observations from the preceding day. The initial case study was impacted by this issue and subsequently withdrawn. Furthermore, all case studies with observation hours exceeding 22:00 UTC were omitted to prevent similar occurrences. This time threshold resulted in a reduction in only 5% of the collocations analysed in this study. Hence, Figures 6, 7, 8, 9 and 10 and Tables 5, 6, 8 and 9 have been amended with the changes. The changes did not significantly affect the overall results, aside from minor variations in decimal values.The following are the text changes made in the Manuscript.

  Line 254: Line was added in Methodology 'While analysing the case study data, it was found that AP data intermittently substitutes observations from the preceding day in case of missing orbits towards the end of the day. Hence, all case studies after 22:00 UTC were discarded from the whole year's analyses.'

  Line 379 : The entire section was changed as below.

  4.2.1 Case study

  Figure 6 shows the 11 μm brightness temperature swaths for MODIS and CTH swath for MODIS (MYD), AVHRR PATMOS-x (AP) and AVHRR CMSAF (AC) and in black the merged CloudSat-CALIOP track for the 19-10-2015 at 08:37 UTC. Although the general pattern of CTH is similar between the three passive sensors, there are some significant differences. Although all three sensors see a similar high cloud pattern around 50∘S, AC retrieves the CTH around 11 km, while MYD and AP are around 9 km. AP has a significant region of cloud with CTHs ≈ 5 km at 40∘S and 80∘E, while for the same region, AC and MYD have low clouds (≈ 2 km).

  Figure 7 shows a curtain plot for the CC track and the corresponding CTH, COD, CER, and CPH for each instrument. The top panel shows the CTH for all passive sensors (AC, AP, MYD, and CC), as well as the vertical profile of COD from the CC dataset (2BFL). The CTH for CC at a COD limit of 0.5 is also shown (cyan) to illustrate the potential

differences in optical depth sensitivity between the passive and active sensors. The second panel illustrates the COD for the four sensors, along with the multilayer mask derived from CC. The COD was derived using the 3.7 µm channel as it is common to all passive sensors. The third panel displays the CER, as well as the sea ice mask used in the AP retrieval. The bottom panel shows the cloud phase for each instrument; for CC the phase is the phase detected at the top of the cloud. This example illustrates the highly variable nature of the cloud field across the Southern Ocean. Our discussion focuses on three sub-sections: multilayer clouds (Figure 7a[i]); thick upper-level clouds over thick low-level clouds (Figure 7a[ii]); and optically thick boundary layer clouds (Figure 7a[iii]).

In the first panel, the CTH ranges from 0 to 12 km and shows substantial differences between the CTH of passive and active sensors, with both underestimation and overestimation by passive sensors compared to CC. Figure 7a[i] shows a multilayer cloud structure with an optically thin high-level cloud over optically thick mid-level clouds. MODIS and AP detect the highlevel cloud, however since it is optically thin over an optically thick lower layer cloud, it places the CTH at an effective CTH between the upper and lower layers. For the AC retrieval, the CTH is similar to the CC height. In Figure 7a[ii], AC produces higher CTH values than CC, MYD yields lower values, and AP's results are closest to those of CC. This difference could be due to the different numerical weather prediction (NWP) reanalyses used for temperature profiles, which can vary substantially, particularly at high altitudes and high latitudes, and the difficulty in identifying the inversion at the tropopause (Wu et al., 2012; Müller et al., 2018).

The section Figure 7a[iii] around 41.5∘S shows a region of boundary layer clouds. AC does a good job of retrieving the boundary layer cloud, while AP boundary layer CTH can show large over estimations, and MYD occasionally over and underestimates. Boundary layer clouds can be difficult to estimate, particularly when temperature inversions are present, as they can present multiple solutions when using the thermal characteristics to retrieve CTH (Huang et al., 2012; Wall et al., 2017; Truong et al., 2020). AP misclassifies the phase between 41.5∘S to 40∘S which may explain the high CTH retrieval and highlights some internal dependency in the retrieval approach.

The second panel displays the CC multilayer mask and identifies the areas with multilayer clouds well. The COD for each retrieval is also compared in the second panel. The COD of CC (2BFL) exhibits high variability when compared to the COD of the passive sensors, particularly for ML clouds. CC COD is substantially higher than that of the other sensors. The COD has large differences among the passive sensors and active sensors in the presence of a multilayer structure (Figure 7a[i] and [ii]); In single-layer areas, the COD of MYD, AP and AC are more in agreement with the active sensor measurements. The results align with the product statement for CC COD (Henderson and L'Ecuyer, 2023) which states that the COD is similar to MYD for single layer and varies with multilayer clouds. The third panel (Figure 7c) shows the CER comparison and the sea ice flag. While the general pattern of the CER from the passive sensors is in

agreement, the values, particularly where ice clouds are detected are quite different, AP CER are higher than MYD and AC CER retrievals in general.

The cloud phase is shown in the fourth panel (Figure 7d). For CPH, each retrieval algorithm uses a slightly different phase classification scheme. The CC retrievals classify the phase as mixed, liquid or ice phase on the basis of cloud top and base temperatures, radar reflectivity and integrated attenuated backscattering coefficient across all the cloud layers in the vertical column (Wang, 2019). For the passive sensors (MYD, AP and AC), the CPH is the phase detected at the top of the cloud. In the case study, all sensors share ice and liquid as common classes, with an additional clear sky class for MYD, a supercooled liquid class for AP, and a mixed phase class for CC. Further classification of the phase can be seen in 4.2.4. The agreement between the passive sensors overall is good with occasional differences; for example, at 43∘ S, AC and MYD classify the cloud phase as liquid while CC and AP classify it as ice, with cloud optical depth indicating a thin upper layer.

To summarise, we observe significant variations in the retrieved cloud properties in the presence of multilayer clouds, with passive sensors both overestimating and underestimating the retrieved CTH compared to active sensors. The dependency on the optical depth on the topmost cloud layer, the penetrative property, and the sensitivity of passive sensors are seen in this case study. The case study demonstrates that cloud property retrievals differ based on cloud type, retrieval algorithm, and sensor sensitivity. Multilayer clouds often exhibit the greatest discrepancies.

Figure 6 changed from

*Figure 6. Swath for brightness temperature retrieved from MODIS (MYD) and merged CloudSat-CALIOP (CC) for Case study on 10-01-2015 at 22:50 UTC.*

[Figure]

To

Figure 6. Swath for a) brightness temperature retrieved from MODIS (MYD), b) Cloud top height retrieved from MODIS (MYD), c) AVHRR PATMOS-x and d) AVHRR CMSAF with the track of merged CloudSat-CALIOP (CC), shown in black, for a case study on 19-10-2015 at 08:37 UTC.

[Figure]

Figure 7 changed from

*Figure 7. Vertical profiles for cloud properties Cloud Top Height (CTH; Top panel), Cloud Optical Depth (COD; Second Panel), Cloud Effective Radius (CER) ; third panel) and Cloud Phase (CPH; Bottom Panel) retrieved from collocated data of AVHRR CMSAF (AC), AVHRR PATMOS (AP), MODIS (MYD) and merged CloudSat-CALIOP (CC) for Case study on 10-01-2015 at 22:50 UTC*

[Figure]

To

Figure 7. Vertical profiles for cloud properties Cloud Top Height (CTH; Top panel), Cloud Optical Depth (COD; Second Panel), Cloud Effective Radius (CER) ; third panel) and Cloud Phase (CPH; Bottom Panel) retrieved from collocated data of AVHRR CMSAF (AC), AVHRR PATMOS-x (AP), MODIS (MYD) and merged CloudSat-CALIOP (CC) for Case study on 19-10-2015 at 08:37 UTC

[Figure]

Figure 8 replaced from

*Figure 8 Collocated Level 2 MODIS (top), AVHRR CMSAF (middle), AVHRR PATMOS-x (bottom) and CLOUDSAT-CALIOP merged data analysed for Cloud Top Height (CTH), over the Southern Ocean, for 2015 (left) shows the Joint 2D Histogram for CTHs, the 2D histograms for multilayer (right) and single-layer (centre) masked collocated data.*

[Figure]

To

Figure 8 Collocated Level 2 MODIS (top), AVHRR CMSAF (middle), AVHRR PATMOS-x (bottom) and CLOUDSAT CALIOP merged data analysed for Cloud Top Height (CTH), over the Southern Ocean, for 2015 (left) shows the Joint 2D Histogram for CTHs, the 2D histograms for multilayer (right) and single-layer (centre) masked collocated data.

[Figure]

*Table 5*

*From*

| Sensors | True Positive Rate (TPR)/Hit rate | False Positive Rate (FPR) | Kuiper Skill Score (KSS) |
|---|---|---|---|
| CloudSat-CALIOP vs AVHRR CMSAF | 0.94 | 0.22 | 0.71 |
| CloudSat-CALIOP vs MODIS | 0.77 | 0.07 | 0.70 |
| CloudSat-CALIOP vs AVHRR PATMOS | 0.95 | 0.51 | 0.43 |

*Table 5. Cloud mask performance for 2015 over the Southern Ocean comparison for Level 2 AVHRR CMSAF, AVHRR PATMOS-x and MODIS against active sensor CloudSat-CALIOP (CC)*

*To*

| Sensors | True Positive Rate (TPR)/Hit rate | False Positive Rate (FPR) | Kuiper Skill Score (KSS) |
|---|---|---|---|
| CloudSat-CALIOP vs AVHRR CMSAF | 0.94 | 0.22 | 0.71 |
| CloudSat-CALIOP vs MODIS | 0.77 | 0.07 | 0.70 |
| CloudSat-CALIOP vs AVHRR PATMOS-x | 0.95 | 0.52 | 0.43 |

Table 5. Cloud mask performance for 2015 over the Southern Ocean comparison for Level 2 AVHRR CMSAF, AVHRR PATMOS-x and MODIS against active sensor CloudSat-CALIOP (CC*)*

*Table 6*

*From*

| Mean Values of CTHs | | | | | |
|---|---|---|---|---|---|
| | CloudSat_CALIOP (km) > 0.5 COD | CloudSat_CALIOP (km) without COD limit | AVHRR CMSAF | MODIS | AVHRR PATMOS (km) |
| 2015 | 3.91 | 5.46 | 5.12 | 4.12 | 4.22 |
| Single-layer cases | 3.41 | 3.84 | 4.00 | 3.42 | 3.63 |
| Multilayer Cases | 4.81 | 8.40 | 7.18 | 5.42 | 5.32 |

(a)

| Statistics | CC vs AVHRR CMSAF | | | | CC vs MODIS | | | | CC vs AVHRR PATMOS | | | |
|---|---|---|---|---|---|---|---|---|---|---|---|---|
| | RMSE (km) | MAE (km) | MBE (km) | Correlation | RMSE (km) | MAE (km) | MBE (km) | Correlation | RMSE (km) | MAE (km) | MBE (km) | Correlation |
| 2015 | 2.85 | 1.76 | -1.21 | 0.72 | 2.54 | 1.56 | -0.21 | 0.69 | 2.55 | 1.67 | -0.32 | 0.65 |
| SL Cases | 1.99 | 1.12 | -0.59 | 0.83 | 2.01 | 1.18 | -0.01 | 0.78 | 2.18 | 1.37 | -0.21 | 0.72 |
| ML cases | 3.97 | 2.94 | -2.37 | 0.50 | 3.29 | 2.23 | -0.60 | 0.52 | 3.11 | 2.21 | -0.51 | 0.48 |

(b)

| Statistics | CC vs AVHRR CMSAF | | | | CC vs MODIS | | | | CC vs AVHRR PATMOS | | | |
|---|---|---|---|---|---|---|---|---|---|---|---|---|
| | RMSE (km) | MAE (km) | MBE (km) | Correlation | RMSE (km) | MAE (km) | MBE (km) | Correlation | RMSE (km) | MAE (km) | MBE (km) | Correlation |
| 2015 | 2.00 | 1.08 | 0.33 | 0.86 | 2.90 | 1.80 | 1.33 | 0.75 | 3.05 | 2.03 | 1.23 | 0.69 |
| SL cases | 1.26 | 0.65 | -0.16 | 0.93 | 1.68 | 1.03 | 0.42 | 0.88 | 2.03 | 1.31 | 0.21 | 0.80 |
| ML Cases | 2.89 | 1.86 | 1.22 | 0.64 | 4.31 | 3.22 | 2.99 | 0.53 | 4.34 | 3.34 | 3.08 | 0.44 |

(c)

*Table 6. a) Mean Cloud top height (CTH) values for 2015 over the SO for Level 2 CTHs data MODIS (MYD), CloudSat-CALIOP (CC), AVHRR CMSAF (AC) and AVHRR PATMOS-x (AP). Statistics of the comparison of CC CTH with the COD limit > 0.5 is shown in b) and c) is for CC CTH without the COD limit.*

*To*

| Mean Values of CTHs | | | | | |
|---|---|---|---|---|---|
| | CloudSat_CALIOP (km) > 0.5 COD | CloudSat_CALIOP (km) without COD limit | AVHRR CMSAF | MODIS | AVHRR PATMOS-x (km) |
| 2015 | 3.92 | 5.48 | 5.15 | 4.12 | 4.25 |
| Single-layer cases | 3.43 | 3.86 | 4.00 | 3.42 | 3.63 |
| Multilayer Cases | 4.82 | 8.41 | 7.20 | 5.40 | 5.37 |

(a)

| Statistics | CC vs AVHRR CMSAF | | | | CC vs MODIS | | | | CC vs AVHRR PATMOS-x | | | |
|---|---|---|---|---|---|---|---|---|---|---|---|---|
| | RMSE (km) | MAE (km) | MBE (km) | Correlation | RMSE (km) | MAE (km) | MBE (km) | Correlation | RMSE (km) | MAE (km) | MBE (km) | Correlation |
| 2015 | 2.87 | 1.77 | -1.23 | 0.72 | 2.53 | 1.55 | -0.19 | 0.69 | 2.53 | 1.65 | -0.32 | 0.66 |
| SL Cases | 2.00 | 1.13 | -0.59 | 0.83 | 2.00 | 1.18 | 0.01 | 0.78 | 2.16 | 1.35 | -0.21 | 0.72 |
| ML cases | 3.98 | 2.94 | -2.38 | 0.50 | 3.27 | 2.21 | -0.58 | 0.52 | 3.09 | 2.19 | -0.54 | 0.49 |

(b)

| Statistics | CC vs AVHRR CMSAF | | | | CC vs MODIS | | | | CC vs AVHRR PATMOS-x | | | |
|---|---|---|---|---|---|---|---|---|---|---|---|---|
| | RMSE (km) | MAE (km) | MBE (km) | Correlation | RMSE (km) | MAE (km) | MBE (km) | Correlation | RMSE (km) | MAE (km) | MBE (km) | Correlation |
| 2015 | 2.00 | 1.08 | 0.33 | 0.86 | 2.90 | 1.81 | 1.36 | 0.75 | 3.01 | 2.00 | 1.23 | 0.70 |
| SL cases | 1.27 | 0.65 | -0.16 | 0.93 | 1.67 | 1.02 | 0.44 | 0.88 | 1.97 | 1.28 | 0.22 | 0.81 |
| ML Cases | 2.88 | 1.85 | 1.21 | 0.64 | 4.31 | 3.23 | 3.01 | 0.54 | 4.29 | 3.30 | 3.05 | 0.45 |

(c)

Table 6. a) Mean Cloud top height (CTH) values for 2015 over the SO for Level 2 CTHs data MODIS (MYD), CloudSat-CALIOP (CC), AVHRR CMSAF (AC) and AVHRR PATMOS-x (AP). Statistics of the comparison of CC CTH with the COD limit > 0.5 is shown in b) and c) is for CC CTH without the COD limit.

- Figure 2 [R1]: R1 commented, *'It seems strange to me that the collocation coverage for CMSAF and PATMOS-x are different since they are derived using the same AVHRR measurements.'*

The Figure 2 caption was amended to reflect the collocation similarity as below

*'Figure 2 The distribution of collocated Pixels for all 4 sensors. The collocation of MODIS and CloudSat-CALIOP (MODIS_CC) is shown in red, AVHRR CMSAF and CloudSat-CALIOP (AVHRRCMSAF_CC) is shown in blue and AVHRR PATMOS-x and CloudSat-CALIOP (AVHRRPATMOS_CC) is shown in green.'*

To

'Figure 2 The distribution of collocated pixels for all 4 sensors. The collocation of MODIS and CloudSat-CALIOP (MODIS_CC) is shown in red, AVHRR CMSAF and CloudSat-CALIOP (AVHRRCMSAF_CC) is shown in blue and AVHRR PATMOS-x and CloudSat-CALIOP (AVHRRPATMOS_CC) is shown in green. The colours used in the plot to indicate the same data selection for each sensor are different but overlapping locations make them appear visually similar.'

**Reply to individual comments from Reviewer 2**

*The paper is reasonably well organized and written. The study compares cloud properties derived from different platforms and algorithms over the Southern Ocean, which could be worthwhile if it leads to improved understanding of algorithm deficiencies and uncertainties. Unfortunately, I don't think the paper sheds new light on these things so the purpose for the paper isn't very clear to me. Furthermore, the results are confusing, don't make sense in some cases, and at times contradict previous findings. As a result, I find some of the results difficult to understand as presented and am concerned that the analysis may be flawed, perhaps due to collocation problems and sampling differences. I think the authors need to reevaluate and verify their methods to ensure the results are robust for all datasets.'*

We appreciate the reviewer's insightful feedback and constructive suggestions.

**Comment 1:** *Lines 57, 742: replace Minnis et al., 2011 with Minnis et al., 2020 as the 2020 reference is the correct one that describes the data used in the Hinkelman analysis*

The reference has been replaced with the suggested reference in Line 64 and 766

*Minnis, P., S. Sun-Mack, Y. Chen, F.-L. Chang, C. R. Yost, W. L. Smith, Jr., P. W. Heck, R. F. Arduini, S. Bedka, Y.Yi, G. Hong, Z. Jin, D. Painemal, R. Palikonda, B. Scarino, D. A. Spangenberg, R. Smith, Q. Z. Trepte, P. Yang, and Y. Xie, 2020: CERES MODIS cloud product retrievals for Edition 4, Part 1: Algorithm changes. IEEE Trans. Geosci. Remote Sens., doi: 10.1109/TGRS.2020.3008866.*

**Comment 2:** *Line 133: NASA should be NOAA*

*'NASA'* has been replaced with *'NOAA'*

**Comment 3:** *Line 224: how well do MODIS along track assessments compare to the 0.25 deg resampled L3 products?*

As the MODIS along-track is L2 data and the 0.25 resampled is L3, we have not done an intercomparison. But a preliminary comparison with the active sensors indicates that the MODIS along-track data is more in line with active sensor retrievals.

**Comment 4:** *Line 265: It isn't clear what bin the uncertain pixels were put. Please explain*

*" Similarly, MYD06 cloud mask, confidently clear (11), probably clear (10), uncertain (01) and cloudy (00) were classified into cloudy (0) and clear (1)."* Clear and probably clear in clear (1) and uncertain and cloudy in cloudy (0).

The sentence in the manuscript has been modified to bring clarity in Line 274. 'Similarly, MYD06 cloud masks were classified as cloudy (0) for both uncertain (01) and cloudy (00) and as clear (1) for both confidently clear (11) and probably clear (10).'

**Comment 5:** *Line 290-298. Check to see that the values in the text match those in the table. I think there are some discrepancies.*

Thanks for your comment. The values have been reviewed and matched to the table values. Mismatched values from text reviewed and corrected accordingly

> *'Table 2 summarises the comparisons, with mean CTH values in Table 2a and the statistics for the comparison with CALIOP in Table 2b by season and overall. It can be observed that the active sensor correlation with the passive sensors for the overall monthly CTHs is low with values ranging from 0.35 for CALIOP-AVHRR CMSAF (C-A) and 0.32 for CALIOP-MODIS (CM). The CTH difference between the C-A sensors for DJF ranges from -1 to 1 km for the study domain, while for C-M it ranges from 0 to 4 km. The MBE for passive with the active sensor is 0.46 km for C-A and 1.12 km for C-M. The RMSE for C-A is 2.33 km; for C-M, the value is 2.60 km; and for A-M, it is 1.23 km, which is consistent with both correlation and MBE trend. Overall, the MBE between the MODIS-AVHRR CMSAF (A-M) passive sensors is larger for the CLARA-A3 dataset (0.95 km), but the correlation is better (0.67). Overall, there is a positive mean bias error for the passive sensors AVHRR CMSAF and MODIS CTHs.'*

has been modified to Line 298-306,

'Table 2 summarises the comparisons, with mean CTH values in Table 2a and the statistics for the comparison with CALIOP in Table 2b by season and overall. It can be observed that the active sensor correlation with the passive sensors for the overall monthly CTHs is low with values ranging from 0.36 for CALIOP-AVHRR CMSAF (C-A) and 0.32 for CALIOP-MODIS (CM). The CTH difference between the C-A sensors for DJF ranges from -1 to 1 km for the study domain, while for C-M it ranges from 0 to 4 km. The MBE for passive with the active sensor is 0.46 km for C-A and 1.12 km for C-M. The RMSE for C-A is 2.35 km; for C-M, the value is 2.60 km; and for A-M, it is 0.98 km, which is consistent with both correlation and MBE trend. Overall, the MBE between the MODIS-AVHRR CMSAF (A-M) passive sensors is larger for the CLARA-A3 dataset (0.98 km), but the correlation is better (0.68). Overall, there is a positive mean bias error for the passive sensors AVHRR CMSAF and MODIS CTHs.'

**Comment 6:** *Line 315-316: What finding is consistent, the values or the relative differences with CALIOP? Bromwich 2012 attributed the lower CALIOP cloud fraction to sampling, right? What's the implication for your finding here? How does this support or contradict the fact that CALIOP is more sensitive to clouds than MODIS?*

We refer to the relative differences with CALIOP, rather than the absolute values. Bromwich (2012) indicates that CALIOP's reduced cloud fraction may arise from sampling constraints. This corroborates our result that disparities stem from both sensor sensitivity and sampling attributes.

**Comment 7:** *Line 347: remove 'this'*

The word 'this' has been removed.

*'The overall cloud fraction of liquid water clouds (hereafter CFL) and seasonal distributions are illustrated in this Figure 5.'*

has been modified to Line 355,

'The overall cloud fraction of liquid water clouds (hereafter CFL) and seasonal distributions are illustrated in Figure 5.'

**Comment 8:** *Line 355: you already stated this on line 349*

The suggestion was accepted, and the redundant sentence *'In the A-M seven-year monthly comparison, the overall correlation stands at a higher level (0.71) compared to both passive-active comparisons.'* was removed.

**Comment 9:** *Line 376: This is a great figure, but the results are suspicious, particularly for AP which looks like it may be from a different swath. You should double check this and make sure that a mistake hasn't been made in the collocation process.*

Rechecked and found that AP is using previous day swath information; hence, discarded the case study and identified a new case study. Please refer to Appendix Figure 2 and below for further information on the AP swath issue. Case studies with UTC times of 22:00 hours or later were discarded to avoid potential recurrence of the issue. This additional condition affected only about 5% of the collocated data and had a negligible effect on the overall analysis.

**Comment 10 :** *Line 381: Consider rephrasing this: "The COD shown on each product uses the passive sensors' 1.6 μm channel." Instead of 'uses', perhaps 'was derived using' would make more sense.*

The suggestion was accepted and amended accordingly. *'The COD shown on each product uses the passive sensors' 1.6 μm channel.'* was changed to

'The COD shown in each product was derived using the passive sensors' 3.7 μm channel'.

**Comment 11:** *Line 394-395: Why is AC sensitive to the midlevel cloud and MODIS not sensitive? Is this possibly the result of a space/time mismatch. Did you look at the BT's? Are you sure the sensors are viewing the same space?*

Thank you for your comment. The case study has been retracted, and a new case study introduced. With the withdrawal of the case study, the overall structure of the section has been revised. Hence, the comment is no longer applicable.

**Comment 12:** *Line 403-404: This statement about passive radiometers has not been proven here and is generally incorrect. It would be more appropriate to say "This suggests that the passive sensor algorithms with a single layer assumption are not well suited for distinguishing thin cirrus*

*in some overlapping cloud conditions.” Furthermore, why is CC indicating a cloud top phase of liquid water for the clouds at 6 and 8 km? Is that trustworthy? What's the temperature at those levels?*

With the withdrawal of the case study, the overall structure of the section has been revised. Nevertheless, regarding the classification of liquid cloud phase by CC, we hypothesise that it results from the phase classification of the second cloud layer. Since the top cloud is very thin, with an optical thickness less than 0.5, as indicated by the cyan CC CTH in Panel 1, the cloud phase of the underlying second layer is likely being considered.

**Comment 13:** *Line 405: clouds,*

With the withdrawal of the case study, the overall structure of the section has been revised. Hence, the comment is no longer applicable.

**Comment 14:** *Line 405-409: Here again, the results look suspicious. Why is MODIS systematically 2-3km too high. Are we learning anything here?*

Thank you for your comment. With the withdrawal of the case study, the overall structure of the section has been revised. Hence, the comment is no longer applicable.

**Comment 15:** *Line 420: convective cloud? Where is that?*

With the withdrawal of the case study, the overall structure of the section has been revised. The line has been removed.

**Comment 16:** *Line 441: Here and elsewhere, be careful with wording that suggests active sensors serve as ground-truth for COD (e.g. passive sensors underestimate…) The lidar, radar and passive all have different sensitivities to the cloud PSD's, and the active sensors can actually miss information in the vertical column.*

We acknowledge that active sensors such as CALIOP and CloudSat can miss information in the vertical column due to signal attenuation in optically thick clouds or reduced sensitivity near the surface. However, in our study, we have considered this as the reference (or "truth") for COD, consistent with previous literature and quality statements. Hence, our conclusions are drawn relative to these active sensor estimates, while recognising their inherent limitations. The sentence was modified to reflect the modified case study part.

**Comment 17:** *Line 449-461: This section is consistent with previous sections in that the results don't make a lot of sense and further call to question the veracity of the collocation process adopted here. Can the two AVHRR datasets have this much difference in skill? I don't believe I've ever seen a MODIS hit rate this low for daytime clouds. Aren't these results (e.g. for MODIS) also inconsistent with what you show in figure 4?*

Figure 4 shows L3 data retrievals. We have also analysed AVHRR CLARA 2.1 and the difference between the sensors was smaller and comparable. After the release of a new dataset of CLARA 3.0 in late 2023, we observed a significant increase in the CMSAF product correlation for cloud properties. The CMSAF team completely changed the algorithm for this version to a neural net

algorithm. As observed in Figure 4 MODIS has lesser CFC compared to AVHRR CMSAF and also Figure 4 presents Level 3 data averaged over 7 years. Our analysis is for a year for L2 data.

**Comment 18:** *Line 473: Why would the AC algorithm be more sensitive to high thin clouds than the MODIS algorithm given how much more information is available from MODIS? Just because the AC neural net produces better mean CTH's, it also leads to lot of height overestimates. How does this imply that AC is more sensitive to thin cirrus?*

Yes, we agree that the information content is much higher for MODIS than AC, however we surmise that the neural net algorithm was able to uncover some additional relationships to improve the accuracy of the retrieval. If MODIS adopted a similar approach, we would expect even better results given the extra channels.

**Comment 19:** *Line 475: You already stated on line 463 what Fig 8 shows.*

This statement is to redraw the attention of the readers to the figure and to explain in depth the different analysis results.

**Comment 20:** *Line 481-483: CTH 'overestimates' are due to inversions, where the BT matches the profile temperature at a 'lower' height. Do you mean higher height?*

*There is a good discussion of the inversion problem here (and references within):*

*Dong, X., P. Minnis, B. Xi, S. Sun-Mack, and Y. Chen (2008), Comparison of CERES-MODIS stratus cloud properties with ground-based measurements at the DOE ARM Southern Great Plains site, J. Geophys. Res., 113, D03204, doi:10.1029/2007JD008438.*

- new referenced added to the line and 'lower' changed to 'higher' *'An explanation for the overestimation of low CTH is the effect of atmospheric temperature inversion, where the cloud brightness temperature taken by the passive sensor from the NWP reanalyses profile might be for a lower height than the actual CTH (Figure 7a[i]).'*

Changed to line 467-469

An explanation for the overestimation of low CTH is the presence of a sharp temperature inversion that provides two solutions for the same IR temperature from the NWP reanalyse profile, one above and one below the actual CTH (Figure 7a[iii], Dong et al., 2008).

And references added to References section in Line 638

Dong, X., Minnis, P., Xi, B., Sun-Mack, S., and Chen, Y.: Comparison of CERES-MODIS stratus cloud properties with ground-based measurements at the DOE ARM Southern Great Plains site, Journal of Geophysical Research: Atmospheres, 113, 2008.

**Comment 21:** *Line 507-540: Section lacks discussion regarding difference across algorithms with respect to CER magnitudes (i.e. why is MODIS ice Re so much higher than the others), and CER distributions (i.e. why does AC ice look like AC water, unlike AP and MODIS)*

The discussion has been expanded and changed as part of the change in the MODIS retrieval channel change (3.7μm to 1.6um) as below (Line 493- 529)

**4.2.4 L2 Cloud Effective Radius Comparison**

This section compares CER from the passive sensors as a function of CPH and the presence of single or multilayer clouds. The passive sensors categorise thermodynamic phase into various classifications according to their retrieval algorithms; for example, MYD has five phase categories, while AC has eight, including a supercooled water class. The categories used by each dataset are presented in Table 7. In the results presented, the number of classes was reduced: classifications of water, fog, and supercooled liquid are grouped as liquid; ice, cirrus, and overlap are treated as ice. When this generalised CPH classification is used, more liquid-phase clouds are found in SL cases, while ice dominates in ML cases. A comparison between the different sensors AC, MYD, and AP is illustrated in Figure 9 for the year 2015, stratified for SL and ML cases, and summarised in Table 8.

The liquid CER probability density functions (PDFs) are in broad agreement across the three sensors for all cases. The mean values for liquid clouds are approximately 14 μm for AC, 13 μm for MYD, and 17 μm for AP. The higher CER values from AP suggest potential misclassification of liquid clouds as ice clouds as the proportion of ice clouds in AP is over five times higher than that of liquid clouds. The liquid phase occurs more frequently in SL clouds than ML clouds across all sensors, with similar mean values for both cases (Table 8).

In contrast, the CER comparisons for ice-phase clouds reveal notable differences across the sensors. The overall mean for AC is 15.57 μm, while MYD and AP report means of 20.90 μm and 20.40 μm, respectively. AP also exhibits a distinct bimodal CER distribution with peaks at approximately 10 μm and 30 μm, particularly evident in SL ice-phase retrievals. ML cases show more similar CER distributions across the sensors. The SL–ML CER difference is relatively small for AC (≈ 1μm), modest for AP (≈ 2 μm), and larger for MYD (≈ 5 μm). Sensitivity to ice crystal habit selection and the corresponding LUT structure also influences retrievals. MYD uses scattering functions derived from the General Habit Mixture (GHM) developed by Baum et al. (2005a, b) and Yang et al. (2013), implemented as band-specific LUTs (Platnick et al., 2016; Amarasinghe et al., 16 2017). The ice habitat selection includes severely roughened aggregates, hollow columns, bullet rosettes, plates, and droxtals. AC uses radiative transfer simulations based on narrower droplet size distributions and an ice crystal model including severely roughened aggregated solid columns, hollow columns, and bullet rosettes (Yang et al., 2013; Baum et al., 2011). AP uses LUTs from Baum/Yang scattering models with reduced spectral dimensionality , and its ice habit selection includes roughened bullet rosettes, aggregates, and hollow columns (Walther and Heidinger, 2012; Heidinger and Foster, 2021). These distinctions in habit

selection and LUT configuration influence the consistency and comparability of retrieved cloud microphysical properties across sensors.

Supercooled liquid clouds are consistently retrieved by both AC and AP, particularly in SL regimes. AC reports a substantial supercooled component in full-year statistics, with a mean CER of 14.65 μm, due to its extended phase classification (Karlsson et al., 2023a). AP also distinguishes supercooled liquid from warm liquid, frequently identifying it in SL clouds, with a slightly higher mean CER of 17.20 μm. MYD lacks a dedicated supercooled category but includes an "undetermined liquid" retrieval class (ULQ; mean 19.71 μm), likely encompassing supercooled or glaciating clouds not clearly resolved by its phase algorithm (Platnick et al., 2016). Both AC and AP indicate persistent supercooled liquid presence over the Southern Ocean, consistent with prior observational studies (Huang et al., 2015; Bodas-Salcedo et al., 2016).

The differences observed between ML and SL cases suggest that multilayer clouds influence retrieval outcomes. Yost et al. (2023) had similar findings and demonstrated that multilayer conditions, especially non-opaque clouds over liquid, are among the leading sources of phase retrieval uncertainty in passive satellite observations.

Figure 9 replaced from

*Figure 9 Cloud Effective Radius (CER) distribution according to the cloud phase for the collocated data for the year 2015, over the Southern Ocean, for all passive sensors.*

[Figure]

To

Figure 9 Cloud Effective Radius (CER) distribution according to the cloud phase for the collocated data for the year 2015, over the Southern Ocean, for all passive sensors.

[Figure]

*Table 8*

*From*

| Cloud Effective Radius | | | | | | | |
|---|---|---|---|---|---|---|---|
| **Phase** | **Sensors** | Full year | | single-layer | | Multi Layer | |
| | | Mean | Median | Mean | Median | Mean | Median |
| **Water** | AC | 14.75 | 13.37 | 14.01 | 13.45 | 14.19 | 12.66 |
| | MYD | 15.59 | 15.00 | 15.21 | 15.00 | 16.38 | 16.00 |
| | AP | 16.72 | 15.51 | 16.01 | 15.68 | 16.26 | 14.79 |
| **Ice** | AC | 15.54 | 14.19 | 14.82 | 14.80 | 14.50 | 13.06 |
| | MYD | 30.90 | 30.00 | 30.68 | 35.00 | 26.54 | 25.00 |
| | AP | 20.32 | 19.57 | 19.51 | 21.53 | 19.23 | 18.08 |
| **Supercooled Liquid** | AC | 14.68 | 13.34 | 13.98 | 13.60 | 14.24 | 12.86 |
| | MYD | nan | nan | nan | nan | nan | nan |
| | AP | 14.62 | 13.96 | 14.46 | 13.87 | 15.11 | 14.28 |

*Table 8. Mean and Median Cloud Effective Radius (CER) comparison for 2015 over the Southern Ocean, for water, ice and supercooled liquid phases from Level 2 AVHRR CMSAF (AC), AVHRR PATMOS-x (AP), MODIS (MYD) and CloudSat-CALIOP (CC).*

*To*

| Cloud Effective Radius [$\mu$m] | | | | | | | |
|---|---|---|---|---|---|---|---|
| **Phase** | **Sensors** | Full year | | single-layer | | Multi Layer | |
| | | Mean | Median | Mean | Median | Mean | Median |
| **Liquid** | AC | 14.75 | 13.36 | 14.81 | 13.45 | 14.16 | 12.59 |
| | MYD | 13.28 | 13.00 | 13.38 | 13.00 | 13.07 | 12.00 |
| | AP | 17.20 | 15.51 | 17.35 | 15.95 | 16.61 | 15.03 |
| **Ice** | AC | 15.57 | 14.24 | 16.20 | 14.84 | 14.53 | 13.15 |
| | MYD | 20.90 | 21.00 | 23.36 | 23.00 | 18.33 | 18.00 |
| | AP | 20.40 | 19.67 | 21.39 | 21.74 | 19.26 | 18.09 |
| **Supercooled Liquid** | AC | 14.65 | 13.31 | 14.91 | 13.59 | 14.18 | 12.78 |
| | MYD | nan | nan | nan | nan | nan | nan |
| | AP | 14.70 | 13.97 | 14.54 | 13.88 | 15.21 | 14.33 |

Table 8. Mean and Median Cloud Effective Radius (CER) comparison for 2015 over the Southern Ocean, for water, ice and supercooled liquid phases from Level 2 AVHRR CMSAF (AC), AVHRR PATMOS-x (AP), MODIS (MYD) and CloudSat-CALIOP (CC).

**Comment 22:** *Line 549-550: Why do you think the results disagree from Frey (2018) and Palm (2010)? Their results make more sense to me from a physical standpoint and from a retrieval difficulty standpoint..*

The disagreement with the results of Frey et al. (2018) and Palm et al. (2010) is likely due to differences in both the temporal coverage and the spatial extent of the studies. Our analysis is limited to a single year and focuses on a narrow 10° latitude band, whereas Frey et al. (2018) examine the Southern Ocean from 60°S to 82°S over a 10-year period, and Palm et al. (2010) focus on the Arctic region using five years of data. These differences in study domain and duration likely contribute to the contrasting results. The differences observed in our study likely reflect the limitations of short-term satellite observations and the challenges of retrievals over high-albedo surfaces such as sea ice, rather than a fundamental contradiction of their findings.

The discussion has been changed as part of the change in the MODIS retrieval channel (3.7µm to 1.6µm); Line 530-560.

**4.2.5 L2 cloud property comparison as a function of surface type**

In this section, we investigate the distribution of cloud properties as a function of sea ice coverage. The sea ice flag in AVHRR PATMOS-x(AP) retrieval was used to define the presence of sea ice, based on fractional sea ice coverage from NOAA/NCEP Climate Forecast System Reanalysis (CFSR) data (Foster et al., 2023). The distributions of the cloud properties CER, CTH, and COD were analysed as a function of sea ice coverage for latitudes > 60◦S, as shown in Figure 10. The narrow latitude range was selected to reduce the differences in microphysics caused by different meteorological regimes. The MYD cloud properties are shown in red, AC in black, and AP in yellow. Cloud properties such as

COD, CER, and CTH are compared for sea ice detected and non-sea ice detected; the mean and median are summarised in Table 9.

The mean COD for AC was 33.9 and 14.4; for MYD it was 24.1 and 18.6; and for AP it was 41.7 and 12.7 for sea ice and non-sea ice cases, respectively. The optical depth of the retrieved pixels increased in the presence of sea ice for all sensors. The results disagree with the findings of Frey et al. (2018) and Palm et al. (2010), which concluded that clouds were more frequent, and cloud optical depth tends to be higher over ice free regions. While the disagreement could arise from differences in latitude range and the limited temporal scope of our dataset compared to the multi-year analyses Frey et al. (2018) and Palm et al. (2010). It is hypothesised that the difference is due to two effects both originating in flaws of the retrieval algorithms: 1 - sea ice (with no associated cloud) is misidentified as cloud; this could be the cause of the spike in observations with greater than 100 COD over sea ice, and 2 - the albedo of sea ice used in the retrieval algorithms introducing a systematic positive COD 17 bias. It is difficult to define an accurate sea ice albedo; a systematic underestimation of the albedo could cause the optical depth of clouds to be overestimated.

In the comparison of CTH and CER between sea ice and non-sea ice cases, the differences for AP are small (0.1 µm for CER and -0.01 km for CTH). The MYD and AC products also show small positive differences in CER (≈1 µm) over sea ice. For non sea ice cases, the CTHs are 0.05 km (MYD) and 0.6 km (AC) lower than sea ice cases. Although CER and CTH exhibit positive differences, these were much smaller than the differences observed for COD.

The seasonality of the COD, CER, and CTH anomalies over sea ice was also examined. We observed that the disagreement between retrievals over sea ice and non sea ice is higher in JJA, where sea ice cases show (not shown here) lower COD and CTH values. COD values remain below 50 for AC and MYD and below 80 for AP across all seasons, with generally lower values over sea ice in winter (JJA). CER distributions are relatively consistent across seasons, ranging from 0–35 µm for AP, 5–65 µm for MYD, and 5–45 µm for AC. Seasonal variation is more evident in CTH: low clouds (less than 2 km) dominate for MYD and AP in JJA and for all three sensors in SON, while DJF and MAM show a greater presence of mid-level clouds (3–6 km), though low clouds remain prominent for MYD and AP. It can be concluded that in the case of CTH and CER, the seasonality is not an important factor, while for COD, the seasonality and presence of sea ice are important factors in the retrieval of the product.

Figure 10 replaced from

*Figure 10 Cloud Optical Depth (COD), Cloud Effective Radius (CER) and Cloud Top Height (CTH) as a function of sea ice Flag from AVHRR PATMOS-x for all passive sensors, over the*

To

[Figure]

Figure 10 Cloud Optical Depth (COD), Cloud Effective Radius (CER) and Cloud Top Height (CTH) as a function of Sea ice Flag from AVHRR PATMOS-x for all passive sensors, over the Southern Ocean less than 60 degrees S. MODIS (MYD) is shown in red, AVHRR CMSAF (AC) is shown in black and AVHRR PATMOS-x (AP) is shown in yellow.

*Table 9*

*From*

| Sensors | COD | | | | CER | | | | CTH | | | |
|---|---|---|---|---|---|---|---|---|---|---|---|---|
| | Sea ice | | Non Seaice | | Sea ice | | Non Seaice | | Sea ice | | Non Seaice | |
| | Mean | Median | Mean | median | Mean | Median | Mean | median | Mean | Median | Mean | median |
| MYD | 23.30 | 13.0 | 20.46 | 13.0 | 20.82 | 16.0 | 19.45 | 15.00 | 4.15 | 3.3 | 3.87 | 2.75 |
| ACM | 25.98 | 11.46 | 19.19 | 10.35 | 14.25 | 12.66 | 13.01 | 11.32 | 4.91 | 4.17 | 4.27 | 3.56 |
| AP | 28.30 | 13.31 | 17.37 | 9.952 | 16.38 | 13.89 | 14.55 | 11.94 | 3.85 | 3.95 | 3.67 | 3.79 |

*Table 9.Mean and Median values of Cloud Optical Depth (COD), Cloud Effective Radius (CER) and Cloud Top Height (CTH) over sea and over sea ice for Level 2 AVHRR CMSAF, AVHRR PATMOS-x and MODIS pixels at latitudes (> 60◦ S).*

To

| | COD | | | | CER [$\mu$m] | | | | CTH [km] | | | |
|---|---|---|---|---|---|---|---|---|---|---|---|---|
| | Sea ice | | Non Seaice | | Sea ice | | Non Seaice | | Sea ice | | Non Seaice | |
| Sensors | Mean | Median | Mean | median | Mean | Median | Mean | median | Mean | Median | Mean | median |
| MYD | 24.08 | 13.00 | 18.55 | 12.00 | 15.92 | 13.00 | 14.46 | 12.00 | 4.02 | 3.30 | 3.93 | 2.80 |
| ACM | 33.89 | 11.46 | 14.44 | 9.17 | 14.02 | 12.25 | 13.01 | 11.91 | 5.07 | 4.52 | 4.40 | 3.60 |
| AP | 41.71 | 24.28 | 12.70 | 8.37 | 15.64 | 13.37 | 15.54 | 12.94 | 3.74 | 3.62 | 3.75 | 3.85 |

Table 9. Mean and Median values of Cloud Optical Depth (COD), Cloud Effective Radius (CER) and Cloud Top Height (CTH) over sea and over sea ice for Level 2 AVHRR CMSAF, AVHRR PATMOS-x and MODIS pixels at latitudes (> 60$\circ$ S).

**Comment 23:** *Line 570-575. I'd prefer the following convention 'more than 20% larger', rather than 'was >20%'*

Thank you for your comments. I have changed the text from *'In comparison to CALIOP CFC, the AVHRR CMSAF CFC was > 20% and the MODIS CFC was > 10%. On the other hand, over the high latitudes (60$\circ$S to 70$\circ$ ), MODIS CFC was < 10% compared to CALIOP CFC. We observed that over Antarctica, CALIOP CFC was > 10% of MODIS CFC and > 20% of AVHRR CMSAF.'* to Line 569-572

' In comparison to CALIOP CFC, the AVHRR CMSAF CFC was 20% greater and the MODIS CFC was 10% greater. On the other hand, over the high latitudes (60$\circ$S to 70$\circ$), MODIS CFC was 10% less compared to CALIOP CFC. We observed that over Antarctica, CALIOP CFC was 10% greater than MODIS CFC and 20% greater than AVHRR CMSAF.' accommodate this suggestion.

**Comment 24:** *Line 611-612. This is an incomplete sentence.*

The suggestion was accepted and amended accordingly. Removed *"Since reanalysis and climate models utilise these passive sensor retrievals."*

**References:**

1. Baum, B. A., Heymsfield, A. J., Yang, P., and Bedka, S. T.: Bulk scattering properties for the remote sensing of ice clouds. Part I: Microphysical data and models, Journal of Applied Meteorology, 44, 1885–1895, 2005a.

2. Baum, B. A., Yang, P., Heymsfield, A. J., Platnick, S., King, M. D., Hu, Y., and Bedka, S. T.: Bulk scattering properties for the remote sensing of ice clouds. Part II: Narrowband models, Journal of Applied Meteorology, 44, 1896–1911, 2005b.

3. Bodas-Salcedo, A., Hill, P., Furtado, K., Williams, K., Field, P., Manners, J., Hyder, P., and Kato, S.: Large contribution of supercooled liquid clouds to the solar radiation budget of the Southern Ocean, Journal of Climate, 29, 4213–4228, 2016.

4. Dong, X., Minnis, P., Xi, B., Sun-Mack, S., and Chen, Y.: Comparison of CERES-MODIS stratus cloud properties with ground-based measurements at the DOE ARM Southern Great Plains site, Journal of Geophysical Research: Atmospheres, 113, 2008.

5. Henderson, D. and L'Ecuyer, T.: level 2B fluxes and heating rates with lidar [2B-FLXHR-LIDAR] process description and interface control document, CloudSat Project Rep,https://www.cloudsat.cira.colostate.edu/cloudsat-static/info/dl/2b-flxhr-lidar/2B-FLXHR-LIDAR_PDICD.P2_R05.rev0.pdf, 2023.

6. Heidinger, A. K., Foster, M. J., Walther, A., and Zhao, X. T.: THE PATHFINDER ATMOSPHERES–EXTENDED AVHRR CLIMATE DATASET, Bulletin of the American Meteorological Society, 95, 909–922, http://www.jstor.org/stable/26218976, 2014

7. Huang, Y., Siems, S. T., Manton, M. J., Protat, A., and Delano., J.: A study on the low-altitude clouds over the Southern Ocean using the DARDAR-MASK, Journal of Geophysical Research: Atmospheres, 117, 2012.

8. Huang, Y., Protat, A., Siems, S. T., and Manton, M. J.: A-Train observations of maritime midlatitude storm-track cloud systems: Comparing the Southern Ocean against the North Atlantic, Journal of Climate, 28, 1920–1939, 2015.

9. Platnick, S., King, M. D., Meyer, K. G., Wind, G., Amarasinghe, N., Marchant, B., Arnold, G. T., Zhang, Z., Hubanks, P. A., Ridgway, B., et al.: MODIS cloud optical properties: User guide for the Collection 6 Level-2 MOD06/MYD06 product and associated Level-3 Datasets, Version, 1, 145, 2015.

10. Minnis, P., Sun-Mack, S., Chen, Y., Chang, F.-L., Yost, C. R., Smith, W. L., Heck, P. W., Arduini, R. F., Bedka, S. T., Yi, Y., et al.: CERES MODIS cloud product retrievals for edition 4—Part I: Algorithm changes, IEEE Transactions on Geoscience and Remote Sensing, 59, 2744–2780, 2020.

11. Müller, R., Haussler, S., and Jerg, M.: The role of NWP filter for the satellite based detection of cumulonimbus clouds, Remote Sensing, 10, 386, 2018.

12. Noh, Y.-J., Seaman, C. J., Vonder Haar, T. H., Hudak, D. R., and Rodriguez, P.: Comparisons and analyses of aircraft and satellite observations for wintertime mixed-phase clouds, Journal of Geophysical Research: Atmospheres, 116, 2011.

13. Truong, S., Huang, Y., Lang, F., Messmer, M., Simmonds, I., Siems, S., and Manton, M.: A Climatology of the Marine Atmospheric Boundary Layer Over the Southern Ocean from Four Field Campaigns During 2016–2018, Journal of Geophysical Research: Atmospheres, 125, https://doi.org/10.1029/2020JD033214, 2020.

14. Wall, C. J., Kohyama, T., and Hartmann, D. L.: Low-cloud, boundary layer, and sea ice interactions over the Southern Ocean during winter, Journal of Climate, 30, 4857–4871, 2017.

15. Walther, A. and Heidinger, A. K.: Implementation of the daytime cloud optical and microphysical properties algorithm (DCOMP) in PATMOS-x, Journal of Applied Meteorology and Climatology, 51, 1371–1390, 2012.

16. Wang, Z., Vane, D., Stephens, G., and Reinke, D.: CloudSat Project: Level 2 combined radar and lidar cloud scenario classification product process description and interface control document, California Institute of Technology, Calif, p. 61, 2013.

17. Wang, Z.: CloudSat 2B-CLDCLASS-LIDAR product process description and interface control document, Process Description and Interface Control Document (PDICD) P1_R05, CloudSat Project Rep., https://www.cloudsat.cira.colostate.edu/cloudsat-static/info/dl/2b-cldclass-lidar/2B-CLDCLASS-LIDAR_PDICD.P1_R05.rev0_.pdf (last access: 18 February 2024), 2019.

18. Wu, W., Liu, Y., and Betts, A. K.: Observationally based evaluation of NWP reanalyses in modeling cloud properties over the Southern Great Plains, Journal of Geophysical Research: Atmospheres, 117, 2012.

19. Yang, P., Bi, L., Baum, B. A., Liou, K.-N., Kattawar, G. W., Mishchenko, M. I., and Cole, B.: Spectrally consistent scattering, absorption, and polarization properties of atmospheric ice crystals at wavelengths from 0.2 to 100 $\mu$ m, Journal of the atmospheric sciences, 70, 330–347, 2013.
20. Yost, C. R., Minnis, P., Sun-Mack, S., Smith Jr, W. L., and Trepte, Q. Z.: VIIRS Edition 1 Cloud Properties for CERES, Part 2: Evaluation with CALIPSO, Remote Sensing, 15, 1349, 2023.

**Appendix**

**1. Collocation**

Below is a figure to demonstrate that all four sensors were collocated appropriately.

[Figure]

a)

b)

[Figure]

c)

**Appendix Figure 1:** a) Collocation of all sensors, b) Each sensor pixel distribution, and c) 3D plot for the collocation for each individual passive sensor with the active sensor.

**2. Case Study Swaths for Clarifications**

The figures below display the swaths for the replaced case study 10-01-2015 UTC. Appendix Figure 2 shows swaths from the previous day, while Appendix Figure 3 presents the case study day's swath.

[Figure]

**Appendix Figure 2**: Cloud top height swaths of AVHRR PATMOS-x, AVHRR CMSAF and MODIS for 09-01-2015 UTC.

[Figure]

**Appendix Figure 3:** Cloud top height swaths of AVHRR PATMOS-x, AVHRR CMSAF and MODIS for 10-01-2015 UTC.

It's clearly visible that AVHRR PATMOS-x uses the previous day swath. Hence the case study was discarded.

**3. New Case Study Swaths for COD and CER**

The below figures show the cloud optical depth and cloud effective radius swaths for case study 19-10-2015 UTC.

[Figure]

**Appendix Figure 4:** Cloud optical depth swaths of a) MODIS, b) AVHRR PATMOS-x and c) AVHRR CMSAF for 19-10-2015 UTC.

[Figure]

**Appendix Figure 5:** Cloud effective radius swaths of a) MODIS, b) AVHRR PATMOS-x and c) AVHRR CMSAF and for 19-10-2015.